# Explicit representation of germline and non-germline residues improves antibody language modeling

Jeonghyeon Kim [1]   Nathaniel Blalock [2]   Ameya Kulkarni [1]   Kensuke Nakamura [3]   Philip A. Romero [1]

## Abstract

Antibodies originate from germline templates and are diversified by somatic hypermutation, producing sequences in which conserved germline residues scaffold structure while rare non-germline (NGL) substitutions refine antigen binding. Current antibody language models (ALMs) treat all residues equivalently and inherit a germline bias that systematically down-weights functionally critical NGL mutations as statistical noise. We introduce PRISM, a germline-aware ALM that explicitly represents germline and non-germline residues as distinct token types over a factorized 53-token vocabulary. PRISM achieves state-of-the-art pseudo-perplexity in hypervariable CDRs and is uniquely positively correlated with experimental binding affinity across three deep mutational scanning landscapes on which all compared ALMs anti-correlate. The dual-vocabulary further enables property-specific controllable generation previously unattainable with entangled ALMs. NGL-directed sampling improves physics-based binding scores while GL-directed sampling preserves stability and solubility. These results establish disentangled germline/non-germline representation as a substantive advance in antibody language modeling.

## 1. Introduction

While general protein language models (pLMs) have established a foundation for biological sequence modeling (Lin et al., 2023; Madani et al., 2023), the unique generative constraints of immunoglobulins necessitate specialized anti-body language models (ALMs) (Olsen et al., 2022; Shuai et al., 2023). However, even these specialized architectures face a fundamental statistical challenge: antibody generation is a hierarchical process combining static germline templates with dynamic diversification mechanisms (Murugan et al., 2012). While framework regions largely maintain conservation, functional specificity is driven by non-germline (NGL) residues from somatic hypermutation (SHM) or junctional diversity. These mutations optimize binding energy, but standard ALMs often fail to model them explicitly.

A critical bottleneck is the severe data imbalance in natural repertoires. Standard ALMs suffer from a "germline bias" since over 90% of residues remain identical to their germline progenitors, conflating evolutionary conservation with biochemical identity (Olsen et al., 2024). Consequently, standard ALMs systematically underweight NGL residues, treating critical binding-determining mutations as noise. This results in generated sequences that are structurally plausible but functionally conservative, lacking the targeted diversity required for high-affinity binding (Hie et al., 2024).

To address this, we introduce PRISM (**P**artitioning **R**esidue **I**dentity in **S**omatic **M**aturation), a framework designed to intentionally decouple germline conservation from functional variation (Figure 1A). Unlike approaches that simply fine-tune existing architectures, PRISM formalizes sequence generation as a factorized inference problem: $P(\text{token}) = P(\text{origin}) \times P(\text{aa} \mid \text{origin})$. By utilizing a custom 53-token vocabulary that strictly separates NGL residues from germline templates, PRISM enables explicit control over the generative source. This holistic design achieves state-of-the-art pseudo-perplexity on hypervariable regions and enables controllable generation for targeted antibody design.

**Conflict of Interest Disclosure** K.N. is employed by Daiichi Sankyo Co., Ltd.; the company had no role in study design, analysis, or the decision to publish. The remaining authors declare no competing interests.

## 2. Preliminaries: Antibody Generation

Antibody variable domains are generated through two evolutionary phases. **V(D)J recombination** (Tonegawa, 1983)

[1]Department of Biomedical Engineering, Duke University, Durham, NC, USA [2]Department of Chemical and Biological Engineering, University of Wisconsin-Madison, Madison, WI, USA [3]Daiichi Sankyo Co., Ltd., Tokyo, Japan.. Correspondence to: Philip Romero <philip.romero@duke.edu>.

*Proceedings of the $43^{rd}$ International Conference on Machine Learning*, Seoul, South Korea. PMLR 306, 2026. Copyright 2026 by the author(s).

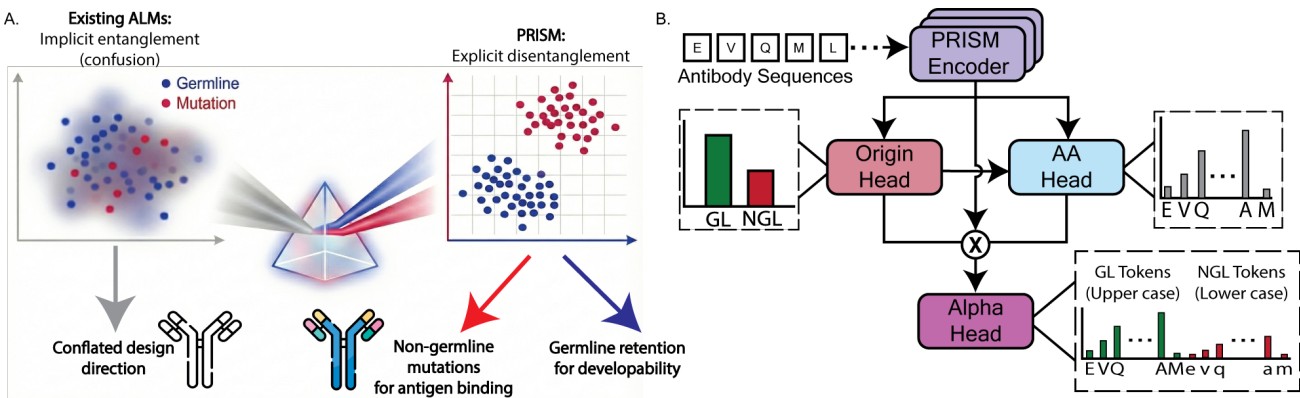

*Figure 1.* **The PRISM Framework. (A)** From Implicit Entaglement to Decomposition: Unlike existing ALMs that entangle conservation and variation, PRISM explicitly disentangles these factors. **(B)** Origin-Conditioned LM Head: PRISM employs an Origin-Conditioned LM Head in which the Origin Head first predicts the probability of NGL deviation, gating the AA Head's predictions over a factorized 53-token vocabulary.

combinatorially joins V/D/J germline DNA segments to form the framework and initial CDRs against largely static templates (Chothia & Lesk, 1987). Subsequent **somatic hypermutation** (SHM) during affinity maturation (Victora & Nussenzweig, 2012) introduces point mutations, termed **non-germline (NGL)** variants, at a rate $10^6\times$ background (Muramatsu et al., 2000), concentrated in the CDR loops and evolutionarily selected for higher binding affinity.

**Problem formulation.** Standard pLMs treat the observed sequence $x = (x_1, \ldots, x_L)$ as a flat string, but biologically each token $x_i$ is a realization of either the germline template $G$ or a non-germline variant $M$:

$$x_i \sim \begin{cases} P_{\text{germ}}(x_i|\text{gene}) & \text{if conserved,} \\ P_{\text{non-germline}}(x_i|x_{\text{context}}) & \text{if mutated (NGL).} \end{cases} \quad (1)$$

Existing models conflate these distributions. PRISM disentangles them explicitly.

## 3. PRISM Framework

Standard protein language models implicitly entangle the evolutionary origin of a residue with its physicochemical identity, leading to a strong bias toward germline sequences. Recent approaches, such as AbLang2 (Olsen et al., 2024), have attempted to mitigate this by employing reweighted loss functions (e.g., Focal Loss) to emphasize rare nongermline variants. However, these methods operate primarily at the optimization level, leaving the underlying representation entangled.

PRISM advances this paradigm by enforcing a fundamental disentanglement at the *representational* level. By structurally separating evolutionary origin from biochemical identity, PRISM enables emergent controllability over the generation process. This allows us to treat evolutionary divergence not merely as a prediction target, but as a tun-

able constraint. We adopt the architectural specifications of ESM-2 (Lin et al., 2023) (35M parameters, 12 layers) as our backbone, replacing its standard masked-LM head with our novel **Origin-Conditioned LM Head** — a three-component output module (Origin Head, AA Head, Alpha Head) in which the Origin Head conditions the AA Head and an Alpha Head learns to gate the evolutionary signal (Figure 1B).

### 3.1. Factorized Vocabulary

Based on the high-confidence non-germline mutation signals curated in our dataset (Appendix A), we construct an extended vocabulary $\mathcal{V}_{\text{ext}}$ of size 53 to support disentanglement (Appendix B.2). Uppercase tokens represent germline (GL) residues matching the template, while lowercase tokens denote non-germline (NGL) variants (2). This distinction allows the model to structurally separate evolutionary origin from biochemical identity.

### 3.2. Origin-Conditioned LM Head

Let $\mathbf{H} \in \mathbb{R}^{L \times d}$ denote the output of the encoder, where $L$ is the sequence length and $d = 480$ is the embedding dimension. The Origin-Conditioned LM Head is a three-component output module — Origin Head, AA Head, Alpha Head — in which the evolutionary prediction explicitly conditions the biochemical prediction.

**Origin Head (Evolutionary Prior):** First, we predict the probability of non-germline status at position $i$. The Origin Head projects the contextual embedding $\mathbf{h}_i$ to a scalar logit $o_i$:

$$o_i = \mathbf{w}_{\text{orig}}^\top \mathbf{h}_i + b_{\text{orig}} \in \mathbb{R} \quad (2)$$

The probability of a non-germline (NGL) origin is given by $\pi_i = \sigma(o_i)$, where $\sigma$ refers to sigmoid function.

**Amino Acid (AA) Head:** This head predicts the fundamental biochemical identity of the residue given the evolutionary context defined by $\mathbf{h}_i^{\text{cond}}$. It projects the embedding into the base vocabulary space:

$$\mathbf{l}_i^{\text{aa}} = \text{LogSoftmax}(\mathbf{W}_{\text{aa}}\mathbf{h}_i^{\text{cond}} + \mathbf{b}_{\text{aa}}) \in \mathbb{R}^{|\mathcal{V}_{\text{base}}|} \quad (3)$$

**Gradient-Detached Conditioning:** To ensure the Origin Head is optimized solely by evolutionary signals, we apply a stop-gradient operator (SG) before conditioning the downstream heads:

$$\mathbf{h}_i^{\text{cond}} = \mathbf{h}_i + \text{Linear}(\text{SG}(o_i)) \quad (4)$$

Here, $\text{SG}(\cdot)$ acts as an identity function during the forward pass but blocks error propagation during the backward pass. We empirically verify in Appendix K that this prevents the AA Head from delegating NGL prediction to the Origin Head, which results in a $\sim 2\times$ improvement in validation NGL perplexity and does not introduce early-training instability relative to ablation experiments.

**Alpha Head (Task-Relevance Gating):** To modulate the influence of the evolutionary prior, we introduce a scalar gating mechanism. This head predicts a position-specific mixing coefficient $\alpha_i$, representing a position-specific gate on the evolutionary prior:

$$\alpha_i = \sigma(\mathbf{w}_\alpha^\top \mathbf{h}_i^{\text{cond}} + b_\alpha) \in [0, 1] \quad (5)$$

Although the Origin Head already predicts $\pi_i$, a separate Alpha Head is architecturally necessary as it absorbs the gradient pressure from the language modeling loss (thereby preserving the Origin Head as a pure evolutionary classifier) and admits the evolutionary prior independently of Origin Head accuracy. Appendix J shows this is isomorphic to a position-specific tempered posterior whose tempering exponent is *learned end-to-end*, and that empirically $\alpha_i$ behaves as a *task-relevance gate* — saturating in hypermutation-prone CDRs and collapsing at IMGT-conserved structural anchors (Cys23, Trp41, Cys104, Phe/Trp118) — rather than a confidence decay on the Origin Head.

### 3.3. Alpha-Gated Logit Construction

The final distribution over the 53-token vocabulary is constructed via multiplicative gating in log-space. For any target token $y$, let $\mathcal{M}(y)$ be its biochemical identity index and $\mathcal{Z}(y) \in \{GL, NGL\}$ be its evolutionary class. The un-normalized logit $s_{i,y}$ is formulated as:

$$s_{i,y} = \underbrace{\mathbf{l}_{i,[\mathcal{M}(y)]}^{\text{aa}}}_{\text{Biochemical Prior}} + \underbrace{\alpha_i \cdot \log P_{\text{orig}}(\mathcal{Z}(y)|\mathbf{h}_i)}_{\text{Gated Evolutionary Constraint}} \quad (6)$$

where $P_{\text{orig}}(NGL|\mathbf{h}_i) = \pi_i$ and $P_{\text{orig}}(GL|\mathbf{h}_i) = 1 - \pi_i$.

This mechanism admits the Origin pathway only where it carries incremental information for residue identity, collapsing to the marginal AA distribution at structurally constrained sites, and provides a controllable knob for evolutionary exploration. Equation 6 implies a natural Bayesian interpretation as a position-specific tempered posterior, which we analyze formally in Appendix J. Finally, we apply temperature scaling with a factor $T = 0.5$ to sharpen the distribution for therapeutic engineering applications:

$$P(y_i = k) = \frac{\exp(s_{i,k}/T)}{\sum_{j \in \mathcal{V}_{\text{ext}}} \exp(s_{i,j}/T)} \quad (7)$$

### 3.4. Training Objective and Data

**Multi-head training objective.** PRISM is trained with a weighted combination of three masked-token losses corresponding to the architectural decomposition described in Sec. 3.1–3.3: a *final-distribution loss* $\mathcal{L}_{\text{final}}$ over the 53-token alpha-gated logits (Eq. 6), an *AA Head loss* $\mathcal{L}_{\text{AA}}$ over the 20-canonical amino-acid prediction, and an *Origin Head loss* $\mathcal{L}_{\text{origin}}$ over the binary GL/NGL label,

$$\mathcal{L}_{\text{total}} = \lambda_f \mathcal{L}_{\text{final}} + \lambda_{\text{AA}} \mathcal{L}_{\text{AA}} + \lambda_o \mathcal{L}_{\text{origin}}, \quad (8)$$

with $\lambda_f = 2.0$, $\lambda_{\text{AA}} = 1.0$, $\lambda_o = 1.5$. The two amino-acid losses ($\mathcal{L}_{\text{final}}, \mathcal{L}_{\text{AA}}$) are computed as a per-token-weighted focal loss (Lin et al., 2017; Olsen et al., 2024) over masked positions; to counteract the severe germline imbalance ($>90\%$ of residues are GL), every NGL position receives a per-token weight $\alpha_{\text{NGL}} = 3.0$ while GL positions retain weight $1.0$, applied *inside* the focal-loss reduction so that gradients from rare somatic substitutions are amplified at the token level rather than at the sequence level. This per-token reweighting is the loss-side counterpart to the dual-vocabulary disentanglement: it prevents the dominant germline signal from washing out the rare NGL events whose modeling fidelity ultimately drives binding-affinity prediction (Sec. 6.1). Crucially, this loss-side reweighting is *not* what drives the representational disentanglement reported in Sec. 7: Ablation 2 retains the identical focal loss with $\alpha_{\text{NGL}} = 3.0$ reweighting and the extended 53-token vocabulary while replacing only the Origin-Conditioned LM Head with a simple LM head, yet collapses to near-random GL/NGL separability (ARI 0.009 vs. PRISM Full's 0.162) — the architectural factorization is separable from, and not substitutable by, the loss reweighting.

**Region-aware masking.** Standard 15% uniform masking under-samples the hypervariable loops where NGL substitutions concentrate. We therefore apply region-conditional masking probabilities – 50% on CDRs, 30% on framework regions, and 15% background – so the model is forced to reconstruct the very positions whose supervision signal is statistically rarest, again coupling the training distribution

to the GL/NGL contribution PRISM is designed to recover (full schedule and ablation in Appendix C).

**Data.** We train on antibody sequences from the Observed Antibody Space (OAS) database (Kovaltsuk et al., 2018). Two corpora are used in a two-stage curriculum: an *unpaired* corpus of $\sim 58M$ heavy and $\sim 8.5M$ light chains for pretraining, and a higher-quality *paired* corpus of 619,675 heavy/light pairs for finetuning. To mitigate germline bias at the dataset level, we exclude near-germline "noise-floor" sequences using a chain-specific naive-B-cell mutation threshold ($\tau_{HC} = 3$, $\tau_{LC} = 2$ corresponding to the 90th percentile of mutation counts in naive paired sequences), shifting the training distribution toward sequences carrying meaningful somatic variation while retaining $> 80\%$ of memory B-cell sequences (Appendix Fig. S1). Train/validation/test splits are constructed via Linclust (Steinegger & Söding, 2018) clustering at 100% CDR3 identity and 95% whole-sequence identity, with whole clusters assigned to a single split, eliminating clonally related leakage between train and test (test set $\sim 22,600$ sequences). All hyperparameters, gene/region embedding details, and the full filtering pipeline are deferred to Appendix A–C.

**Downstream evaluation data.** We evaluate zero-shot binding affinity on three Deep Mutational Scanning (DMS) sets spanning a broad complexity spectrum: the single-mutant scan of **G6.31** ($N = 4,274$, anti-VEGF, $K_d \approx 0.4$ nM (Koenig et al., 2017a)), the $\sim 16$-site combinatorial **CR9114** ($N = 65,093$, anti-influenza HA (Phillips et al., 2021)), and the $\sim 10$-site combinatorial **Trastuzumab** ($N = 36,496$, anti-HER2 (Mason et al., 2021)); plus the 41-antibody **FLAb2** binding panel (Chungyoun & Gray, 2025) for breadth. Developability is evaluated on the 246-antibody Arsiwala/Marks biophysical panel (Arsiwala et al., 2025) (Self-Interaction, Hydrophobicity, Thermal Stability, Polyreactivity, Expression, Immunogenicity), clinical Anti-Drug-Antibody response rates (Marks et al., 2021), and the FLAb2 developability assays (DSC, AC-SINS, HEK, PSR, ADA). For controllable generation (Sec. 5), the same three DMS antibodies (G6.31, CR9114, Trastuzumab) provide WT structural templates (PDB 2FJH, 4FQI, 1N8Z) for Rosetta interface and stability scoring.

## 4. Disentanglement Improves Antibody Modeling

We evaluate PRISM against general pLMs (ESM2-35M, ESM2-650M (Lin et al., 2023)) and specialized ALMs (the masked AbLang2 (Olsen et al., 2024), AntiBERTy (Ruffolo et al., 2021), Sapiens (Prihoda et al., 2022), and the autoregressive generative IgLM (Shuai et al., 2023)), and show that PRISM's factorized architecture jointly delivers explicit GL/NGL disentanglement and superior predictive performance on functional residues.

### 4.1. Explicit Separation in Latent Space

Linear probing on frozen residue embeddings (Fig. 2A) shows PRISM achieving near-perfect GL/NGL separability (PR-AUC 0.980, F1 0.896; Fig. 2B), substantially above ESM2-35M (PR-AUC 0.354), AntiBERTy (0.929), and the generation-oriented IgLM (0.324, F1 0.312); the zero-shot Origin Head (0.958) already rivals trained probes. UMAP projections after residue-type mean centering (Appendix D) confirm this geometrically (Fig. 2C): PRISM exhibits a clear GL/NGL cleavage (ARI 0.172), whereas baselines remain entangled (ESM2-35M 0.009, IgLM 0.004).

### 4.2. Sequence Modeling Performance Stratified by Structure and Origin

We assess generative capability via Marginalized PPL (summing GL+NGL probabilities for fair comparison with standard-vocabulary baselines) and Exact PPL (Fig. S3E), which penalizes origin misclassification. On whole sequences (Fig. 2E) PRISM attains the lowest marginalized PPL on both heavy (1.54) and light (1.25) chains, beating ESM2-35M ($\sim 3.0$), AbLang2 ($\sim 1.6$), and IgLM (1.73/1.26). On the hypervariable CDR3-NGL residues that drive specificity, PRISM reaches 6.15 on the heavy chain versus AntiBERTy 24.08, AbLang2 9.26, and IgLM 25.12 – the last being particularly notable since IgLM is explicitly optimized for antibody generation yet still struggles on the loops defining specificity. On conserved framework regions, PRISM maintains parity with specialized baselines ($\sim 1.18$), confirming that disentanglement enables targeted CDR diversity without compromising scaffold developability (full breakdown in Appendix F).

## 5. Controllable Generation via Disentangled Logits

PRISM's disentangled representation also yields a directly controllable generative interface: by selecting which slice of the alpha-gated 53-token logit drives sampling, the same model is steered toward distinct biophysical objectives without task-specific finetuning. **Protocol.** For each WT antibody, we identify the 20 positions whose WT residue receives the lowest PRISM logit – the positions most amenable to substitution – and for each generated variant, draw 10 of them without replacement, mask each, and resample from the masked-LM distribution under one of three channels: **PRISM-Full** (Eq. 6), **PRISM-NGL** (NGL tokens only), or **PRISM-GL** (GL tokens only). Baselines (ESM2-35M, AbLang2, IgLM) are generated from their native distributions under the same position selection. Variants ($n_{mut} = 10$, 100 per antibody per method) are scored along four axes: Rosetta interface $\Delta\Delta G$ (physics-based binding), a ridge-based predictor trained on each antibody's

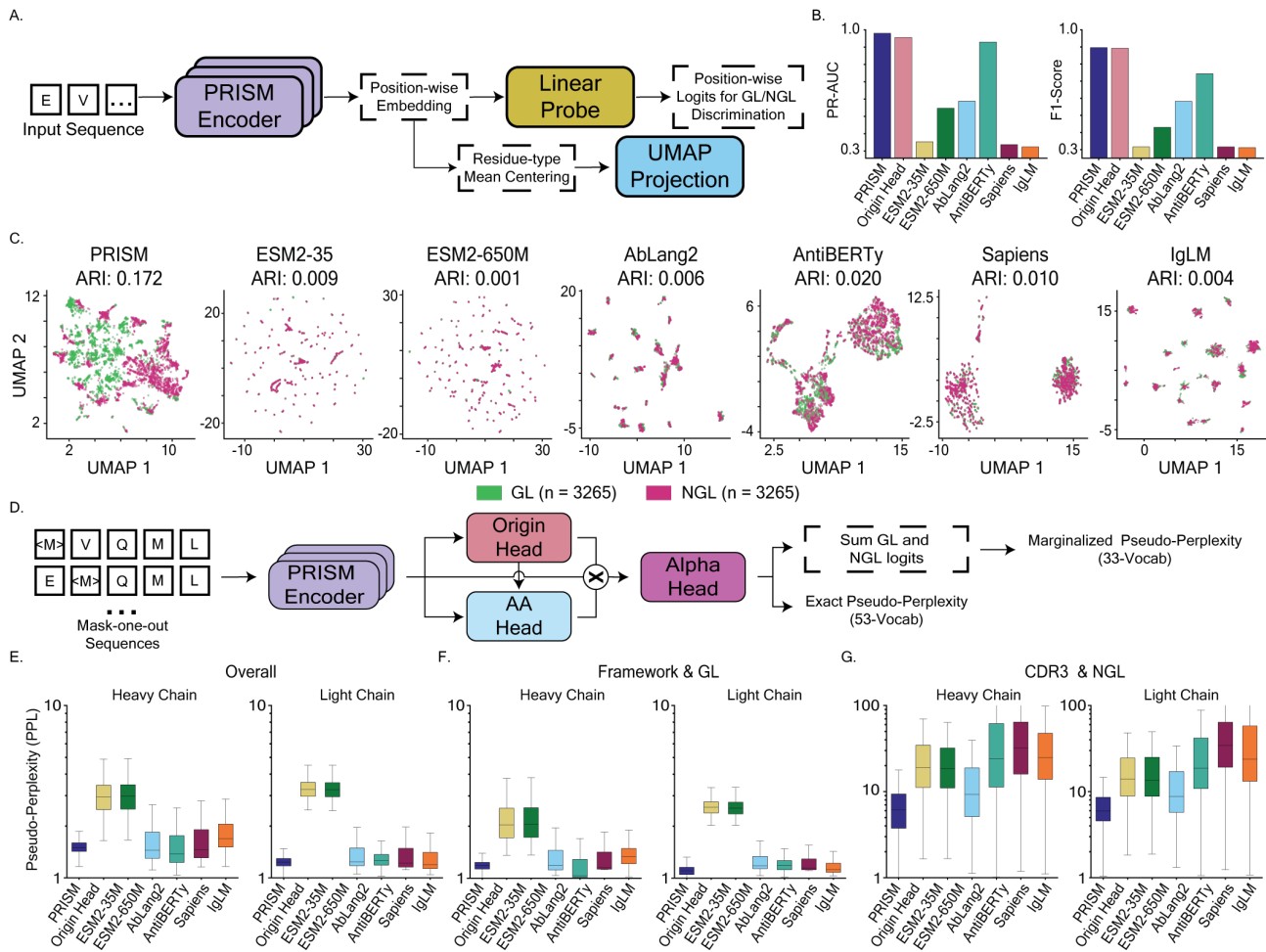

*Figure 2.* **PRISM achieves explicit disentanglement and superior generative performance.** **(A-C)** Linear probing and UMAP projections demonstrate that PRISM achieves near-perfect separation of GL and NGL residues (PR-AUC 0.980, ARI 0.172), whereas both masked ALMs and the generative IgLM show entangled distributions (ARI ≤ 0.020). **(D-G)** Generative evaluation via Pseudo-perplexity confirms this structural advantage. **(E)** PRISM achieves the lowest marginalized PPL overall across heavy and light chains. **(F)** While maintaining parity on stable Framework regions, **(G)** the model dramatically outperforms baselines on hypervariable CDR3-NGL residues, including the generation-specialized IgLM (6.15 vs. 25.12 on heavy chain).

DMS data (sequence-based binding), Rosetta antibody-only $\Delta\Delta G$ (stability), and CamSol pH 7.0 (solubility). The full methodology is in Appendix E.

The dual-logit decomposition surfaces a clean separation of regimes (Fig. 3B–D): **PRISM-NGL** produces the strongest binding-favorable variants on the Rosetta interface metric across all three antibodies, while **PRISM-GL** dominates on Rosetta stability and CamSol. For G6.31, where the site-saturated DMS renders the ridge-based predictor most reliable, PRISM-NGL leads on both the ridge-based predictor and Rosetta interface, providing the cleanest evidence that NGL-biased sampling captures genuine binding-relevant variation.

For CR9114 and Trastuzumab, the ridge-based predictor ranks PRISM-NGL slightly below some baselines, but the

Rosetta interface signal, independent of the DMS coverage limitations, still places PRISM-NGL on top, and the PRISM-GL advantage on stability and CamSol is consistent across all three antibodies; per-variant distributions and spatial mutation patterns (Appendix Figs. S4, S5) confirm this geometrically, with PRISM-NGL concentrating substitutions in CDRs and PRISM-GL spreading them through framework-adjacent sites.

**PRISM-Full** lies between these specialized modes but compares favorably against all generative baselines on the aggregate of binding, stability, and solubility. Together, these results establish PRISM as a controllable generator with two practically useful modes—NGL for affinity and GL for developability—and a strong unconstrained default.

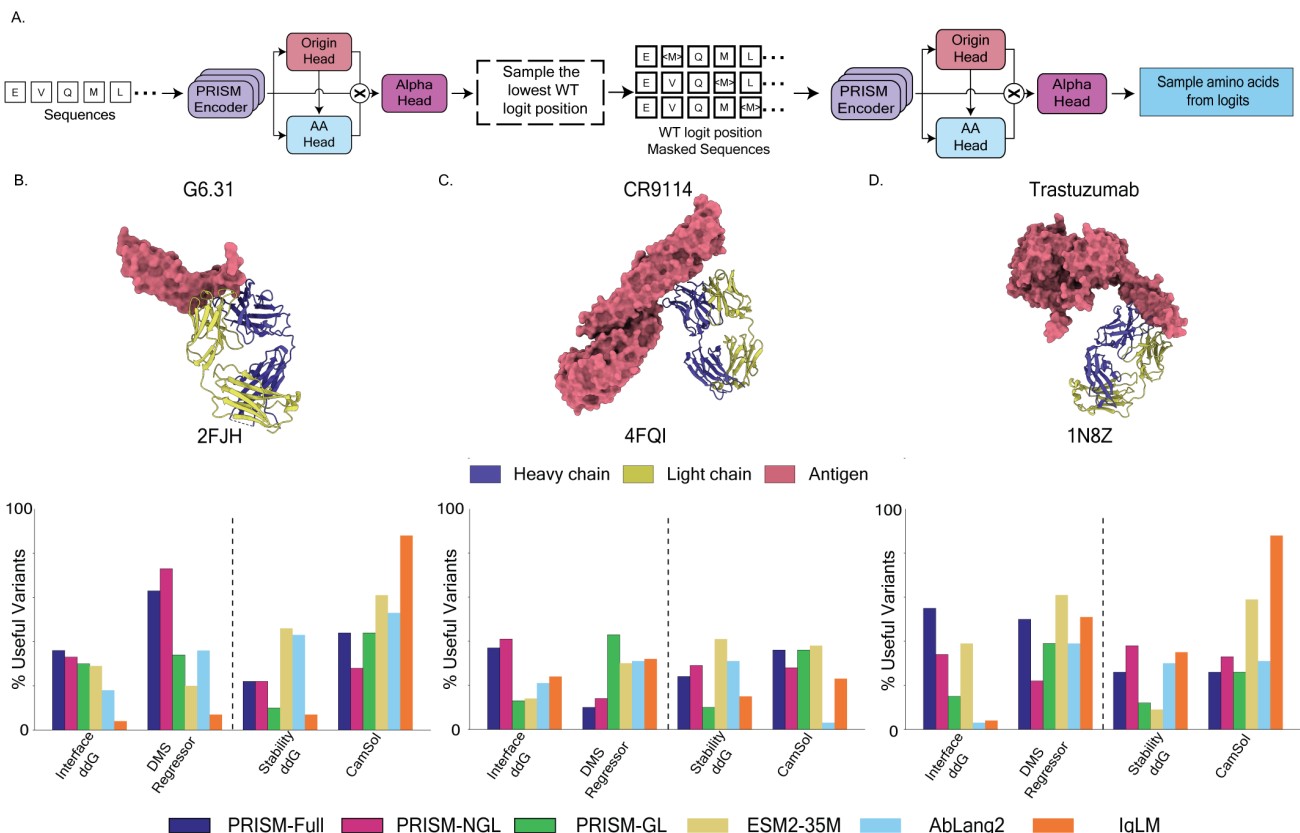

*Figure 3.* **Disentangled-logit generation enables property-specific control. (A)** Protocol: for each WT antibody we take the 20 positions with the lowest WT logit under PRISM, draw 10 per variant, mask, and re-sample under one of three channels – PRISM-Full (alpha-gated), PRISM-NGL (NGL only), PRISM-GL (GL only). **(B–D)** Fraction of generated 10-mutation variants improving over WT on each axis for G6.31 (2FJH), CR9114 (4FQI), and Trastuzumab (1N8Z). PRISM-NGL leads on Rosetta interface $\Delta\Delta G$ (binding); PRISM-GL leads on Rosetta stability and CamSol (solubility); PRISM-Full compares favorably against all generative baselines (ESM2-35M, AbLang2, IgLM) on the aggregate. The DMS regressor axis is a DMS-trained ridge regression binding affinity predictors; G6.31 alone has site-saturated DMS coverage (Appendix E.4).

## 6. Zero-Shot Prediction of Binding Affinity and Developability

All evaluation datasets are introduced in Section 3.4. We score binding affinity via NGL probability and developability via GL probability (Jain et al., 2017), sign-adjusting all Spearman correlations ($\rho$) so positive values denote favorable outcomes.

### 6.1. Binding Affinity Prediction

PRISM is uniquely positively correlated with experimental affinity on *every* landscape tested (Fig. 4B–D). On the combinatorial **CR9114** ($\rho = +0.391$, $4.6\times$ enrichment at top-10%), every baseline correlates *negatively* ($\rho \in [-0.428, -0.326]$) and yields anti-enrichment ($0.1\times$–$0.4\times$), making PRISM the only method delivering practical screening guidance in this regime. On **Trastuzumab**, PRISM ($\rho = +0.327$, $2.6\times$) edges out the strongest specialized ALM (AntiBERTy: $+0.297$, $2.5\times$). On the strongly affinity-matured

**G6.31** ($K_d \approx 0.4$ nM), PRISM is the only model with a positive Spearman ($+0.158$), while every baseline yields negative ranks (Sapiens $-0.018$ down to ESM2-35M $-0.202$); G6.31's narrower enrichment dynamic range ($2.3\times$, comparable to IgLM's $2.6\times$ despite IgLM's $\rho = -0.065$) is the expected consequence of scoring near a local affinity optimum, and PRISM's advantage is clearest in rank correlation (Appendix H.3). On **FLAb2** (Fig. 4E–G), PRISM has the strongest Spearman on $41.5\%$ of antibodies (next-best ESM2-650M: $19.5\%$), the lowest mean rank ($3.59$ vs. all baselines $> 4.00$), and the highest median per-antibody Spearman ($0.091$).

### 6.2. Developability Assessment

PRISM delivers a balanced developability profile (Fig. 5). On the Arsiwala/Marks 6-trait panel (Appendix Fig. S6A–F), PRISM ranks **first** on Self-Interaction, Hydrophobicity, and Thermal Stability, with positive correlations across all six traits and 2nd/4th on Immunogenicity/Expression.

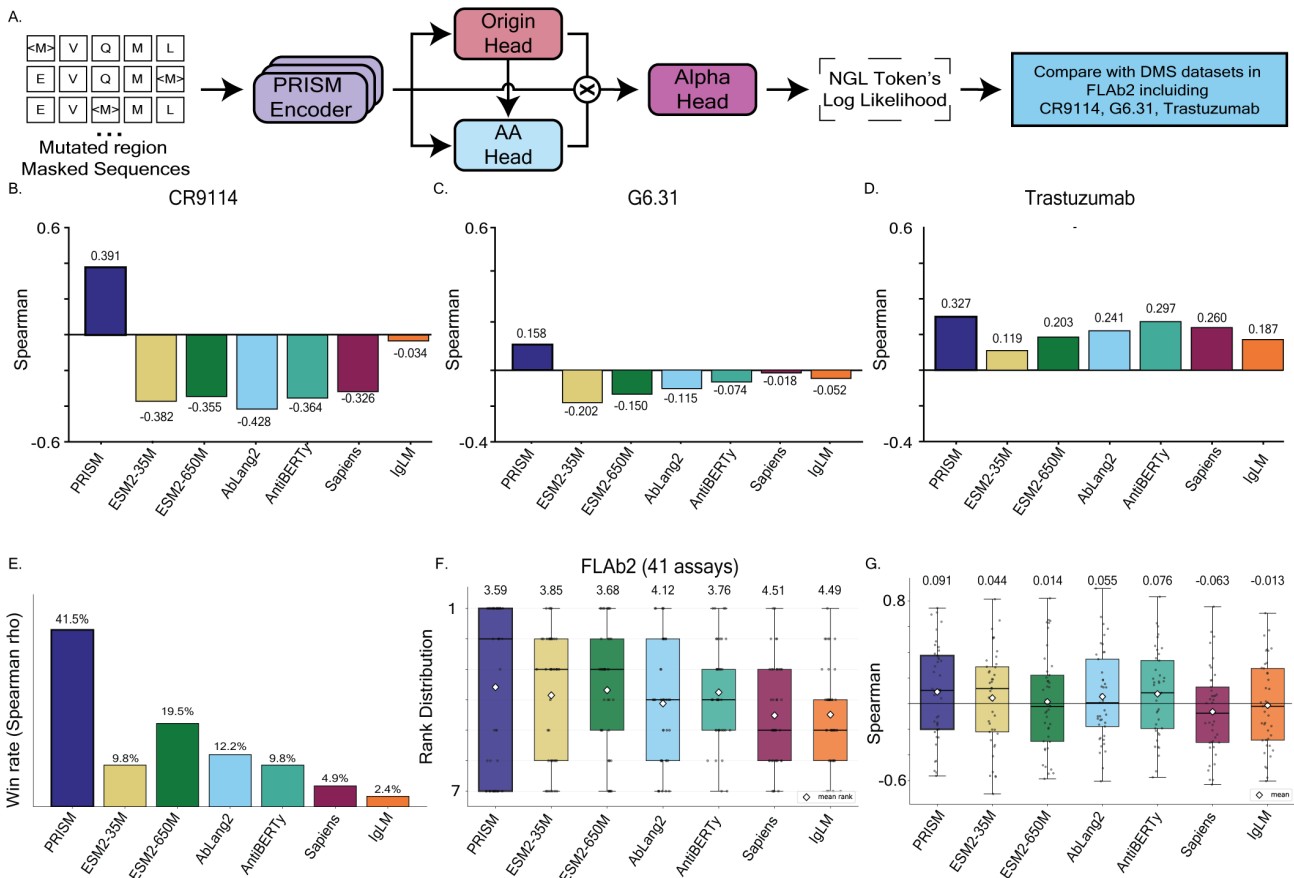

*Figure 4.* **Zero-shot prediction of binding affinity. (A)** Mask-one-out evaluation: PRISM's NGL-constrained log-likelihood vs. experimental affinity on three DMS sets and the 41-antibody **FLAb2** panel. **(B–D)** Spearman $\rho$ on CR9114 ($N = 65{,}093$), G6.31 ($N = 4{,}274$), Trastuzumab ($N = 36{,}496$): PRISM is the *only* method positive on all three (CR9114 $+0.391$, G6.31 $+0.158$, Trastuzumab $+0.327$); every baseline is negative on CR9114 ($[-0.428, -0.326]$) and G6.31 ($[-0.018, -0.202]$), indicating systematic germline bias. **(E–G) FLAb2 (41 antibodies):** PRISM has the top per-assay win rate ($41.5\%$ vs. ESM2-650M $19.5\%$), the lowest mean rank ($3.59$ vs. all baselines $>4.00$), and the highest median per-antibody Spearman ($0.091$ vs. AntiBERTy $0.078$, ESM2-650M $0.026$). See Sec. 6.1, Appendix H.3.

On the broader **FLAb2** developability panel (Appendix Fig. S6G–K), PRISM leads on DSC (thermal stability) and AC-SINS (aggregation) and stays competitive on HEK, PSR, and ADA – a state-of-the-art stability profile without compromising reactivity or immunogenicity. Pearson analysis (Appendix Fig. S7) reproduces the same ordering.

### 6.3. Validation of Dual-Vocabulary Strategy: The Necessity of Disentanglement

To test the premise that NGL mutations drive function while GL identity scaffolds developability, we decomposed PRISM's output into **GL-only**, **NGL-only**, and **Marginal** (sum) scores and compared each against AbLang2 (Fig. 5B). On *binding affinity*, NGL (Magenta) consistently outperforms GL (Green) across CR9114, G6.31, and Trastuzumab, with the gap widest on CR9114 where AbLang2 (Light Blue) collapses to a strongly negative correlation while both PRISM scores remain positive – the dual-vocabulary de-

composition cleanly recovers the SHM-driven affinity signal where AbLang2 cannot. On developability, GL and NGL are largely comparable across Hydrophobicity, Self-Interaction, Polyreactivity, and Thermal Stability, but on the two traits most directly tied to germline identity – **Expression** (yield is determined partly by germline V-gene) and **Immunogenicity** (framework similarity to germline shapes ADA risk) – the GL score carries slightly more information than NGL.

The **Marginal** score (Navy) – mimicking standard pLMs that do not separate origin – frequently underperforms the better-aligned specialized score (e.g., on Trastuzumab and CR9114), consistent with a "signal cancellation" effect when complementary axes are entangled in a single distribution. Together, (i) the clear GL/NGL split on binding, (ii) the biologically aligned GL advantage on Expression and Immunogenicity, and (iii) the cost of marginalizing the two channels establish PRISM as a controllable generator

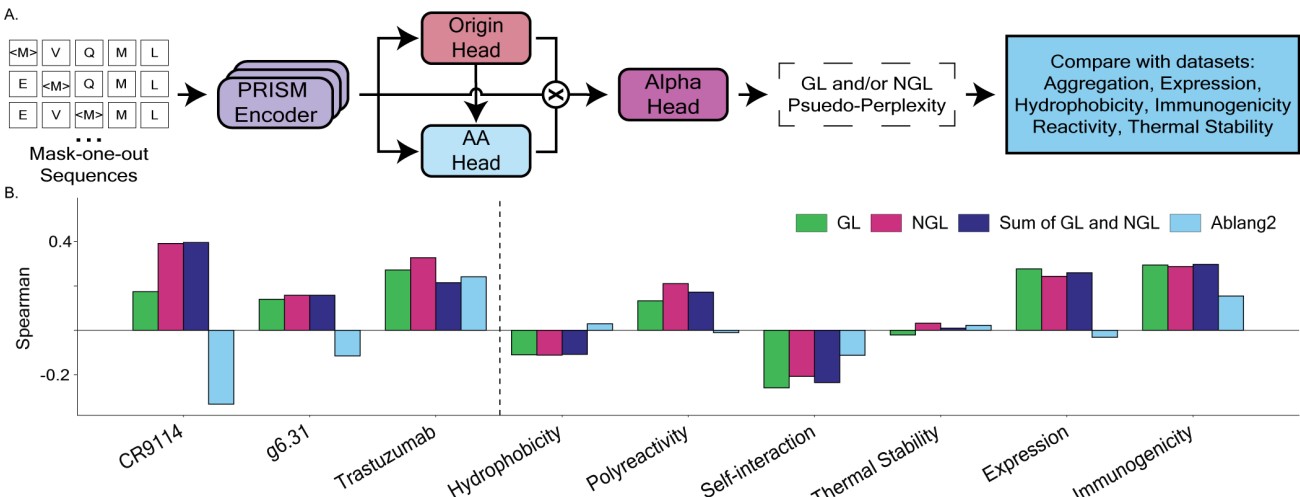

*Figure 5.* **Disentanglement of Function and Developability. (A)** Mask-one-out sequences are scored by PRISM's GL-, NGL-, and Sum-of-(GL,NGL) log-likelihoods via the Origin → AA → Alpha Head pipeline, then compared against binding-affinity DMS sets and biophysical assays. **(B)** Zero-shot Spearman of GL (green), NGL (magenta), Marginal/Sum (navy), and AbLang2 (light blue) across three binding-affinity DMS sets (left of dashed line) and six developability traits (right). NGL consistently outperforms GL on binding, with the gap widest on CR9114 where AbLang2 collapses to strongly negative; GL carries marginally more signal on Expression and Immunogenicity; the Marginal score frequently underperforms the better-aligned individual channel (Sec. 6.3).

that can independently modulate developability and binding – a capability unattainable by entangled models.

## 7. Ablation Studies

We ablate three orthogonal axes – architecture (multi-head vs. simple head), training (pretrained vs. scratch), and initialization (random vs. ESM2) – via four variants (Fig. 6A): **Ablation 1** drops unpaired pretraining while keeping the multi-head; **Ablation 2** keeps pretraining but uses a simple LM head; **Ablation 3** drops both; and **PRISM-less** fine-tunes ESM2-35M on antibody data with a simple head over the standard (upper-case) vocabulary – the strongest "naive transfer" baseline. All variants share the identical training pipeline — focal loss with per-token NGL reweighting $\alpha_{NGL} = 3.0$, region-aware masking, and the hyperparameters of Sec. C — and Ablations 1–3 additionally share the extended 53-token vocabulary; variants differ *only* in the axes named above, isolating the architectural and pretraining contributions from the loss-side reweighting mechanism.

### 7.1. Representational Disentanglement and Sequence Modeling

Architectural factorization and representation quality are separable axes. **PRISM Full** attains near-perfect GL/NGL separability (PR-AUC 0.980, F1 0.896, ARI 0.162), while **Ablation 2** collapses to near-random separability (0.377, ARI 0.009) despite identical pretraining data, confirming that the multi-head is the primary driver of disentanglement; Ablation 1 retains partial separability (0.517/0.079)

and Ablation 3 falls to the floor (0.217). **PRISM-less** attains strong probe separability (PR-AUC 0.921, F1 0.762) but near-zero ARI (0.006), indicating ESM2 already encodes a recoverable GL/NGL direction without geometric organization in the embedding space. On generative PPL (Fig. 6D), PRISM Full leads on both chains followed closely by PRISM-less; Ablations 2 and 3 collapse (∼50 and ∼25 heavy-chain PPL) because the simple head cannot map entangled features to the extended GL/NGL vocabulary, while PRISM-less sidesteps this on the standard vocabulary at the cost of forfeiting origin-aware generation (Appendix I).

### 7.2. Zero-Shot Prediction Impact

The functional consequences of factorization are starkest on zero-shot affinity. On CR9114, PRISM Full reaches $\rho = +0.393$ while every ablation fails (Ablation 1: $-0.205$, Ablation 2: $-0.279$), and **PRISM-less** produces the strongest *negative* correlation of all variants ($-0.409$); the same ordering holds on G6.31 (PRISM-less $-0.115$) and on Trastuzumab (PRISM-less 0.176 vs. PRISM Full's 0.327). This directly answers whether ESM2 scale plus antibody fine-tuning can substitute for explicit factorization: it cannot. Without an Origin Head to gate predictions, PRISM-less inherits ESM2's representational competence but defaults to germline bias and systematically penalizes the SHM substitutions that drive binding – representation quality and architectural routing are complementary, not substitutable.

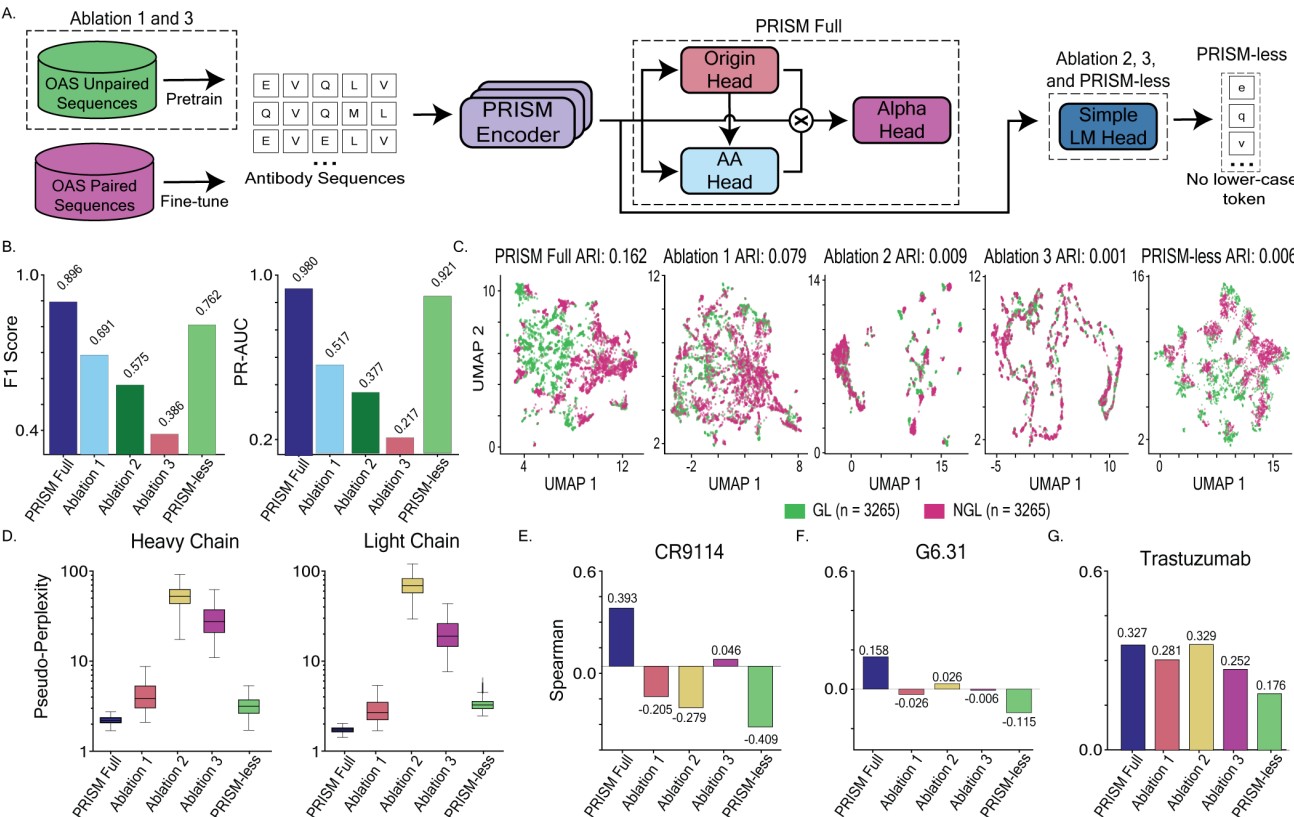

*Figure 6.* **Ablation analysis. (A)** Comparison of PRISM Full against four ablated variants: Ablation 1 (multi-head, no unpaired pretraining), Ablation 2 (simple LM head, pretrained), Ablation 3 (simple LM head, no pretraining), and PRISM-less (ESM2 initialization with antibody fine-tuning, simple head, upper-case tokens only). **(B, C)** Factorization is essential for GL/NGL separability in the learned representation. **(D)** Impact on generative perplexity across heavy and light chains. **(E-G)** Effect on zero-shot affinity correlations across CR9114, G6.31, and Trastuzumab. PRISM-less attains competitive disentanglement and perplexity but collapses on affinity prediction, isolating architectural factorization—rather than representation quality alone—as the driver of functional performance.

## 8. Discussion

PRISM mitigates "germline bias" by factoring antibody generation into a joint distribution of evolutionary origin and physicochemical identity, yielding geometrically separable latent representations, controllable generation that independently modulates binding (NGL) and developability (GL), and zero-shot affinity prediction across landscapes where standard ALMs collapse to germline-conservative scoring. Our ablations isolate architectural factorization – not data scale or ESM2 initialization – as the driver of these gains, and we view PRISM as an instance of a broader modeling principle: when sequence generation is governed by a strong hierarchical decomposition (germline scaffold vs. functional diversification, conserved vs. catalytic residues, scaffold-preserving vs. function-modifying mutations), explicitly factorizing the generative process is more effective than relying on a single entangled distribution – particularly in regimes with highly imbalanced signals where rare but functionally important deviations are otherwise suppressed.

**Limitations and Future Direction.** Zero-shot developabil-

ity remains challenging. We plan to leverage the GL/NGL decomposition (treating developability and binding as independent control variables) to resolve the developability-affinity trade-off via reinforcement learning (Blalock et al., 2025).

## Impact Statement

This work accelerates therapeutic antibody design. While we acknowledge dual-use risks in generative biology, our focus is strictly clinical and we emphasize the need for ethical guardrails in protein engineering.

## Code and Data Availability.

Source code, inference scripts, the inference/evaluation dataset, and raw prediction scores for PRISM and all baselines are available at https://github.com/Romerolab/PRISM.

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

## A. Data Curation and Statistics

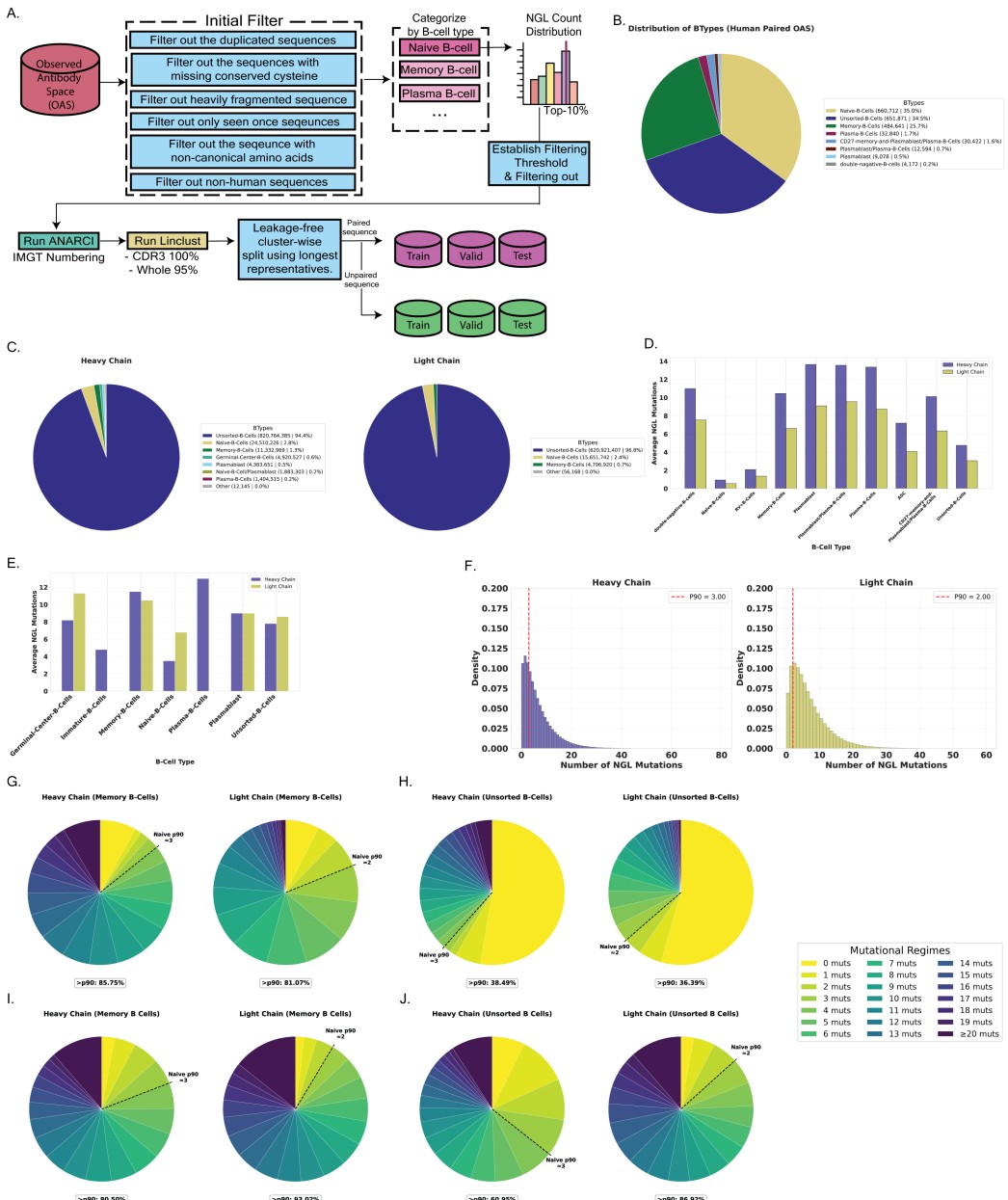

*Figure S1.* **Data curation pipeline and mutational statistics of the Observed Antibody Space (OAS). (A)** The multi-stage preprocessing pipeline filters raw sequences for quality, canonical residues, and species (Human) before establishing mutation thresholds. Sequences are split into Train/Validation/Test sets using leakage-free clustering (Linclust). **(B, D, F, G, H) Paired Dataset Statistics: (B)** Distribution of B-cell types, dominated by Naive (35.0%) and Unsorted (34.5%) cells. **(D)** Average non-germline (NGL) mutation counts, showing distinct mutational loads across types. **(F)** Density histograms of NGL mutations in Naive B-cells used to calibrate the noise floor ($\tau_{HC} = 3$, $\tau_{LC} = 2$). **(G-H)** Mutational regimes of Memory and Unsorted B-cells in the paired set, demonstrating high data retention ($> 80\%$) after p90 filtering. **(C, E, I, J) Unpaired Dataset Statistics: (C)** Raw distribution of B-cell types in the heavy and light chain unpaired datasets. **(E)** Average NGL mutation counts for unpaired heavy chains. **(I-J)** Mutational regimes for unpaired Memory and Unsorted B-cells, validating that the filtering strategy remains robust across the larger unpaired corpus.

### A.1. OAS Database Preprocessing

We train PRISM on antibody sequences sourced from the Observed Antibody Space (OAS) database (Kovaltsuk et al., 2018). As illustrated in **Figure S1A**, our preprocessing pipeline consists of sequential filtering stages designed to ensure sequence

quality, biological validity, and appropriate mutation levels for generative modeling.

**Data Source.** We utilize two distinct subsets of the OAS database:

- **Paired Dataset (Fig. S1 B, D, F, G, H):** Used for fine-tuning. Starting from a quality-controlled pool of 1,888,609 sequences, we apply strict mutation thresholds and clustering to yield a final high-quality corpus of 619,675 paired sequences.

- **Unpaired Dataset (Fig. S1 C, E, I, J):** Used for pretraining. Following the filtering pipeline, the final dataset comprises 57,968,879 heavy chain and 8,466,697 light chain sequences.

**Filtering Pipeline.** We apply a rigorous filtering process, which significantly refines the dataset size at each stage. The sequence counts for both datasets at each step are as follows:

1. **Quality Control (QC):** We initially filter for human-derived sequences, valid conserved cysteine residues (via ANARCI), complete FR1/FR4 regions, and standard amino acids.

   - *Paired Count:* 1,888,609.
   - *Unpaired Count:* 869,211,721 (Heavy) / 251,426,237 (Light).

2. **Naive B-cell Thresholding (p90):** To remove noisy artifacts, we exclude Naive sequences exceeding the 90th percentile mutation count ($\tau_{HC} = 3, \tau_{LC} = 2$). This step removes high-mutation artifacts often misclassified as Naive.

   - *Paired Count:* 763,989.
   - *Unpaired Count:* 526,427,480 (Heavy) / 31,171,246 (Light).

3. **Clustering & Deduplication:** We employ `seqkit rmdup` for fast exact deduplication followed by Linclust (100% CDR3, 95% whole-sequence identity) to remove redundancy and construct the final training corpus.

   - *Final Paired Count:* 619,675.
   - *Final Unpaired Count:* 57,968,879 (Heavy) / 8,466,697 (Light).

## A.2. Mutation-Based Filtering to Mitigate Germline Bias

Because naive B-cells are expected to exhibit minimal somatic hypermutation (SHM), observed mutations in this subset largely reflect sequencing artifacts or baseline noise rather than true adaptive variation. We therefore use naive repertoires to estimate a practical floor for identifying sequences with at least some affinity maturation.

Specifically, we compute the empirical distribution of mutation counts in paired naive B-cell sequences from OAS and define a threshold based on the 90th percentile (p90). As shown in **Figure S1F**, 90% of naive sequences contain $\leq 3$ mutations. We therefore set chain-specific thresholds:

$$\tau_{HC} = 3 \quad \text{(Heavy Chain)} \tag{9}$$
$$\tau_{LC} = 2 \quad \text{(Light Chain)} \tag{10}$$

and exclude all sequences with mutation counts at or below this threshold ($N_{NGL} \leq \tau$), regardless of their annotated B-cell subtype.

Importantly, this procedure is not intended to eliminate sequencing errors. Instead, it removes a large volume of near-germline sequences that are statistically indistinguishable from germline templates plus baseline noise. Including such sequences disproportionately reinforces germline signal and exacerbates germline bias during training. By filtering this low-information regime, we shift the training distribution toward sequences containing meaningful non-germline variation, improving the model's ability to learn rare but functionally relevant mutations.

As shown in **Figure S1G–J**, this filtering is highly selective: it removes the majority of the naive noise floor while preserving most antigen-experienced sequences. For example, over 80% of memory B-cells are retained (**Fig. S1G**), indicating that the procedure enriches for biologically relevant diversity without discarding the signal associated with affinity maturation.

## A.3. Leakage-Free Train/Test Split

Standard random splitting leads to data leakage due to clonal expansion. We employ a cluster-based splitting strategy using Linclust (Steinegger & Söding, 2018). Sequences are grouped by 100% CDR3 identity and 95% whole-sequence identity. Entire clusters are assigned to a single split, ensuring no validation sequence is clonally related to the training set.

We maintain a fixed external test set of approximately 22,600 sequences. The remaining clusters are split into training and validation sets with a 9:1 ratio, as detailed in Table S1.

*Table S1.* Final paired dataset split statistics (Filtered NGL).

| Split | Sequence Count | Percentage |
|---|---|---|
| Train | 537,374 | 86.7% |
| Validation | 59,708 | 9.6% |
| Test | 22,593 | 3.7% |
| **Total** | **619,675** | **100.0%** |

## A.4. Dataset Statistics

**Sequence Format.** Input sequences are tokenized as `<CLS>{Heavy}<CLS><CLS>{Light}<EOS>` for paired antibody sequences and `<CLS>{Sequence}<EOS>` for unpaired antibody seuqneces with a maximum length of **320** tokens.

**Mutational Load.** The average mutation load varies significantly by cell type and dataset. In the paired dataset (**Fig. S1D**), Plasma B-cells exhibit high mutation rates (∼14 mutations). Similarly, the unpaired dataset (**Fig. S1E**) confirms that Memory B-cells maintain high mutational diversity, providing a rich signal for learning affinity maturation trajectories.

**Region Annotation.** We use IMGT numbering (Lefranc et al., 2003) to demarcate framework and CDR regions (Table S2).

*Table S2.* IMGT region definitions used for token masking and analysis.

| Region | IMGT Positions | ID |
|---|---|---|
| FR1 | 1–26 | 0 |
| CDR1 | 27–38 | 1 |
| FR2 | 39–55 | 2 |
| CDR2 | 56–65 | 3 |
| FR3 | 66–104 | 4 |
| CDR3 | 105–117 | 5 |
| FR4 | 118–128 | 6 |

**Germline Gene Vocabulary.** We construct a vocabulary of 365 unique V and J genes following IMGT nomenclature (e.g., `IGHV1-2*01`), supplemented with special tokens for padding and unknown genes.

# B. Model Architecture Details

## B.1. Backbone Encoder: Architecture and Initialization

PRISM adopts the structural configuration of the ESM-2 (Lin et al., 2023) transformer encoder (specifically the 35M parameter variant, `esm2_t12_35M_UR50D`). However, we introduce critical architectural modifications and training strategies optimized for antibody sequence modeling.

**Training from Scratch.** Strictly unlike standard ESM-2 applications that rely on transfer learning, PRISM does **not** use pre-trained weights. The entire model is **initialized randomly and trained from scratch**. This prevents the induction of bias from the general protein universe, allowing the model to specialize exclusively on antibody affinity maturation landscapes.

**SwiGLU Activation.** We replace the standard GELU activation in the backbone feed-forward networks (FFN) with **SwiGLU**

(Swish-Gated Linear Unit) (Shazeer, 2020). Formally, the FFN transformation is defined as:

$$\text{FFN}_{\text{SwiGLU}}(\mathbf{x}) = (\text{Swish}(\mathbf{x}\mathbf{W}_g) \odot (\mathbf{x}\mathbf{W}_{\text{in}}))\mathbf{W}_{\text{out}} \tag{11}$$

where $\odot$ denotes the element-wise Hadamard product. Unlike standard activation functions, the multiplicative gating mechanism allows the network to selectively modulate information flow via bilinear interactions. This architecture has been empirically demonstrated to offer superior compute-efficient scaling properties and faster convergence compared to GELU.

**Rotary Positional Embeddings (RoPE).** We utilize Rotary Positional Embeddings (Su et al., 2024) to capture relative positional information, preserving the inductive bias of the original ESM-2 architecture.

### B.2. Vocabulary and Special Tokens

PRISM inherits the standard ESM-2 tokenizer vocabulary (Lin et al., 2023), comprising the 20 canonical amino acids and 13 special tokens (Table S3). These special tokens include control symbols for sequence boundaries (`<cls>`, `<eos>`), padding (`<pad>`), and the masked language modeling objective (`<mask>`); non-canonical or ambiguous residue placeholders (X, B, U, Z, O); and alignment gap symbols (., −). Notably, we **repurpose the `<cls>` token as a chain-separator** in paired inputs (i.e., `<CLS>{Heavy}<CLS><CLS>{Light}<EOS>`), leveraging its contextual embedding capacity to mediate cross-chain information flow between the heavy and light variable regions.

*Table S3.* The 13 special tokens inherited from the ESM-2 tokenizer vocabulary. Token IDs follow the canonical ESM-2 ordering; indices 4–23 (omitted) correspond to the 20 standard amino acids.

| ID | Token | Category | Description |
|----|-------|----------|-------------|
| 0 | `<cls>` | Control | Classification / sequence-start token; repurposed as chain separator for paired inputs |
| 1 | `<pad>` | Control | Padding token for batch-length alignment |
| 2 | `<eos>` | Control | End-of-sequence marker |
| 3 | `<unk>` | Control | Unknown-token placeholder for out-of-vocabulary characters |
| 24 | X | Ambiguous | Fully ambiguous residue (any amino acid) |
| 25 | B | Ambiguous | Aspartate or Asparagine (Asx) |
| 26 | U | Non-standard | Selenocysteine (Sec, 21st amino acid) |
| 27 | Z | Ambiguous | Glutamate or Glutamine (Glx) |
| 28 | O | Non-standard | Pyrrolysine (Pyl, 22nd amino acid) |
| 29 | . | Gap | Alignment gap (insertion-preserving notation) |
| 30 | − | Gap | Standard alignment gap |
| 31 | `<null_1>` | Reserved | Reserved placeholder token (unused in training) |
| 32 | `<mask>` | Control | Masking token used for the masked language modeling objective |

### B.3. Multimodal Input Embeddings

To condition the model on biological context, we utilize a composite embedding strategy. The gene and region embeddings are projected to match the hidden dimension ($d = 480$) before being combined.

The input representation $\mathbf{x}_i$ for token $i$ is calculated as:

$$\mathbf{x}_i = \text{Emb}_{\text{token}}(t_i) + \text{Proj}_{\text{reg}}(\text{Emb}_{\text{reg}}(r_i)) + \text{Proj}_{\text{gene}}([\text{Emb}_{\text{V}}(g_v); \text{Emb}_{\text{J}}(g_j)]) \tag{12}$$

where:

- $\text{Emb}_{\text{token}} \in \mathbb{R}^{480}$ is the token embedding.

- $\text{Emb}_{\text{reg}} \in \mathbb{R}^{32}$ is the region embedding, projected to $\mathbb{R}^{480}$ via a linear layer.

- $\text{Emb}_{\text{V}}, \text{Emb}_{\text{J}} \in \mathbb{R}^{64}$ are V and J gene embeddings. They are concatenated ($d = 128$) and projected to $\mathbb{R}^{480}$.

- The gene embedding vector is broadcasted across the sequence length $L$.

### B.4. Multi-Head Architecture

The prediction heads operate on contextualized embeddings $\mathbf{h}_i \in \mathbb{R}^{480}$ from the transformer encoder. Note that while the backbone uses SwiGLU, the prediction heads utilize standard GELU activations.

**Origin Head.** A linear layer predicting the probability that a residue is a mutation (non-germline):

$$\text{Origin Head: Linear}(480 \to 1) \tag{13}$$

**AA Identity Head.** A two-layer MLP predicting amino acid identity. Unlike the backbone, this head uses standard **GELU** activation:

$$\text{AA Head: Linear}(480 \to 480) \to \text{GELU} \to \text{LayerNorm} \to \text{Linear}(480 \to V) \tag{14}$$

where $V = |\mathcal{V}_{\text{base}}|$ denotes the base vocabulary size, comprising standard amino acids and special control tokens (e.g., `<pad>`, `<eos>`), prior to evolutionary factorization.

**Alpha Head (Gating Mechanism).** We introduce a learnable scalar gate $\alpha_i \in [0, 1]$ for each position, predicted by a lightweight projection:

$$\alpha_i = \sigma(\text{Linear}(480 \to 1)(\mathbf{h}_i)) \tag{15}$$

The $\alpha$ value acts as a confidence gate: $\alpha \approx 1$ indicates a high probability of the position being Germline, while $\alpha \approx 0$ suggests a Mutation. This gate is used to modulate the contribution of logits during training.

### B.5. Parameter Count

Table S4 details the specific parameter breakdown, including the projection layers for the multimodal embeddings.

*Table S4.* PRISM parameter breakdown.

| Component | Configuration | Parameters |
|---|---|---|
| **Backbone Encoder** | 12 Layers, 480 dim, **SwiGLU** | $\sim$35,000,000 |
| **Heads** | | |
| AA Identity Head | Dense(480)+GELU+LN+Dec(53) | $\sim$507,000 |
| Origin Head | Linear(480, 1) | 481 |
| Alpha Head | Linear(480, 1) + Sigmoid | 481 |
| Origin Projection | Linear(1, 480) | 960 |
| **Embeddings** | | |
| Gene Embeddings | Emb(367, 64) $\times$ 2 + Proj(128, 480) | $\sim$108,000 |
| Region Embeddings | Emb(8, 32) + Proj(32, 480) | $\sim$15,600 |
| **Total** | | $\sim$**35.63M** |

## C. Training Details

### C.1. Hyperparameters and Optimization

We train PRISM using the AdamW optimizer (Loshchilov & Hutter, 2017) with a cosine learning rate decay schedule (Loshchilov & Hutter, 2016). Gene conditioning is applied throughout both pretraining and fine-tuning stages. We utilize Gradient Accumulation to achieve a large effective batch size of 8,192, which is critical for stabilizing the training of generative protein models.

Table S5 details the specific hyperparameters based on our v34.1b configuration.

### C.2. Objective Function and Loss Configuration

The optimization objective is a weighted sum of three distinct loss components. We employ a Focal Loss (Lin et al., 2017) with specific re-weighting to address the extreme class imbalance between conserved germline residues and rare non-germline mutations.

*Table S5.* Training hyperparameters and optimization configuration.

| Configuration | Pretraining (Stage 1) | Fine-tuning (Stage 2) |
|---|---|---|
| *Optimization* | | |
| Optimizer | AdamW | AdamW |
| Learning rate | $4 \times 10^{-4}$ | $1 \times 10^{-4}$ |
| Betas $(\beta_1, \beta_2)$ | (0.9, 0.999) | (0.9, 0.999) |
| Weight decay | 0.01 | 0.01 |
| Gradient clipping | 1.0 | 1.0 |
| Warmup steps | 1,000 | 500 |
| Max steps | 15,000 | 12,000 |
| *Batching Strategy* | | |
| Per-device Batch Size | 256 | 256 |
| Gradient Accumulation | 16 | 16 |
| Number of GPUs | 2 | 2 |
| **Effective Batch Size** | **8,192** | **8,192** |
| Precision | BFloat16 | BFloat16 |
| Seed | 42 | 42 |

**Total Loss Formulation.** The total loss $\mathcal{L}_{\text{total}}$ is computed as:

$$\mathcal{L}_{\text{total}} = \lambda_{\text{final}}\mathcal{L}_{\text{final}} + \lambda_{\text{AA}}\mathcal{L}_{\text{AA}} + \lambda_{\text{origin}}\mathcal{L}_{\text{origin}} \tag{16}$$

Using the weights from our configuration: $\lambda_{\text{final}} = 2.0$, $\lambda_{\text{AA}} = 1.0$, and $\lambda_{\text{origin}} = 1.5$.

**Per-Token NGL Reweighting.** We do not simply scale the final loss, rather we apply a per-token weight tensor $w_t$ inside the focal loss computation. For strictly non-germline (NGL) positions, we assign a higher weight $\alpha_{\text{NGL}} = 3.0$, while germline positions retain a weight of 1.0.

$$w_t = \begin{cases} 3.0 & \text{if token } t \text{ is NGL} \\ 1.0 & \text{otherwise} \end{cases} \tag{17}$$

This ensures that gradients from rare mutation events are amplified proportionally at the token level, preventing them from being overwhelmed by the majority class.

*Table S6.* Loss configuration weights (v34.1b).

| Type | Parameter | Value |
|---|---|---|
| **Scalar Weights** | Final Head Weight ($\lambda_{\text{final}}$) | 2.0 |
| ($\lambda$) | Origin Head Weight ($\lambda_{\text{origin}}$) | 1.5 |
| | AA Head Weight ($\lambda_{\text{AA}}$) | 1.0 |
| **Token Weights** | NGL Position Alpha ($\alpha_{\text{NGL}}$) | **3.0** |
| (Internal Tensor) | GL Position Weight | 1.0 |

## C.3. Region-Aware Masking Strategy

We implement a Region-Aware Masking strategy that dynamically overrides masking probabilities based on biological regions. Unlike simple multiplication, we strictly assign specific masking rates to different regions to force the model to learn structural dependencies.

- **CDR Regions (0.50):** We apply a high masking rate of 50% to Complementarity-Determining Regions (CDRs) to challenge the model in reconstructing hypervariable loops.

- **Framework Regions (0.30):** Conserved Framework Regions (FRs) are masked at 30%.

- **Background (0.15):** Any remaining tokens follow the standard BERT (Devlin et al., 2019) masking rate of 15%.

### C.4. Computational Resources

Training was performed on NVIDIA L40S (48GB) GPUs. We utilized distributed data parallelism (DDP) combined with gradient accumulation steps (16 steps) to fit the large effective batch size of 8,192 within VRAM constraints.

*Table S7.* Computational infrastructure and training duration.

| Resource | Pretraining | Fine-tuning |
|---|---|---|
| GPU Model | **NVIDIA L40S (48GB)** | **NVIDIA L40S (48GB)** |
| Parallelism Strategy | DDP + Grad Accumulation | DDP + Grad Accumulation |
| Training Duration | $\sim$48 hours (2 days) | $\sim$12 hours |

## D. Evaluation Protocols and Baselines

### D.1. Baseline Models

To benchmark PRISM's performance, we compare against state-of-the-art protein and antibody language models. Table S8 summarizes their specifications. All baselines are evaluated using standard masked language modeling protocols: masking each position sequentially and computing the log probability of the true amino acid.

*Table S8.* Baseline model specifications.

| Model | Parameters | Training Data | Reference |
|---|---|---|---|
| ESM2-35M | 35M | UniRef50 | Lin et al. (2023) |
| ESM2-650M | 650M | UniRef50 | Lin et al. (2023) |
| AbLang2 | $\sim$45M | OAS | Olsen et al. (2022) |
| AntiBERTy | $\sim$26M | OAS | Ruffolo et al. (2021) |
| Sapiens | $\sim$569K | OAS | Prihoda et al. (2022) |

### D.2. Linear Probing Protocol

To assess the representational disentanglement of non-germline residues versus germline residues, we train linear probes on frozen embeddings extracted from the final hidden layer.

- **Input:** Contextualized embeddings $\mathbf{h}_i \in \mathbb{R}^{480}$ for each token position.

- **Task:** Binary classification (Germline vs. Non-Germline).

- **Configuration:** Single linear layer trained with AdamW ($lr = 10^{-3}$) and class-weighted cross-entropy loss.

- **Metric:** Area Under the Precision-Recall Curve (PR-AUC) on the test set.

### D.3. UMAP Projection Analysis

To visualize the latent space organization, we employ Uniform Manifold Approximation and Projection (UMAP) (McInnes et al., 2018).

- **Input Features:** Per-residue embeddings ($\mathbb{R}^{480}$) from the last transformer layer.

- **Preprocessing (Mean Centering):** Crucially, to remove the dominant signal of amino acid identity and reveal evolutionary structure, we subtract the mean embedding of each amino acid type before projection:

$$\mathbf{h}'_i = \mathbf{h}_i - \mu_{\text{AA}(x_i)} \tag{18}$$

- **Parameters:** We apply UMAP directly to the centered embeddings (no PCA) using `umap-learn` with `n_neighbors=100`, `min_dist=0.1`, and `metric='cosine'`.

## D.4. Pseudo-Perplexity (PPL) Calculation

Since baseline models utilize varying vocabulary sizes and special tokens, a direct comparison of raw perplexity is inequitable. To ensure a strictly fair comparison, we standardize the evaluation space to the 20 canonical amino acids for all models.

**1. Normalized PPL (Standardized Baseline Evaluation).** For all baseline models, we restrict the probability distribution to the 20 canonical amino acids. We extract the logits corresponding to these 20 residues and normalize them via softmax. This ensures that the evaluation focuses solely on the model's ability to predict the correct amino acid identity, disregarding vocabulary discrepancies:

$$\text{PPL}_{\text{norm}} = \exp\left(-\frac{1}{N}\sum_{i=1}^{N}\log P_{\text{20-class}}(x_i \mid \mathbf{x}_{\backslash i})\right) \tag{19}$$

**2. Marginalized PPL (PRISM Evaluation for Comparison).** To align PRISM with this standardized 20-amino acid benchmark, we project our extended vocabulary predictions into the canonical space. Since PRISM explicitly models evolutionary origin, the probability of a standard amino acid $x$ is derived by marginalizing (summing) over its germline (uppercase) and non-germline (lowercase) variants:

$$P_{\text{20-class}}(x) = P(x^{\text{upper}}) + P(x^{\text{lower}}) \tag{20}$$

We then compute the perplexity using these summed probabilities. This allows for a direct, "apples-to-apples" comparison with the Normalized PPL of baseline models.

**3. Exact PPL (PRISM Internal Performance).** This metric evaluates PRISM's full generative capability on the native 53-token vocabulary. Unlike the comparison metrics above, Exact PPL penalizes the model if it correctly predicts the amino acid identity but fails to identify the correct evolutionary origin (GL vs. NGL):

$$\text{PPL}_{\text{exact}} = \exp\left(-\frac{1}{N}\sum_{i=1}^{N}\log P_{\text{53-class}}(t_i \mid \mathbf{x}_{\backslash i})\right) \tag{21}$$

## D.5. Thera-SAbDab Benchmark

To assess generalizability to therapeutic antibodies, we utilize the Thera-SAbDab dataset (Raybould et al., 2020).

- **Dataset Curation:** We curated a dataset of 1,104 therapeutic antibody sequences (Heavy/Light pairs) from the Thera-SAbDab database (Feb 2025).

- **Filtering:** To ensure rigorous zero-shot evaluation, we removed any sequence sharing $> 95\%$ sequence identity with our training corpus.

- **Masking Protocol:** We employed a random masking strategy, selecting approximately 20% of residue positions in each sequence for evaluation. To simulate a reconstruction task, selected tokens were masked, and the model was tasked with recovering the original identity.

- **Metric & Analysis:** Recovery performance was evaluated using Top-1 Accuracy based on logit ranking. Crucially, we performed a stratified post-hoc analysis to inspect model behavior across different biological contexts. We report accuracy metrics separately for:
  - **Structural Regions:** Complementarity-Determining Regions (CDR) vs. Framework Regions (FR).
  - **Evolutionary Origin:** Germline (GL) vs. Non-Germline (NGL) residues.

## D.6. Zero-Shot Scoring

We leverage the model's conditional likelihood estimates to score variants for binding affinity and developability without any supervised fine-tuning.

**Binding Affinity.** We define the variant score $S_{\text{binding}}$ as the masked log-probability of the mutant residue. Uniquely, PRISM utilizes its disentangled vocabulary to explicitly score the probability of a non-germline mutation event. For a variant

sequence with mutations at indices $\mathcal{M}$, the score is computed as:

$$S_{\text{binding}} = \sum_{i \in \mathcal{M}} \log P(\text{mut\_lower}_i \mid \mathbf{x}_{\backslash i}) \tag{22}$$

By summing the log-probabilities of the **lowercase** tokens, we explicitly isolate the evolutionary drive towards affinity maturation. Baseline models, lacking this distinction, are scored using standard amino acid probabilities.

**Developability.** We approximate biophysical fitness using the model's perplexity when forced to the **Germline (Uppercase)** manifold. Lower perplexity indicates a sequence that appears more "natural" and structurally stable:

$$S_{\text{dev}} = -\text{PPL}(\mathbf{x}^{\text{upper}}) \tag{23}$$

### D.6.1. EVALUATION DATASETS

We evaluate zero-shot performance on three deep mutational scanning (DMS) datasets for binding affinity (Table S9) and two high-throughput datasets for developability.

*Table S9.* Benchmarks for zero-shot binding affinity prediction.

| Dataset | Target | Study Type | N (Seqs) | Mutations/Seq (Mean) | Fitness Metric | Reference |
|---|---|---|---|---|---|---|
| CR9114 | Influenza HA (H1) | Combinatorial ($2^{16}$) | 65,093 | 8.00 | $-\log(K_D)$ | Phillips et al. (2021) |
| G6.31 | VEGF | DMS (Site-saturation) | 4,274 | 1.00 | $-\log(K_D)$ | Koenig et al. (2017a) |
| Trastuzumab | HER2 | Combinatorial ($2^{10}$) | 36,496 | 7.49 | Log Enrichment | Mason et al. (2021) |

**Developability Benchmarks.** We utilize a high-throughput biophysical dataset comprising 246 clinical-stage antibodies characterized by Arsiwala et al. (2025). The dataset covers multiple properties including Expression Titer, Aggregation (SEC %Monomer), Hydrophobicity (HIC), Thermal Stability (Tm1), and Self-interaction (AC-SINS). Additionally, we evaluate immunogenicity using the Anti-Drug Antibody (ADA) dataset ($N = 206$) curated by Marks et al. (2021).

### D.7. Statistical Testing

We use the following statistical procedures to ensure robustness:

- **Paired comparisons:** Wilcoxon signed-rank test (Wilcoxon, 1945) was used to assess significance between model variants.

- **Correlation:** We report Spearman's $\rho$ for rank-order correlation and Pearson's $r$ for linear relationships.

- **Confidence intervals:** 95% confidence intervals were computed via bootstrap resampling (Efron, 1992) with 1,000 iterations.

- **Significance levels:** **** $p < 0.0001$, *** $p < 0.001$, ** $p < 0.01$, * $p < 0.05$.

## E. Generation Methodology

This section documents the generation experiment that supports Section 5: the position-selection and sampling protocol, the Rosetta and CamSol scoring pipelines, and the ridge regression binding-affinity predictors. The diagnostic figures of the generated variants, per-metric distributions and the spatial distribution of generated mutations, are deferred to Appendix G.

### E.1. Position Selection and Sampling

**Candidate position pool.** For each wild-type antibody, we run PRISM on the unmasked sequence and record, at every position, the logit assigned to the wild-type residue. We rank positions ascending by this logit and retain the 20 lowest-ranked positions as the candidate pool – these are the positions where PRISM judges the wild-type residue to be the least preferred under its learned distribution and therefore the positions most amenable to substitution. The pool is shared across all sampling runs of a given antibody.

**Per-variant subsampling.** Each generated variant independently draws 10 positions *without replacement* from the pool of 20. We do not fix the same 10 positions across variants: each variant samples a distinct subset, so the union of mutated positions over the 100 variants per antibody covers all 20 candidate positions while every individual variant carries the requested $n_{\mathrm{mut}} = 10$ mutations.

**Sampling channel.** At each chosen position, the residue is masked, the model is re-run on the partially masked sequence, and the new amino acid is sampled from the masked-LM distribution at that position. The three PRISM variants differ only in which slice of the 53-token output drives the sampling distribution:

- **PRISM-Full**: the full alpha-gated logit (Eq. 6), i.e., the standard generative output of the trained model.

- **PRISM-NGL**: restrict sampling to lowercase (NGL) tokens only, i.e., the model is forced to emit a non-germline substitution at every chosen position.

- **PRISM-GL**: restrict sampling to uppercase (GL) tokens only, i.e., the model is forced to emit a germline-consistent residue at every chosen position.

Baselines (ESM2-35M, AbLang2, IgLM) generate from their native distributions under the same position-selection protocol; for ESM2-35M and AbLang2 we sample from the masked-LM logit, and for IgLM we sample from the corresponding autoregressive distribution conditioned on the unmasked context. We generate 100 variants per antibody per method at $n_{\mathrm{mut}} = 10$, yielding 600 variants per antibody and 1,800 variants in total across the three antibodies.

### E.2. Rosetta Interface and Stability $\Delta\Delta G$

**Software and configuration.** We use PyRosetta-4 (release 2025.43+release.b230e431d8ef0bcdea01dbb0065ca62c7dd694ad, Python 3.12) with the `ref2015` score function (`ScoreFunctionFactory.create_score_function("ref2015")`). PyRosetta is initialized with `-ignore_unrecognized_res -ex1 -ex2 -mute all -constant_seed -jran 1234`: `-ex1 -ex2` enables extra rotamer $\chi_1/\chi_2$ sampling, and `-constant_seed -jran 1234` pins the random seed for cross-worker reproducibility.

**Crystal templates and interface definitions.** We use cleaned PDB templates with HETATM ligands stripped (e.g., NAG, EDO, which lack $C_\alpha$ atoms and crash the packer). The complexes used and the corresponding interface strings supplied to `InterfaceAnalyzerMover` are summarized in Table S10.

*Table S10.* PDB templates and Rosetta interface strings used for binding-energy scoring.

| Antibody | PDB | Antibody chains | Antigen chain(s) | Interface string |
|---|---|---|---|---|
| Trastuzumab (anti-HER2) | 1N8Z (Cho et al., 2003) | H = B, L = A | C (HER2) | BA_C |
| CR9114 (anti-influenza HA) | 4FQI (Dreyfus et al., 2012) | H = H, L = L | A (HA1) + B (HA2) | HL_AB |
| G6.31 (anti-VEGF) | 2FJH (Chen et al., 1999) | H = H, L = L | V + W (VEGF dimer) | HL_VW |

For CR9114 we explicitly include *both* HA1 (chain A) and HA2 (chain B) in the interface string: CR9114 is a stem-binding antibody, and an initial single-chain interface assignment (`HL_A`) yielded an unphysical separated-state energy ($dG_{\mathrm{separated}} = 4.4 \times 10^7$); after switching to `HL_AB` the WT interface energy is well-behaved at $-44.2$ kcal/mol.

**Step 1 – WT FastRelax (per antibody, executed once and cached).** The WT pose is FastRelaxed with backbone coordinate constraints to the crystal positions, one cycle:

```
fr = FastRelax(scorefxn, n_cycles=1)
fr.constrain_relax_to_start_coords(True)
fr.set_scorefxn(scorefxn)
fr.apply(pose)
```

The relaxed WT pose is cached on disk and reused across all mutant evaluations of the same antibody, so every variant is referenced to a single shared baseline. Per-antibody WT baselines after FastRelax are reported in Table S11.

**Step 2 – Mutation introduction.** For each variant we clone the cached WT-relaxed pose and apply the requested mutations through `MutateResidue`:

*Table S11.* Per-antibody WT baselines after FastRelax (single FastRelax cycle, backbone coordinate-constrained to crystal).

| Antibody | wt $d\text{G}_{\text{bind}}$ (kcal/mol) | wt energy (REU) | Relax wall time |
|---|---|---|---|
| Trastuzumab | $-38.7$ | $-2860$ | $\sim24$ min |
| CR9114 | $-44.2$ | $-2839$ | $\sim21$ min |
| G6.31 | $-123.7$ | $-3176$ | $\sim25$ min |

```
mut_pose = wt_relaxed.clone()
for resnum, aa1 in zip(mut_resnums, mut_aas):
    MutateResidue(resnum, AA_1TO3[aa1]).apply(mut_pose)
```

Variant identities (e.g., `H:K30A, L:F53T`) are parsed into chain/position/WT-aa/mut-aa tuples; we validate that the WT amino acid at (`chain, pos`) matches the parsed identity and skip mismatched entries.

**Step 3 – Local FastRelax of the mutant.** To keep total wall time tractable while preserving accuracy on $\Delta\Delta G$, we restrict relaxation to an 8 Å shell ($C_\alpha$–$C_\alpha$) around any mutation site, using a `MoveMap` that frees backbone and side-chain torsions only at residues within the shell:

```
def local_fastrelax(pose, scorefxn, mut_resnums, shell=8.0, n_cycles=1):
    movable = set(mut_resnums)
    for i in range(1, pose.total_residue() + 1):
        if i in movable: continue
        ca_i = pose.residue(i).xyz("CA")
        for m in mut_resnums:
            if ca_i.distance(pose.residue(m).xyz("CA")) <= shell:
                movable.add(i); break
    mm = MoveMap(); mm.set_bb(False); mm.set_chi(False)
    for i in movable: mm.set_bb(i, True); mm.set_chi(i, True)
    fr = FastRelax(scorefxn, n_cycles); fr.set_movemap(mm)
    fr.constrain_relax_to_start_coords(True); fr.apply(pose)
```

The movable set typically contains 30–100 residues depending on how the 10 mutations are distributed. Average per-variant cost is $\sim4$ min on a single thread.

**Step 4 – Interface $\Delta G_{\text{bind}}$ scoring.** We use `InterfaceAnalyzerMover` configured with `compute_interface_energy=True`, `pack_separated=True`, and the chain-specific interface string from Table S10. Setting `pack_separated=True` is essential: it repacks the side chains on the separated state, removing the systematic bias that otherwise arises from comparing a relaxed bound pose against an unrelaxed separated pose. We extract `dG_separated` from the resulting score map.

**Step 5 – Stability score and $\Delta\Delta G$ definitions.** The stability energy is the antibody-only `ref2015` energy of the relaxed mutant pose, evaluated on the antibody chains alone. The two reported $\Delta\Delta G$ quantities are

$$\Delta\Delta G_{\text{interface}} = d\text{G}_{\text{bind}}(\text{mutant}_{\text{relaxed}}) - d\text{G}_{\text{bind}}(\text{WT}_{\text{relaxed}}), \tag{24}$$

$$\Delta\Delta G_{\text{stability}} = E_{\text{total}}(\text{mutant}_{\text{relaxed}}) - E_{\text{total}}(\text{WT}_{\text{relaxed}}). \tag{25}$$

The WT baseline is computed once per antibody (Step 1) and reused across all mutants, so all variants of an antibody share a single reference value, and a variant is reported as "useful" on a given axis if the corresponding $\Delta\Delta G$ is favorable (negative for binding/stability).

**Parallelization and runtime.** Mutant evaluations are parallelized through a `ProcessPoolExecutor` with 32 workers on AMD EPYC 7573X (32-core, 2 sockets, SMT2 = 128 logical CPUs). Each worker runs PyRosetta init and `scorefxn` construction once; the WT pose is cached per antibody at the first hit. We pin `OMP_NUM_THREADS = MKL_NUM_THREADS = OPENBLAS_NUM_THREADS = 1` on each worker to prevent oversubscription, with chunksize 2–4. Aggregate through-put is $\sim0.17$ mutations/s ($\sim4$ min/variant single-thread equivalent). Total wall time for the full generation experiment (3 antibodies $\times$ 6 methods $\times$ 100 variants + a 700-variant CR9114 re-run for the `HL_AB` fix): WT FastRelax $\sim70$ min, mutant local FastRelax + scoring $\sim3.5$ h, CR9114 re-run $\sim1$ h, total $\sim5$–6 h.

### E.3. CamSol Solubility Score

We compute solubility predictions through the CamSol (Sormanni et al., 2015) web server at pH 7.0, with all other parameters at their server defaults. Each generated variant is submitted as a single sequence; we report $\Delta$CamSol = CamSol(mutant) $-$ CamSol(WT), so a positive value corresponds to predicted solubility improvement relative to wild type.

### E.4. Ridge Regression Binding-Affinity Predictors

**Model.** For each binding-affinity DMS dataset (CR9114, G6.31, Trastuzumab), we train a ridge regression model on mean-pooled AntiBERTy embeddings of paired heavy and light chains to predict experimental affinity. Models are fit with mean-squared-error loss and L2 regularization. Performance is evaluated via 5-fold cross-validation, after which each model is retrained on the full dataset and used to score generated variants.

**Cross-validation and reliability.** 5-fold CV Spearman correlations (Fig. S2) are $\rho = 0.715 \pm 0.003$ (CR9114), $0.512 \pm 0.007$ (Trastuzumab), and $0.367 \pm 0.014$ (G6.31). Low variance across folds ($\leq 0.014$) indicates stable training, while differences in $\rho$ reflect dataset size and landscape complexity. These values define a dataset-specific ceiling on predictive fidelity and thus an empirical upper bound on the reliability of scores for out-of-distribution sequences, including generated 10-mutant variants.

**Position coverage and interpretation.** Reliability further depends on positional coverage in the underlying DMS data. G6.31 is fully site-saturated, so predictions for multi-mutant variants are largely interpolative, and we treat its scores as the most reliable despite lower absolute $\rho$. In contrast, CR9114 and Trastuzumab probe only subsets of positions, making predictions on unseen positions extrapolative. Accordingly, we treat their ridge-based scores as supportive but not definitive; where disagreements arise (Fig. 3C–D), Rosetta interface $\Delta\Delta G$ is taken as the primary binding signal.

**Implications.** These limitations do not affect the qualitative conclusions: PRISM-NGL preferentially samples binding-favorable variants—supported by both the learned predictor and Rosetta on G6.31, and by Rosetta on CR9114 and Trastuzumab—while PRISM-GL preferentially samples developability-favorable variants, as consistently supported across all three antibodies by Rosetta stability and CamSol.

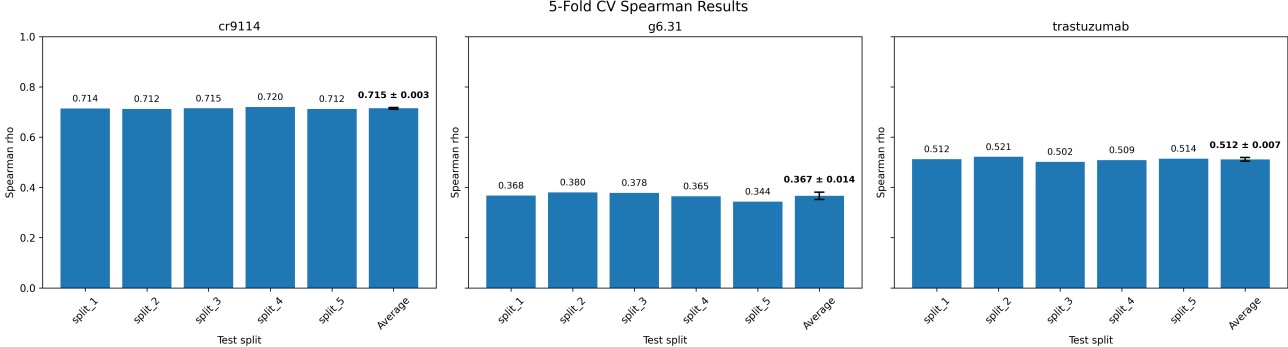

*Figure S2.* **5-fold cross-validation of the ridge regression binding-affinity predictors against DMS labels.** Spearman $\rho$ on each held-out fold and the across-fold mean ($\pm$ standard deviation): CR9114 attains $\rho = 0.715 \pm 0.003$ (left), G6.31 attains $\rho = 0.367 \pm 0.014$ (middle), and Trastuzumab attains $\rho = 0.512 \pm 0.007$ (right). The CR9114 predictor is the strongest of the three, consistent with the larger DMS sample size ($N = 65,093$); the G6.31 predictor is the weakest, reflecting the smaller dataset ($N = 4,274$) and the harder near-optimum landscape. All three predictors are well above chance, but the absolute correlation ceilings indicate that ridge-based scores should be read as approximate rather than definitive on out-of-distribution variants (see Appendix E.4).

## F. Detailed Pseudo-Perplexity Performance Analysis by Region and Origin

To comprehensively assess PRISM's generative capabilities and its mastery of antibody sequence grammar, we decomposed the pseudo-perplexity evaluation by biological regions and evolutionary origins (**Figure S3**).

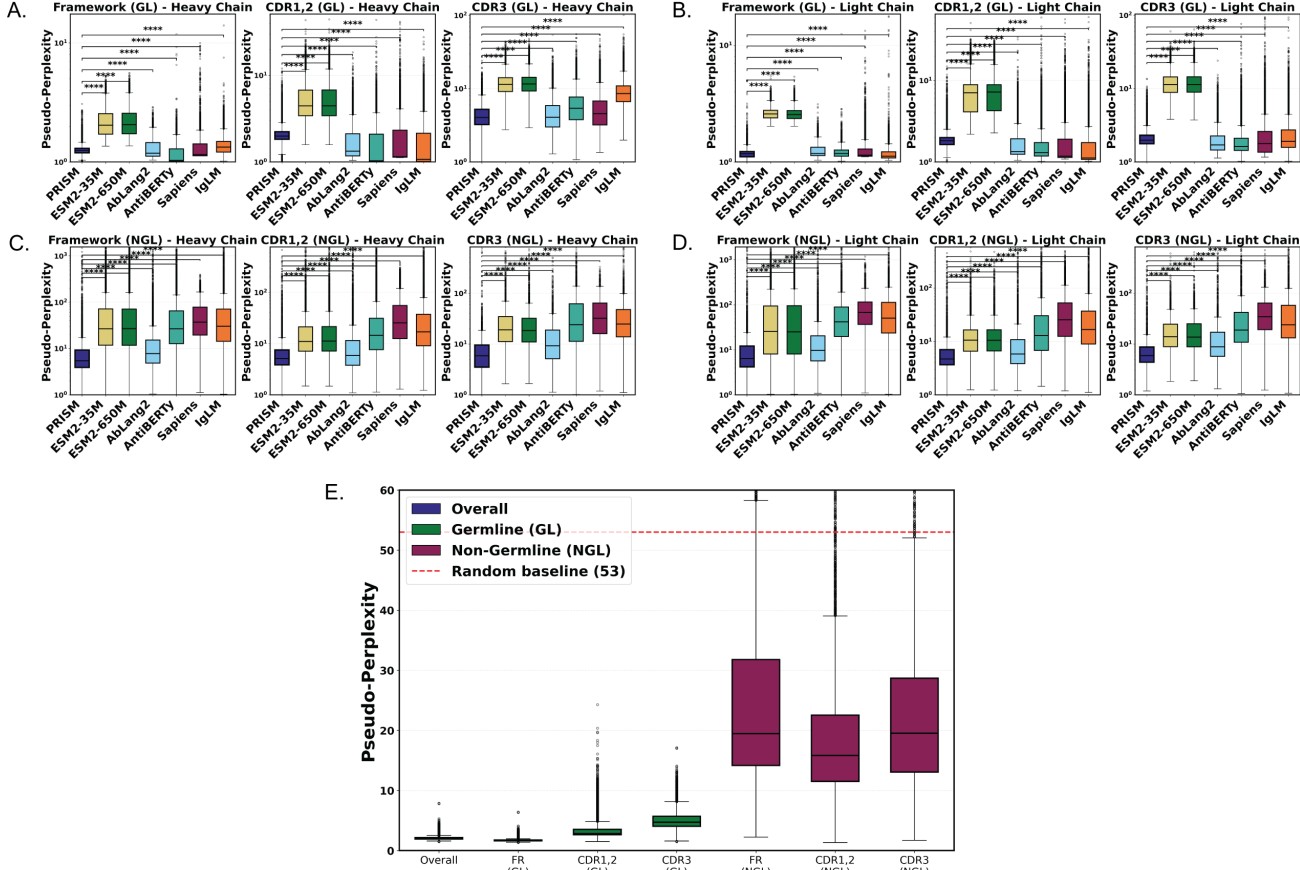

*Figure S3.* **Stratified Pseudo-Perplexity Analysis across Biological Regions and Evolutionary Origins. (A-D) Comparative analysis using Marginalized PPL.** Performance is stratified by chain (Heavy vs. Light) and residue origin (Germline [GL] vs. Non-Germline [NGL]). PRISM (dark blue) is compared against general protein language models (ESM2-35M, ESM2-650M), masked antibody language models (AbLang2, AntiBERTy, Sapiens), and the autoregressive antibody model IgLM. PPL values are normalized to the standard 20-amino acid vocabulary for fair comparison. **(A, B)** While all antibody language models perform well on conserved GL residues, **(C, D)** PRISM achieves significantly lower perplexity on the challenging NGL residues, demonstrating superior modeling of non-germline mutations. **(E) Internal Generative Performance (Exact PPL).** Distribution of PRISM's Exact PPL on its native 53-token vocabulary, stratified by region. The red dashed line indicates the random prediction baseline (PPL = 53). While NGL residues (purple) naturally exhibit higher entropy than conserved GL residues (green), PRISM's median NGL perplexity remains significantly below the random baseline across all regions, confirming effective learning of mutational probability distributions. Significance markers: **** $p < 0.0001$ (Wilcoxon signed-rank test).

## F.1. Robustness in non-germline mutations (Marginalized PPL).

Figures S3A-D compare PRISM against baseline models using **Marginalized PPL** to ensure a fair comparison on the standard 20-amino acid vocabulary.

- **Germline (GL) Residues:** On conserved GL residues (Panels A, B), PRISM matches or exceeds the performance of state-of-the-art baselines. This indicates that our dual-token vocabulary effectively preserves the fundamental germline syntax despite the expanded search space.

- **Non-Germline (NGL) Residues:** The most significant advantage is observed in NGL residues (Panels C, D). Baseline models often exhibit high perplexity spikes (PPL > 100) for mutations, interpreting them as unlikely errors or noise. In contrast, PRISM maintains consistently low perplexity across both Heavy and Light chains. This validates that explicitly modeling evolutionary origin allows the model to accept and predict valid non-germline mutations as high-probability events.

## F.2. Learning Mutational Landscapes (Exact PPL).

Figure S3E illustrates PRISM's internal generative capability using **Exact PPL** on its native 53-token vocabulary.

- As expected, highly conserved GL residues (green boxplots) are predictable with high confidence (PPL $\approx 1$).

- Crucially, for NGL residues (purple boxplots), which represent stochastic evolutionary events, PRISM achieves a median perplexity of $\sim 20$ across CDRs. This is substantially lower than the random baseline of 53 (red dashed line). This statistically confirms that the model is not merely guessing mutations but has learned to constrain the mutational search space to biologically plausible trajectories.

## G. Generation Diagnostics: Per-Metric Distributions and Spatial Patterns

Complementing the methodological description in Appendix E, this section reports two diagnostic figures of the generated 10-mutation variants: per-metric distributions across all three antibodies (Fig. S4) and the spatial distribution of generated mutations along the heavy- plus light-chain axis (Fig. S5).

## G.1. Per-Metric Distributions of Generated Variants (Fig. S4)

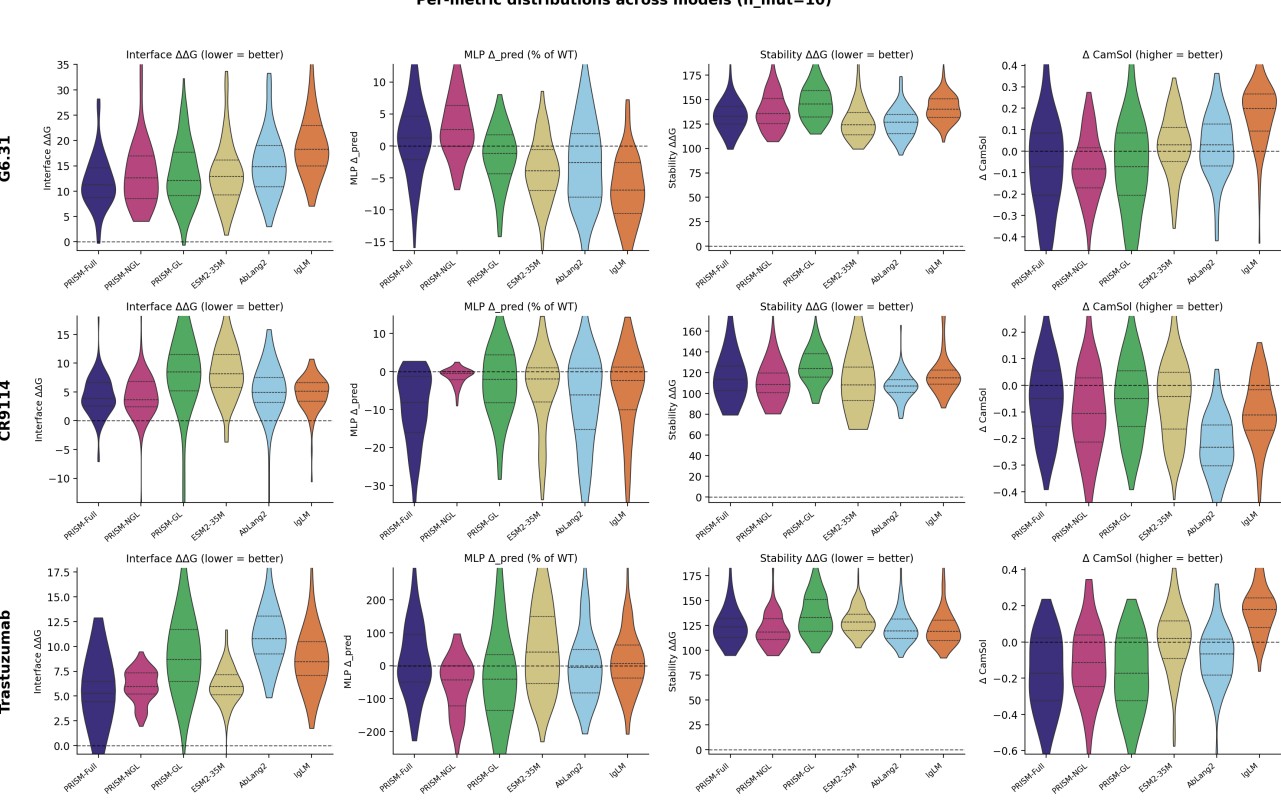

*Figure S4.* **Per-metric distributions of generated variants ($n_{\text{mut}} = 10$), stratified by method and antibody.** Violin plots of the four scoring axes – Rosetta interface $\Delta\Delta G$ (lower = better), ridge-based $\Delta_{\text{pred}}$ (% of WT), Rosetta stability $\Delta\Delta G$ (lower = better), and $\Delta$CamSol (higher = better) – across the three antibodies (rows: G6.31, CR9114, Trastuzumab) and six methods (PRISM-Full, PRISM-NGL, PRISM-GL, ESM2-35M, AbLang2, IgLM). Horizontal dashed lines indicate WT baseline. The distributions complement the %-useful-variants summary in Fig. 3B–D and visualize the magnitude of property changes per variant.

The full per-variant distributions in Fig. S4 sharpen the property-control narrative beyond what the %-useful-variants summary in Fig. 3B–D conveys. On the Rosetta interface $\Delta\Delta G$ axis (left column), PRISM-NGL distributions are systematically shifted toward more negative values across all three antibodies, with the largest shifts on G6.31 and CR9114; PRISM-GL and the baselines concentrate more tightly around zero or in the unfavorable regime. On the Rosetta stability

and $\Delta$CamSol axes (third and fourth columns), the ordering inverts: PRISM-GL produces the tightest, most stability- and solubility-preserving distributions, with PRISM-Full sitting between the two extremes – the signature of the alpha-gated combination integrating evidence from both channels. The ridge-based $\Delta_{\mathrm{pred}}$ axis (second column) shows the heterogeneity flagged in Appendix E.4: on G6.31 the modal PRISM-NGL value lies above WT and the channel ordering matches the Rosetta interface signal, whereas on CR9114 and Trastuzumab the ridge-based distributions are wider and the relative ordering is less clean, consistent with the partial-position DMS coverage on those two datasets.

### G.2. Spatial Pattern of Generated Mutations (Fig. S5)

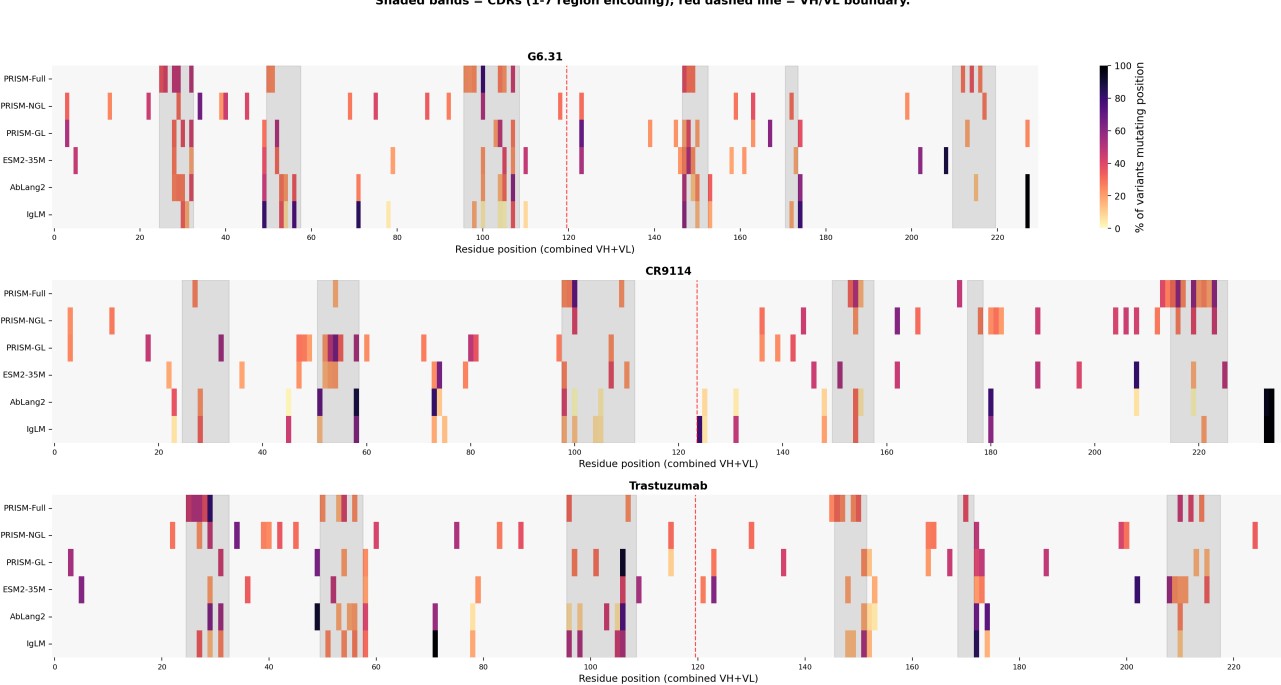

*Figure S5.* **Mutation position frequency along the heavy- plus light-chain axis** ($n_{\mathrm{mut}} = 10$)**.** For each method (rows) and antibody (G6.31, top; CR9114, middle; Trastuzumab, bottom), the heatmap encodes the fraction of generated variants mutating each VH+VL position; shaded bands indicate CDRs (1–7 region encoding) and the red dashed line marks the VH/VL boundary.

The spatial pattern of mutations in Fig. S5 provides a geometric corroboration of the property-control behavior reported in the main text. PRISM-NGL concentrates substitutions at CDR positions (shaded bands) – the structural region where somatic hypermutation, and therefore binding-affinity-modulating variation, is biologically concentrated – with the highest density on the heavy-chain CDR3 region, consistent with CDR3's role as the dominant antigen-contacting loop. PRISM-GL spreads mutations more broadly, with non-trivial density at framework-adjacent positions where germline conservation is the dominant prior. PRISM-Full again sits between the two extremes, distributing mutations across both regions in proportions that depend on the alpha-gate. The baselines (ESM2-35M, AbLang2, IgLM) either concentrate mutations outside the CDRs or distribute them sparsely without the CDR-centric concentration that PRISM-NGL exhibits, providing a position-level explanation for why their generated variants underperform on the binding-favored Rosetta interface metric (Fig. S4, left column).

## H. Extended Zero-Shot Analysis: Metric Robustness and Hypothesis Validation

In this section, we provide a comprehensive validation of PRISM's zero-shot prediction capabilities along three complementary axes. We first present detailed benchmarking of developability using Spearman correlation in **Figure S6**, highlighting PRISM's superior balance compared to baselines; we then examine the robustness of these results under Pearson correlation in **Figure S7**; next, we translate these correlation metrics into practical screening utility via Top-$K$ Recall and Enrichment analysis across three DMS benchmarks (Appendix H.3) to directly address whether weak positive correlations support therapeutic prioritization.

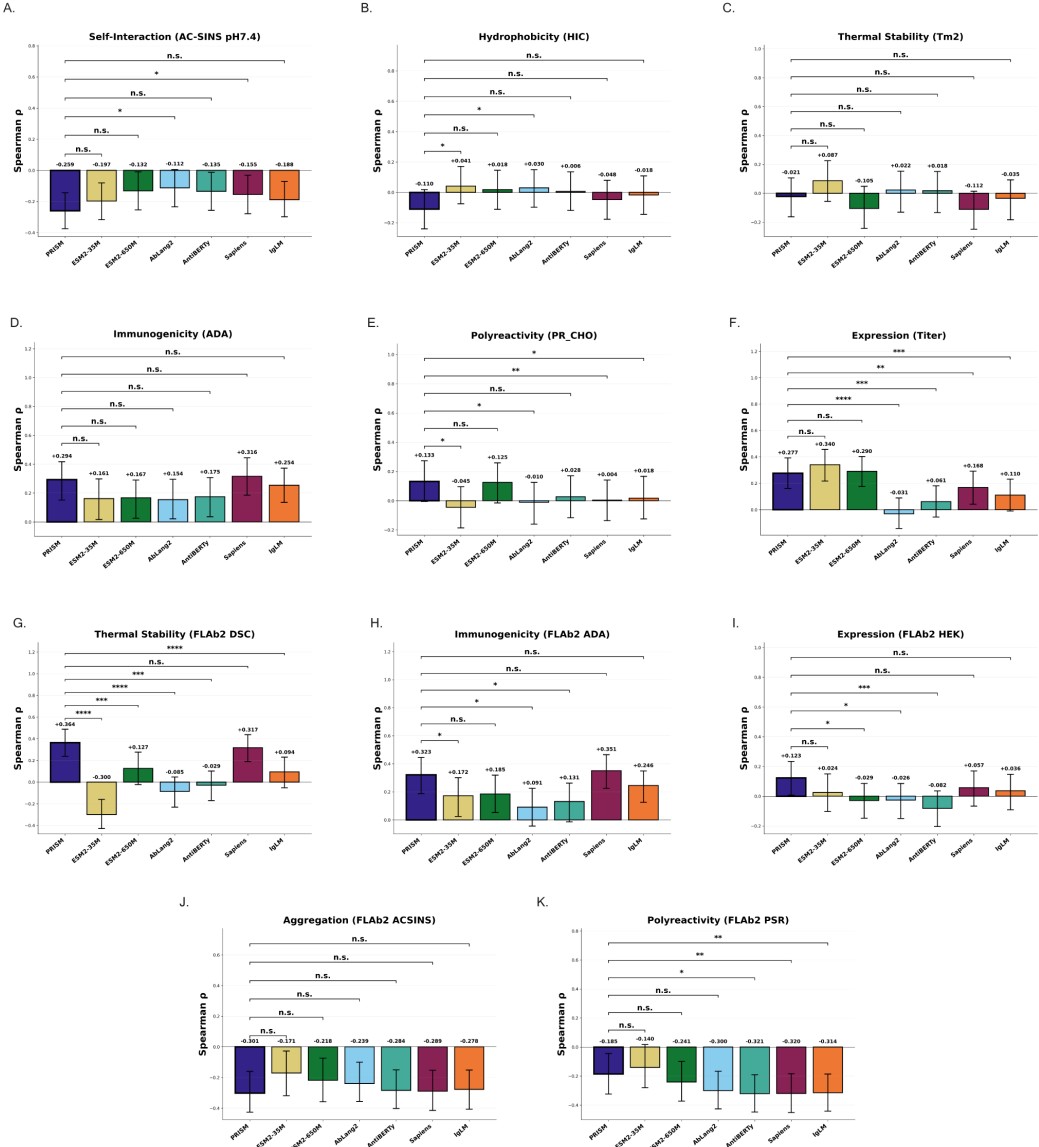

*Figure S6.* **Zero-shot Prediction of Developability (Spearman $\rho$).** We compare the rank correlation of PRISM against general protein language models (ESM2-35M, ESM2-650M), specialized antibody language models (AbLang2, AntiBERTy, Sapiens), and the autoregressive antibody model IgLM, across the original Arsiwala/Marks 6-trait panel (A–F) and the recently released **FLAb2** biophysical-assay benchmark (G–K). **(A–C) Stability traits (Arsiwala et al.):** PRISM ranks first in Self-interaction (AC-SINS), Hydrophobicity (HIC), and Thermal Stability (Tm2), significantly outperforming other ALMs. Notably, AbLang2 shows negative correlations for Self-interaction. **(D–F) Reactivity & Immunogenicity (Arsiwala/Marks):** PRISM remains highly competitive, ranking second in Immunogenicity (ADA) and Polyreactivity (PR_CHO), confirming that stability optimization does not compromise other essential properties. **(G–K) FLAb2 panel:** On the broader FLAb2 benchmark, PRISM leads on Thermal Stability (DSC, panel G) and Aggregation (AC-SINS, panel J), and remains competitive on Immunogenicity (ADA, H), Expression (HEK, I), and Polyreactivity (PSR, K), reinforcing the robustness of PRISM's developability profile across an order-of-magnitude broader assay cross-section.

## H.1. Developability Benchmark: PRISM vs. Baselines

Figure S6 details the zero-shot performance across six developability metrics using Spearman rank correlation ($\rho$). The results demonstrate PRISM's unique capability to model structural stability without explicit supervision:

- **Superiority in Stability (Fig. S6A-C):** PRISM achieves the highest correlations in Self-interaction ($\rho = 0.181$), Hydrophobicity ($\rho = 0.093$), and Thermal Stability ($\rho = 0.167$). In stark contrast, other antibody-specific models struggle in these regimes; for instance, AbLang2 exhibits a negative correlation for Self-interaction ($\rho = -0.124$) and

near-zero performance for Thermal Stability ($\rho = 0.016$), suggesting that standard pretraining on antibody sequences alone fails to capture thermodynamic constraints.

- **Balanced Profile (Fig. S6D-F):** PRISM maintains robust performance in Immunogenicity ($\rho = 0.310$), performing on par with the best baseline, Sapiens ($\rho = 0.316$), and outperforming AbLang2 ($\rho = 0.159$). While ESM2 models show high correlations for Expression, they fail to generalize to other critical metrics like Self-interaction, whereas PRISM offers the most consistent performance across the full developability spectrum.

## H.2. Metric Consistency: Pearson Correlation Analysis

To ensure that our findings are not artifacts of ranking metrics, we present Pearson correlation coefficients ($r$) in **Figure S7**. The results reveal trends strictly consistent with the Spearman analysis:

- **Binding Affinity (Fig. S7A-C):** PRISM maintains positive correlations across all three datasets. In contrast, general PLMs often exhibit strong negative correlations ($r < -0.3$ on CR9114). Notably, even specialized ALMs such as AbLang2 and AntiBERTy fail to consistently capture affinity in complex regimes, yielding negative correlations on CR9114, which indicates that domain pretraining alone is insufficient without explicit disentanglement.

- **Developability (Fig. S7D-I):** PRISM demonstrates superior robustness, particularly in Self-Interaction ($r = 0.156$) and Immunogenicity ($r = 0.450$), validating that the model's generative probability linearly maps to biophysical stability.

## H.3. Practical Screening Utility: Top-$K$ Recall and Enrichment Analysis

Rank-correlation metrics ($\rho, r$) quantify the global agreement between model scores and experimental measurements, but they do not directly answer the question that matters for therapeutic discovery: *if we select the top-scoring candidates and assay them, how many are actually high-affinity binders?* To address this, we report Top-$K$ Recall (the fraction of ground-truth top-10% binders recovered within the model's top $K\%$ predictions) and the Enrichment Factor at 10% (Enrich@10%, the ratio of the true positive rate in the top $10\%$ to the prior rate) across all three DMS benchmarks (Tables S12–S14).

**G6.31: PRISM is the only model with positive monotonic alignment.** G6.31 is a strongly optimized anti-VEGF antibody ($K_d \approx 0.4$ nM) obtained through multiple rounds of phage display and targeted mutagenesis (Lee et al., 2004; Fuh et al., 2006), and the single-mutant dataset of Koenig et al. (2017b) probes mutations around a sequence already near a local affinity optimum, a regime in which the overwhelming majority of single mutations are neutral or deleterious and the signal-to-noise ratio for any mutant-centered likelihood score is inherently low. Despite this difficulty, PRISM is the *only* compared method to achieve a positive Spearman correlation on G6.31 (Table S12, $\rho = +0.158$, 2.3× enrichment); every baseline yields negative ranks, ranging from Sapiens at $\rho = -0.018$ down to ESM2-35M at $\rho = -0.202$. Notably, IgLM nominally exceeds PRISM's enrichment (2.6× vs. 2.3×) yet returns a strongly *negative* Spearman ($\rho = -0.065$), indicating that its top-10% recall is not driven by systematic affinity tracking but rather by an incidental overlap between high-likelihood sequences and high-affinity variants on this near-optimum landscape; this pattern — nominally favorable enrichment paired with anti-correlated rank order — is exactly the failure mode that motivates reporting both metrics jointly, and it underscores that PRISM is the unique method on G6.31 combining positive monotonic alignment with practical screening utility.

*Table S12.* **Practical screening metrics on G6.31 ($N = 4{,}274$, anti-VEGF).** On this strongly affinity-matured single-mutation DMS landscape (Koenig et al., 2017a), PRISM is the only method achieving a positive Spearman correlation; IgLM nominally exceeds PRISM's enrichment (2.6× vs. 2.3×) but with a *negative* Spearman ($\rho = -0.065$), illustrating the failure mode of evaluating enrichment in isolation when a model's global ranking is anti-correlated with affinity.

| Model | Spearman $\rho$ | Recall@1% | Recall@5% | Recall@10% | Enrich@10% |
|---|---|---|---|---|---|
| PRISM | **+0.158** | **0.070** | **0.168** | 0.231 | **2.3×** |
| Sapiens | −0.018 | 0.047 | 0.107 | 0.171 | 1.7× |
| IgLM | −0.065 | 0.047 | 0.192 | 0.257 | 2.6× |
| AntiBERTy | −0.074 | 0.023 | 0.159 | 0.194 | 1.9× |
| AbLang2 | −0.115 | 0.047 | 0.136 | 0.201 | 2.0× |
| ESM2-650M | −0.150 | 0.023 | 0.070 | 0.161 | 1.6× |
| ESM2-35M | −0.202 | 0.000 | 0.065 | 0.114 | 1.1× |

**CR9114: PRISM is the only model delivering practical enrichment.** The CR9114 anti-influenza benchmark (Table S13) shows the largest gap between PRISM and competing methods: PRISM achieves 4.6× enrichment — equivalent to a

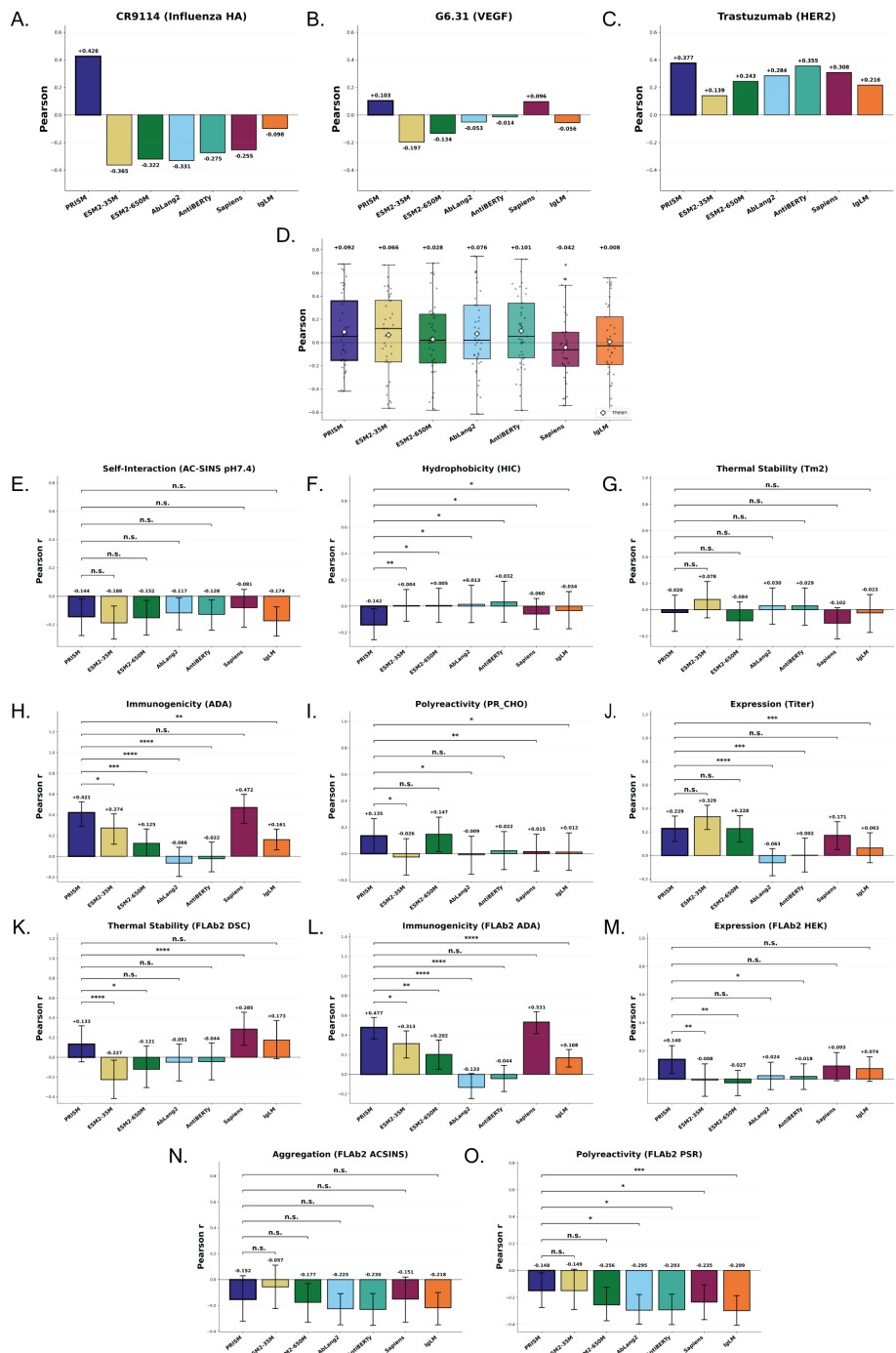

*Figure S7.* **Metric Robustness Check using Pearson Correlation ($r$).** To validate the linear proportionality of model predictions, we extend the analysis to Pearson correlation across all binding-affinity DMS sets and the full developability panel (Arsiwala/Marks 6-trait set + FLAb2 5-assay set). **(A–C) Binding Affinity:** PRISM consistently avoids the negative correlations frequently observed in baselines (e.g., ESM2, AntiBERTy) on the combinatorially complex CR9114 landscape, mirroring the Spearman ranking. **(D) FLAb2 aggregate:** Distribution of per-assay Pearson $r$ across all 41 FLAb2 assays; PRISM's median lies above all baselines. **(E–J) Arsiwala/Marks developability traits:** The trends observed in Spearman rankings are preserved, with PRISM showing robust positive linearity for Immunogenicity and Self-interaction. **(K–O) FLAb2 developability panel (DSC, ADA, HEK, AC-SINS, PSR):** Pearson trends agree with the Spearman analysis in Figure S6G–K, with PRISM leading on FLAb2 thermal stability and aggregation, further validating the model's reliability across an expanded biophysical cross-section.

roughly $\sim 4.6\times$ reduction in phage-display screening burden at fixed recovery — while every baseline produces *anti-enrichment* (Enrich@10% $\in [0.1, 0.4]$), meaning that top-ranked candidates from those models are substantially less likely to bind than random selection. The pattern is unanimous across both general pLMs (ESM2-650M: $\rho = -0.355$, $0.4\times$) and specialized ALMs (AntiBERTy: $\rho = -0.364$, $0.2\times$; AbLang2: $\rho = -0.428$, $0.1\times$). This is the regime in which PRISM's evolutionary disentanglement provides its strongest practical value: the complex epistatic landscape of CR9114 ($N = 65{,}093$ combinatorial variants of a broadly neutralizing antibody) defeats both general pLMs and specialized ALMs, whereas PRISM's factorized prior surfaces true hits at rates meaningful for downstream wet-lab validation.

*Table S13*. **Practical screening metrics on CR9114** ($N = 65{,}093$**, anti-influenza).** PRISM is the only method delivering positive enrichment; all baselines exhibit *anti-enrichment* ($< 1\times$) and strongly negative Spearman correlations ($\rho \in [-0.326, -0.428]$), indicating that top-ranked candidates from those models are *less* likely to bind than random selection on this epistatically complex combinatorial landscape.

| Model | Spearman $\rho$ | Recall@1% | Recall@5% | Recall@10% | Enrich@10% |
|---|---|---|---|---|---|
| PRISM | **+0.391** | **0.319** | **0.445** | **0.462** | **4.6×** |
| Sapiens | −0.326 | 0.002 | 0.012 | 0.022 | 0.2× |
| ESM2-650M | −0.355 | 0.012 | 0.022 | 0.038 | 0.4× |
| AntiBERTy | −0.364 | 0.002 | 0.007 | 0.021 | 0.2× |
| IgLM | −0.374 | 0.002 | 0.007 | 0.019 | 0.2× |
| ESM2-35M | −0.382 | 0.000 | 0.012 | 0.029 | 0.3× |
| AbLang2 | −0.428 | 0.002 | 0.006 | 0.014 | 0.1× |

**Trastuzumab: PRISM leads on both correlation and enrichment.** On the Trastuzumab anti-HER2 benchmark (Table S14), PRISM achieves the highest Spearman correlation ($\rho = +0.327$) and the highest enrichment ($2.6\times$) across all compared methods, narrowly surpassing the strongest specialized ALM baselines (AntiBERTy: $\rho = +0.297$, $2.5\times$; Sapiens: $\rho = +0.260$, $2.0\times$; AbLang2: $\rho = +0.241$, $2.1\times$) and clearly outperforming general pLMs (ESM2-650M: $\rho = +0.203$, $1.8\times$). All antibody-specialized models deliver practically meaningful screening gains on this moderately complex landscape, but PRISM is the only method to combine top-ranked Spearman *and* top-ranked enrichment across the full range of target-difficulty regimes spanned by G6.31, CR9114, and Trastuzumab.

*Table S14*. **Practical screening metrics on Trastuzumab** ($N = 36{,}496$**, anti-HER2).** PRISM leads on both Spearman $\rho$ and enrichment, narrowly surpassing the strongest specialized ALM baseline (AntiBERTy: $\rho = 0.297$, $2.5\times$). All antibody-specialized models deliver practically meaningful screening gains on this moderately complex landscape.

| Model | Spearman $\rho$ | Recall@1% | Recall@5% | Recall@10% | Enrich@10% |
|---|---|---|---|---|---|
| PRISM | **+0.327** | 0.025 | 0.150 | **0.262** | **2.6×** |
| AntiBERTy | +0.297 | **0.038** | 0.141 | 0.250 | 2.5× |
| Sapiens | +0.260 | 0.030 | 0.110 | 0.202 | 2.0× |
| AbLang2 | +0.241 | 0.033 | 0.108 | 0.209 | 2.1× |
| ESM2-650M | +0.203 | 0.019 | 0.096 | 0.183 | 1.8× |
| IgLM | +0.187 | 0.022 | 0.092 | 0.172 | 1.7× |
| ESM2-35M | +0.120 | 0.005 | 0.072 | 0.145 | 1.5× |

**Interpretation.** Taken together, the three benchmarks establish PRISM as the single method with consistent zero-shot binding-affinity signal across the full target-difficulty spectrum. PRISM uniquely maintains a positive Spearman correlation on *every* benchmark (G6.31: $+0.158$; CR9114: $+0.391$; Trastuzumab: $+0.327$), whereas every other compared method either correlates negatively on the more challenging targets — on CR9114, every baseline collapses to $\rho < -0.32$ and to anti-enrichment ($0.1\times$–$0.4\times$); on G6.31, every baseline yields negative ranks — or produces incidental enrichment without monotonic alignment, as IgLM does on G6.31 ($2.6\times$ enrichment paired with $\rho = -0.065$). On the moderately complex Trastuzumab landscape PRISM additionally surpasses all specialized antibody language models on both correlation and enrichment ($\rho = +0.327$, $2.6\times$ vs. AntiBERTy's $+0.297$, $2.5\times$), while on the epistatically complex CR9114 landscape PRISM is the only method delivering meaningful enrichment ($4.6\times$ vs. $0.1\times$–$0.4\times$ anti-enrichment for baselines). Rather than positioning PRISM as a universal screening oracle, we frame it as the one method whose predictive direction and screening enrichment remain aligned with true affinity across the regime of combinatorial complexity relevant to therapeutic affinity maturation; the magnitude of PRISM's advantage scales with each landscape's mutational complexity, and the consistency of its sign across all three regimes is itself a defining property.

# I. Extended Ablation Analysis

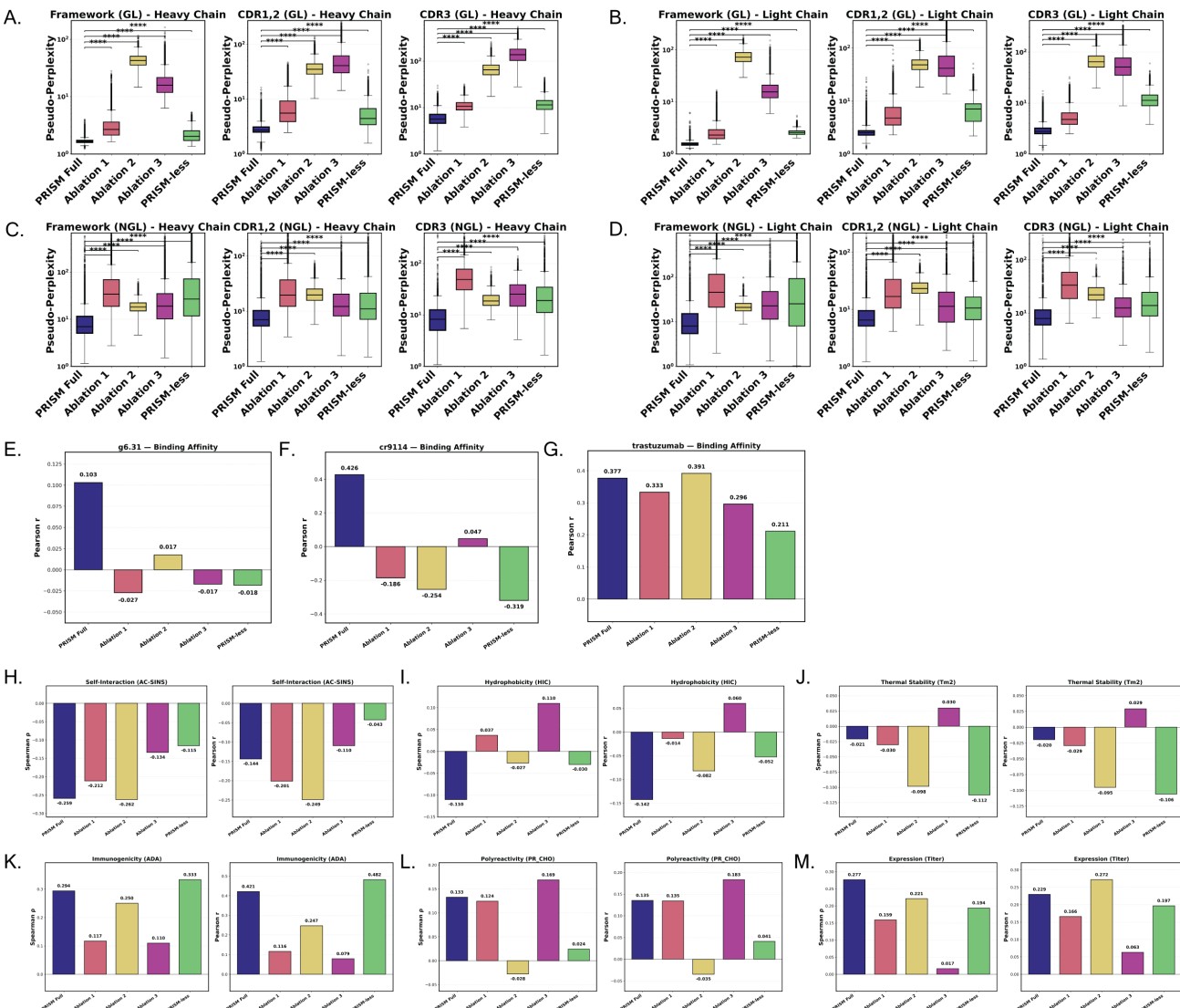

*Figure S8.* **Ablation Study: Region-Stratified Performance, Binding Affinity, and Biophysical Profiling.** Complementing the main text analysis, we decompose the performance of ablation variants across all four axes (Ablation 1: no pretraining; Ablation 2: simple head, pretrained; Ablation 3: simple head, no pretraining; PRISM-less: ESM2-initialized with antibody fine-tuning and simple head). **(A-D) Stratified Pseudo-Perplexity.** Breakdown of failure modes. **(A, B)** Ablation 2 and Ablation 3 suffer primarily in Germline (GL) regions due to the extended-vocabulary bottleneck, while PRISM-less avoids this penalty by operating on the standard vocabulary. **(C, D)** All ablation variants, including PRISM-less, degrade substantially on non-germline (NGL) regions relative to PRISM Full. **(E-G) Binding Affinity Correlations (Pearson** $r$**).** Linear correlation analysis on G6.31, CR9114, and Trastuzumab benchmarks, consistent with the Spearman rankings in the main text. **(H-M) Zero-shot Developability Correlations.** Comparison of Spearman $\rho$ and Pearson $r$ against experimental biophysical labels across six metrics. PRISM Full (dark blue) achieves the strongest or tied-strongest correlations on five of six properties, with Immunogenicity (ADA) as the sole exception where PRISM-less leads. Significance markers: **** $p < 0.0001$, ** $p < 0.01$, * $p < 0.05$, n.s. not significant.

While the main text establishes that architectural factorization is the primary driver of functional performance, **Figure S8** provides a granular view of how each ablation component influences generative quality, binding affinity, and biophysical grounding.

**I.1. Distinct Failure Modes in Sequence Modeling (Fig. S8A-D)**

Decomposing pseudo-perplexity reveals that the "collapse" observed in the main text stems from distinct, variant-specific sources:

- **Ablation 2 and 3 (Simple Head with extended vocabulary):** These variants suffer extremely high perplexity in **Germline (GL)** regions (Fig. S8A-B), confirming that without explicit token separation, a simple head cannot efficiently map conserved residues to the expanded GL/NGL token space.

- **Ablation 1 (No Pretraining, multi-head retained):** Despite preserving the multi-head architecture, removing pretraining most severely impairs **non-germline (NGL)** modeling (Fig. S8C-D). This suggests that large-scale pretraining is essential for learning the subtle evolutionary rules governing somatic mutation, even when the architecture is correctly specified.

- **PRISM-less (ESM2 initialization, simple head, standard vocabulary):** PRISM-less avoids the GL vocabulary-mapping failure of Ablations 2 and 3, achieving perplexity close to PRISM Full in GL regions (Fig. S8A-B). However, in NGL regions (Fig. S8C-D), PRISM-less degrades to levels comparable to other ablations, confirming that ESM2 initialization with antibody fine-tuning—absent explicit origin-aware routing—cannot recover the non-germline modeling advantage of PRISM Full.

**I.2. Consistency in Binding Affinity Prediction (Fig. S8E-G)**

In the main text, we reported Spearman rank correlations ($\rho$) to evaluate the monotonic relationship between model scores and binding affinity. Here, we extend this analysis to Pearson linear correlations ($r$) to assess linear proportionality.

As shown in **Figure S8E-G**, the results exhibit trends strictly consistent with the Spearman metrics reported in the main text. PRISM Full consistently achieves the highest positive correlations across all three datasets ($r = 0.103$, $0.426$, $0.377$ for G6.31, CR9114, and Trastuzumab respectively), while ablation variants frequently show negligible or negative correlations. Notably, PRISM-less attains the strongest *negative* Pearson correlation on CR9114 ($r = -0.319$), mirroring its Spearman result and reinforcing the main text's conclusion: ESM2-derived representational competence, absent architectural factorization, not only fails to predict affinity but systematically inverts the signal on somatic hypermutation benchmarks. This confirms that the architectural benefits of PRISM are robust across statistical metrics.

**I.3. Impact on Biophysical Properties (Fig. S8H-M)**

We extended the zero-shot evaluation to six developability metrics, revealing a more nuanced picture than the affinity results:

- **Complex Surface Properties (Self-Interaction):** For AC-SINS (Panel H), which depends on global surface features, PRISM Full captures the strongest signal ($\rho = -0.259$) while all ablations yield weaker correlations.

- **Structural Stability:** For Thermal Stability (Tm2, Panel J), no variant achieves strong correlation, with all models clustered near zero. This reflects the intrinsic difficulty of zero-shot Tm prediction from sequence alone rather than an architectural distinction.

- **Immunogenicity (ADA, Panel K):** PRISM-less unexpectedly outperforms PRISM Full ($\rho = 0.333$ vs. $0.294$; $r = 0.482$ vs. $0.421$). We interpret this as ESM2's broader pretraining corpus providing a sequence-level prior relevant to anti-drug antibody responses, which are driven partly by motif-level features not fully captured by antibody-only pretraining. This is the sole metric where ESM2 initialization provides a net benefit over PRISM Full.

- **Expression Titer (Panel M):** Ablation 2 (Spearman $\rho = 0.221$, Pearson $r = 0.272$) and PRISM-less ($\rho = 0.194$, $r = 0.197$) both perform comparably to PRISM Full ($\rho = 0.277$, $r = 0.229$), consistent with the hypothesis that expression yield is driven more by simple sequence motifs than the origin-aware semantics captured by the Origin Head.

- **Polyreactivity (PR_CHO, Panel L):** PRISM Full achieves the strongest correlation ($\rho = 0.133$), tied with Ablation 1, with other variants substantially weaker or negative.

Collectively, these results indicate that PRISM's architectural factorization provides robust advantages on properties with complex evolutionary and structural dependencies (affinity, self-interaction, polyreactivity), while offering smaller or comparable benefits on properties driven by local sequence features (expression, immunogenicity).

## J. Alpha-Gating: Bayesian Interpretation and Empirical Dynamics

This section formalizes the role of the learnable gate $\alpha_i$ introduced in Sec. B and analyzes its empirical behavior across biological regions and training stages. We show that $\alpha_i$ is neither a confidence decay factor in the classical Bayesian sense nor a global temperature, but rather a *learned, spatially adaptive task-relevance gate* that controls how strongly the Origin signal is injected into the per-position amino-acid distribution.

### J.1. Tempered Posterior Interpretation of Equation 6

Recall that the Origin Head produces a scalar probability $p_{\text{Origin}}(o_i = \text{NGL} \mid \mathbf{h}_i)$ for the event "position $i$ is non-germline", and the AA Head produces a categorical distribution $p_{\text{AA}}(a_i \mid \mathbf{h}_i)$ over the base vocabulary. The alpha-gated combination (Eq. 6) takes the form

$$p(\text{token}_i \mid \text{pos}) \;\propto\; p_{\text{AA}}(a_i \mid \mathbf{h}_i) \;\cdot\; p_{\text{Origin}}(o_i \mid \mathbf{h}_i)^{\alpha_i}, \tag{26}$$

which, in log-space, is a weighted log-linear pool of two unnormalized densities:

$$\log \tilde{p}(\text{token}_i) \;=\; \underbrace{\log p_{\text{AA}}(a_i \mid \mathbf{h}_i)}_{\text{AA evidence}} + \underbrace{\alpha_i \, \log p_{\text{Origin}}(o_i \mid \mathbf{h}_i)}_{\text{gated Origin evidence}}. \tag{27}$$

This expression is isomorphic to a *tempered (fractional) posterior*

$$p_\alpha(\text{token}_i \mid \mathbf{h}_i) \;\propto\; p_{\text{AA}}(a_i \mid \mathbf{h}_i) \;\cdot\; p_{\text{Origin}}(o_i \mid \mathbf{h}_i)^{\alpha_i}, \tag{28}$$

in the sense of Grünwald (2012) and Holmes & Walker (2017), in which $\alpha_i \in [0, 1]$ plays the role of a position-specific *tempering exponent*; when $\alpha_i \to 1$ the Origin pathway enters the posterior at full strength, corresponding to maximal admission of evolutionary evidence, whereas when $\alpha_i \to 0$ the Origin factor reduces to unity and the posterior collapses onto the marginal AA distribution.

Two properties distinguish this from a classical Bayesian update: *first*, $\alpha_i$ is not a frequentist likelihood-mass contraction but is *learned end-to-end through the downstream masked-language-modeling loss*, receiving no direct supervision from the Origin Head; *second*, $\alpha_i$ is position-specific, enabling per-residue modulation of the prior-vs.-likelihood trade-off, and consequently we interpret $\alpha_i$ as a learned *task-relevance* gate that admits Origin evidence only at positions where it is informative for predicting the residue identity and suppresses it elsewhere.

**Remark on the global $\alpha$ level.** A natural follow-up is to ask why most empirical $\alpha_i$ values lie well above 0.5 (typically in $[0.7, 1.0]$; see Tab. S15 and Fig. S9), given that the $\alpha \to 0$ limit collapses onto the marginal AA distribution. This follows directly from the loss-gradient pressure on $\alpha$: at any position where the Origin pathway carries information predictive of the target token, increasing $\alpha_i$ reduces $\mathcal{L}_{\text{final}}$, so optimization pushes $\alpha$ toward saturation wherever the Origin signal is even mildly informative. Since the evolutionary prior is broadly informative across antibody sequences — germline identity is a strong prior in FRs, and somatic-hypermutation patterns are predictive in CDRs — the gate is pushed up nearly everywhere, and the complementary $\alpha_i \to 0$ regime is reserved for the rare positions where the AA Head alone is already deterministic (the IMGT-conserved structural anchors discussed in Sec. J.3). Architecturally, this factorization frees the AA Head from memorizing position-specific germline identities: the Origin pathway carries the GL/NGL signal, while the AA Head allocates capacity to sequence-context residue patterns beyond rote AA-identity memorization.

The remainder of this section provides three independent lines of empirical evidence that $\alpha_i$ behaves as predicted by this interpretation and *not* as a confidence decay on the Origin Head.

### J.2. Evidence 1: $\alpha$ Decouples from Origin Head Confidence

If $\alpha_i$ were a confidence decay factor on the Origin predictor, regions where the Origin Head is *inaccurate* should receive a *lower* $\alpha$ since the tempering exponent in Eq. 28 would then down-weight unreliable prior evidence; the data show the opposite pattern (Table S15), with CDR2 exhibiting the lowest Origin accuracy (0.527) yet the second-highest median $\alpha$

...

(0.958), while FR2 — a region where the Origin Head is reasonably accurate (0.810) — carries the *lowest* median $\alpha$ (0.848), and a Pearson test on per-position values yields $r = 0.10$, $p = 0.22$, indicating no detectable association between $\alpha_i$ and Origin accuracy: the gate is therefore not calibrating for Origin reliability but selecting positions where the Origin prior is *useful to the AA head*, regardless of how confidently the Origin Head itself estimates it.

*Table S15.* Per-region Origin Head performance and median $\alpha$ on the heavy chain. $\alpha$ does not track Origin accuracy (Pearson $r = 0.10$, $p = 0.22$); instead, it tracks mutational density (NGL rate; Pearson $r = 0.22$, $p = 0.009$).

| Region | Origin Accuracy | NGL Sensitivity | NGL Rate | Median $\alpha$ |
|---|---|---|---|---|
| FR4 | 0.944 | 0.005 | 0.039 | 0.962 |
| CDR3 | 0.892 | 0.069 | 0.068 | 0.991 |
| FR1 | 0.852 | 0.109 | 0.047 | 0.897 |
| FR2 | 0.810 | 0.191 | 0.079 | 0.848 |
| FR3 | 0.790 | 0.158 | 0.112 | 0.882 |
| CDR1 | 0.628 | 0.384 | 0.218 | 0.937 |
| CDR2 | 0.527 | 0.474 | 0.278 | 0.958 |

### J.3. Evidence 2: $\alpha$ Tracks Functional Mutability — CDR Loops and Conserved Anchors

Figure S9 plots $\alpha_i$ as a per-IMGT-position boxplot for both heavy and light chains, and the spatial pattern recapitulates the canonical antibody architecture almost exactly: CDR loops (red) — the regions subject to intense somatic hypermutation during affinity maturation — consistently show elevated $\alpha$ (i.e., the Origin prior is admitted at near-full strength), while framework regions (blue) that stabilize the Ig fold carry systematically lower $\alpha$ (the Origin factor is correspondingly tempered); quantitatively, $\alpha_i$ correlates positively with the empirical NGL mutation rate across heavy-chain positions ($r = 0.22$, $p = 0.009$), which is the behavior predicted by Eq. 28 in which the AA Head learns to *admit* the Origin prior where mutational evidence carries predictive signal and *downweight* it at structurally constrained sites where the canonical germline amino-acid distribution is already an excellent predictor on its own.

**IMGT-conserved structural anchors: a sharper test.** The CDR-vs-FR contrast is a region-level signal; the IMGT numbering scheme additionally affords a residue-level test, since it pins out a small set of *conserved structural anchors* that recur across virtually every variable domain — the disulfide-bridging cysteines at positions 23 and 104, the conserved tryptophan at position 41 in the heavy chain and the analogous conserved residues in the light chain, the hydrophobic-core anchor near position 89, and the conserved Phe/Trp at position 118 marking the FR4 boundary (Lefranc et al., 2003). At these positions the germline amino-acid identity is essentially deterministic across the human repertoire (e.g., position 23 is almost universally Cys), so the AA Head alone is already a near-perfect predictor and the Origin pathway carries *no incremental information* for the target token. The task-relevance interpretation therefore predicts $\alpha_i \rightarrow 0$ precisely at these anchors, whereas a confidence-decay interpretation predicts the opposite: conserved positions are exactly where the Origin Head is most accurate, so $\alpha_i$ should saturate.

Inspecting Fig. S9 the empirical pattern is unambiguous: on *both* the heavy and light chains, the deepest within-FR $\alpha$ collapses occur at these conserved anchor positions and not at adjacent framework residues, while CDR3 remains $\alpha$-saturated. This is the behavior predicted by the tempered-posterior task-relevance gate, the *opposite* of the confidence-decay prediction, and constitutes a residue-level sanity check that the gate is recovering biologically meaningful structure — the canonical IMGT scaffold-vs-loop architecture — purely from the masked-LM signal, with no IMGT supervision at training time.

### J.4. Evidence 3: Non-Monotonic Training Dynamics

If $\alpha_i$ were a simple global temperature, we would expect it to evolve monotonically — either saturating near 1 (full Origin injection) or decaying toward 0 (AA-only prediction) — as the loss landscape settles, but empirically the dynamics are distinctly non-monotonic and track generalization rather than optimization progress (Fig. S10; Tab. S16).

Three regimes emerge: during *pretraining*, $\alpha$ is modest overall (0.684) but CDR3 is already at 0.958, consistent with CDR3 being the most mutation-dense region even before finetuning has calibrated the Origin Head; at the *best validation checkpoint* (epoch 23), $\alpha$ peaks globally at 0.922, indicating maximal utilization of the factorized Origin/AA decomposition at the point of optimal generalization; during *overfitting* (epochs 53–74), $\alpha$ collapses toward the pretraining baseline as the AA Head increasingly memorizes training sequences and no longer needs the Origin prior, and notably CDR3 retains the highest $\alpha$ even at epoch 74 (0.878 vs. 0.608 overall) — the same region where somatic hypermutation is densest and where

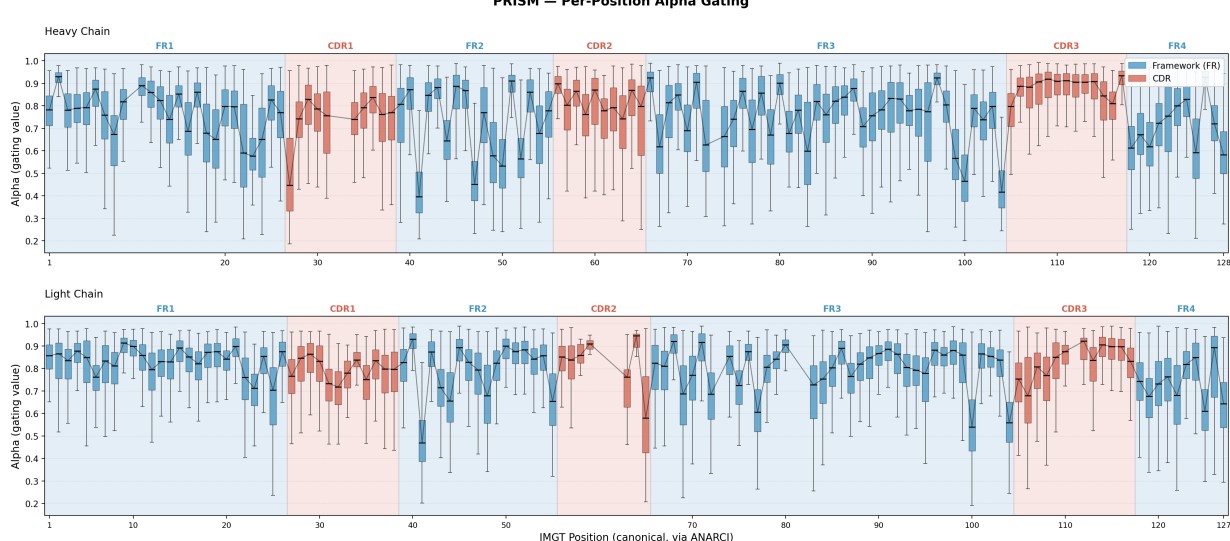

*Figure S9.* **Per-position $\alpha$ recapitulates antibody architecture on both chains.** Boxplot of $\alpha_i$ across IMGT canonical positions (via ANARCI) for the heavy chain (top) and light chain (bottom), colored by IMGT region (CDR, red; FR, blue). CDR loops — particularly heavy-chain CDR3 — saturate near $\alpha \approx 1$, while framework regions are systematically tempered. The deepest $\alpha$ collapses within FRs co-localize with the canonical IMGT-conserved structural anchors (Cys23, Trp41, hydrophobic-89, Cys104, Phe/Trp118), where the germline amino-acid identity is essentially deterministic and the Origin prior carries no incremental information for the AA Head.

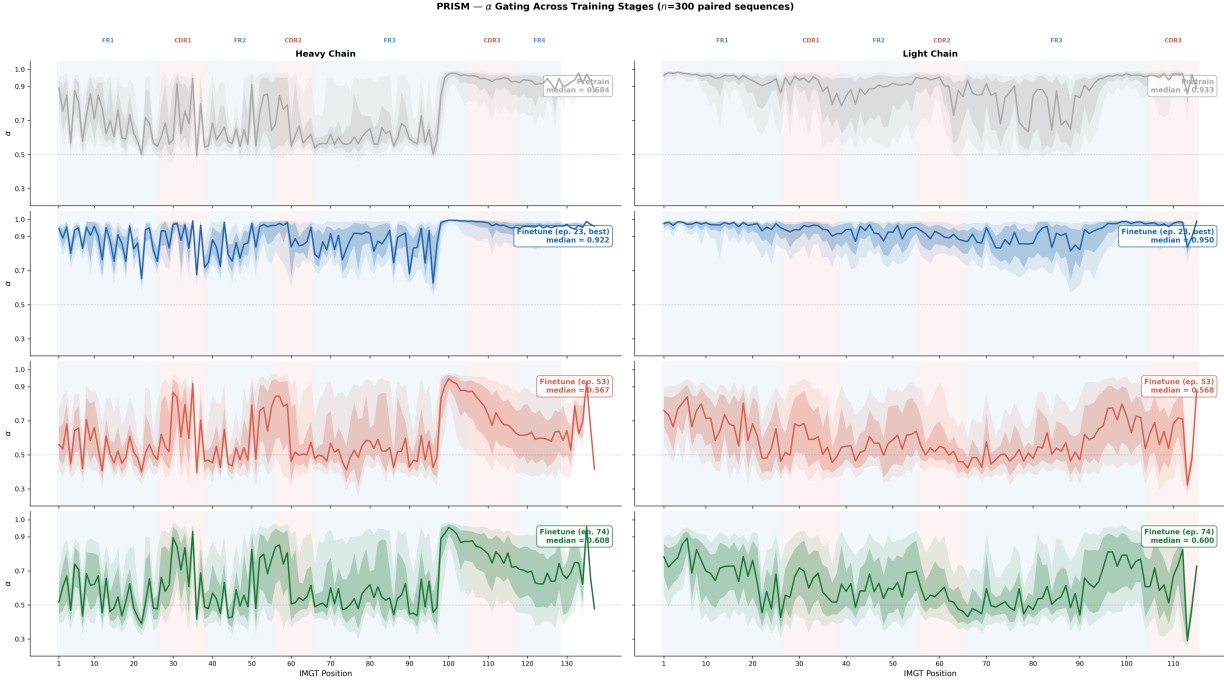

*Figure S10.* $\alpha$ **dynamics are non-monotonic and couple to generalization.** Per-position $\alpha$ profiles at three training stages: pretraining (blue), best validation checkpoint (green), and late finetuning where the model begins to overfit (red). CDR3 preserves high $\alpha$ across all stages, while framework regions are the first to collapse as the AA head memorizes sequences.

memorization is hardest — so this non-monotonic, region-stratified pattern is inconsistent with both a confidence-decay and a temperature interpretation and is the signature we would expect from a learned task-relevance gate.

*Table S16.* Median $\alpha$ across training stages (heavy chain). $\alpha$ peaks at the best-validation checkpoint and collapses during overfitting, with CDR3 showing the most resistance.

| Training Stage | FR1 | FR2 | CDR2 | CDR3 | Overall |
|---|---|---|---|---|---|
| Pretrain (epoch 4) | 0.683 | 0.607 | 0.720 | 0.958 | 0.684 |
| Finetune epoch 23 (best) | 0.894 | 0.848 | 0.960 | 0.991 | **0.922** |
| Finetune epoch 53 | 0.521 | 0.510 | 0.743 | 0.865 | 0.567 |
| Finetune epoch 74 | 0.544 | 0.537 | 0.749 | 0.878 | 0.608 |

## J.5. Summary

Together, Sections J.1–J.4 establish that $\alpha_i$ operationalizes a position-specific tempered posterior whose tempering schedule is *learned from data*: the gate admits Origin evidence where somatic hypermutation is biologically concentrated (CDRs) and suppresses it where germline identity is structurally prescribed (frameworks), and its training dynamics track generalization rather than optimization convergence.

# K. Stop-Gradient Ablation: Stability and Evolutionary Decoupling

A subtle concern with the `stop-gradient` operator applied during conditional injection is that, while it protects the Origin Head from gradient interference originating at the AA Head, it could potentially subject the AA Head itself to *covariate shift* in the early training regime when the Origin estimator is still poorly calibrated: if the Origin Head's output distribution evolves rapidly during the first optimization steps, the AA Head could see a moving conditioning signal and destabilize. This section presents a controlled ablation showing (i) no such instability arises in practice and (ii) the stop-gradient is nevertheless critical — not as a stability device, but as an inductive bias that *prevents the AA Head from delegating NGL prediction to the Origin signal*.

## K.1. Experimental Setup

We compare two training regimes over the first 500 optimization steps, matched in all other respects (same data shard, same initialization, same learning-rate warmup, same batch size of $1{,}024$ sequences via DDP):

- **Learned** $\alpha$ (PRISM default): the $\alpha$-gate modulates the Origin injection term, while `stop-gradient` severs backprop through the Origin logit into the AA pathway.

- **Fixed** $\alpha = 1$ (ablation): the Origin signal is injected at full strength with no modulation and with stop-gradient disabled; this maximally exposes the AA Head to the early-training Origin distribution.

We track (a) the rolling AA-loss curve, (b) rolling-window standard deviation as an instability indicator, (c) the Origin BCE trajectory, and (d) the *NGL perplexity* of the AA Head measured on held-out validation sequences restricted to mutated positions.

## K.2. No Training Instability Is Observed

Figure S11 (A1–B1) shows loss curves for the two regimes that are visually indistinguishable throughout the 500-step window, and rolling loss standard deviations over a 50-step window are essentially identical (0.671 for Learned $\alpha$, 0.672 for Fixed $\alpha$), providing no evidence for covariate-shift-induced oscillations; three mechanisms combine to prevent instability:

1. **Rapid Origin convergence** (Fig. S11 A2): the Origin BCE starts at $\approx 0.089$ (already far below the chance level of $0.69$, because germline position can be partially inferred from the gene embedding alone) and converges to $\approx 0.033$ within the first 200 steps, so the Origin distribution stabilizes faster than the AA Head can adapt to it, physically precluding prolonged covariate shift.

2. **Origin Dropout:** the Origin signal is stochastically masked during training ($p = 0.1$), which implicitly regularizes the AA Head against over-reliance on any particular Origin realization and provides a noise injection path that is empirically known to aid stability.

3. **Standard architectural safeguards:** the Origin logit enters the AA pathway additively after LayerNorm, a learning-rate warmup schedule gently scales gradients during the critical early phase, and a large batch size (1,024, via DDP) averages over any residual noise in the Origin estimate.

These mechanisms are *built into the architecture*, not post-hoc stabilizers, which is why both ablation arms remain smooth.

### K.3. Preventing AA Head Delegation: The True Benefit

The critical difference emerges not in optimization stability but in *what the AA Head learns*: with Fixed $\alpha = 1$ the AA Head can shortcut NGL prediction by reading the Origin signal directly through the injected logit, obviating the need to model mutation preference from sequence context, whereas with Learned $\alpha$ plus stop-gradient the gate forces the AA Head to build its own intrinsic NGL representation before it is allowed to consult Origin evidence.

The effect on validation NGL perplexity is large (Fig. S11 C2): Learned $\alpha$ achieves **2.35**, compared to **4.20** for Fixed $\alpha$ — nearly a $2\times$ gap despite identical data, architecture, and compute — confirming that the stop-gradient is functionally an *inductive bias for evolutionary decoupling* that prevents the AA pathway from outsourcing its representational burden to the Origin pathway, yielding two genuinely independent predictors whose combination at inference time carries complementary information rather than redundant signal.

### K.4. Summary

The stop-gradient serves a dual purpose that is easily conflated but should be distinguished: (i) it protects the Origin Head from gradient contamination by the downstream AA loss, preserving its interpretation as a "pure" evolutionary predictor, and (ii) it forces the AA Head to develop an independent NGL representation, which the learned gate $\alpha$ can then combine with the Origin signal via the tempered-posterior mechanism of Sec. J; both effects are essential to the factorized design of PRISM, and neither effect compromises early-training stability in practice.

## L. Robustness to Noisy GL/NGL Supervision

A natural concern with any label-conditioned framework is that the reported gains may depend on access to unusually clean supervision, in which case the method would not transfer to realistic pipelines where germline/non-germline (GL/NGL) annotations contain systematic errors. This section presents a controlled noise-injection study that addresses two related questions: (i) whether PRISM's advantage persists when GL/NGL supervision is corrupted, and (ii) whether the magnitude of the tolerated noise covers the disagreement observed between current germline-calling pipelines in practice. We find that PRISM retains a substantial performance advantage over all baselines under noise levels that *exceed* the realistic cross-pipeline disagreement rate, and that the method only begins to break down under extreme corruption — evidence that the mechanism is disentangling competing signals rather than overfitting to label artifacts.

### L.1. Calibrating Realistic Noise Levels Against Germline-Caller Disagreement

Modern nucleotide-based germline callers such as IGBLASTN achieve near-perfect V-gene assignments, with a reported mishit frequency of $0.004$ on benchmark repertoires (Ye et al., 2013; Smakaj et al., 2020), so "clean" GL/NGL labels are not a theoretical construct but the status quo on well-curated repertoire datasets. However, the labels a practitioner actually obtains depend on the caller, and switching between pipelines induces a nontrivial disagreement rate: comparing IGBLASTN (nucleotide, closest-V) against the amino-acid-based caller ANARCI (Dunbar & Deane, 2016) on our data yields a $13.9\%$ position-level disagreement rate, which we treat as the empirical ceiling for *realistic* cross-pipeline noise.

To probe this regime and beyond, we fine-tune PRISM with GL/NGL labels that are stochastically flipped during training: at every epoch we independently flip $k$ labels per chain, and we instantiate $k \in \{2, 4\}$. For chains of average mutated-position count this corresponds to effective disagreement rates of approximately $15\%$ (**+Noise2**) and $30\%$ (**+Noise4**), so **+Noise2** approximates the realistic cross-pipeline disagreement ceiling and **+Noise4** probes well beyond it. All other training hyperparameters — optimizer, learning rate, batch size, finetuning horizon — are held fixed, and we evaluate the resulting models on the full suite of metrics used in the main paper. As a complementary validation of this noise model, we additionally train a PRISM variant with labels derived directly from the ANARCI caller, confirming that the controlled-flipping analysis faithfully reproduces the empirical cross-pipeline noise profile.

**Stop-Gradient Stability Ablation: Fixed $\alpha$=1.0 vs Learned $\alpha$ (500 steps)**

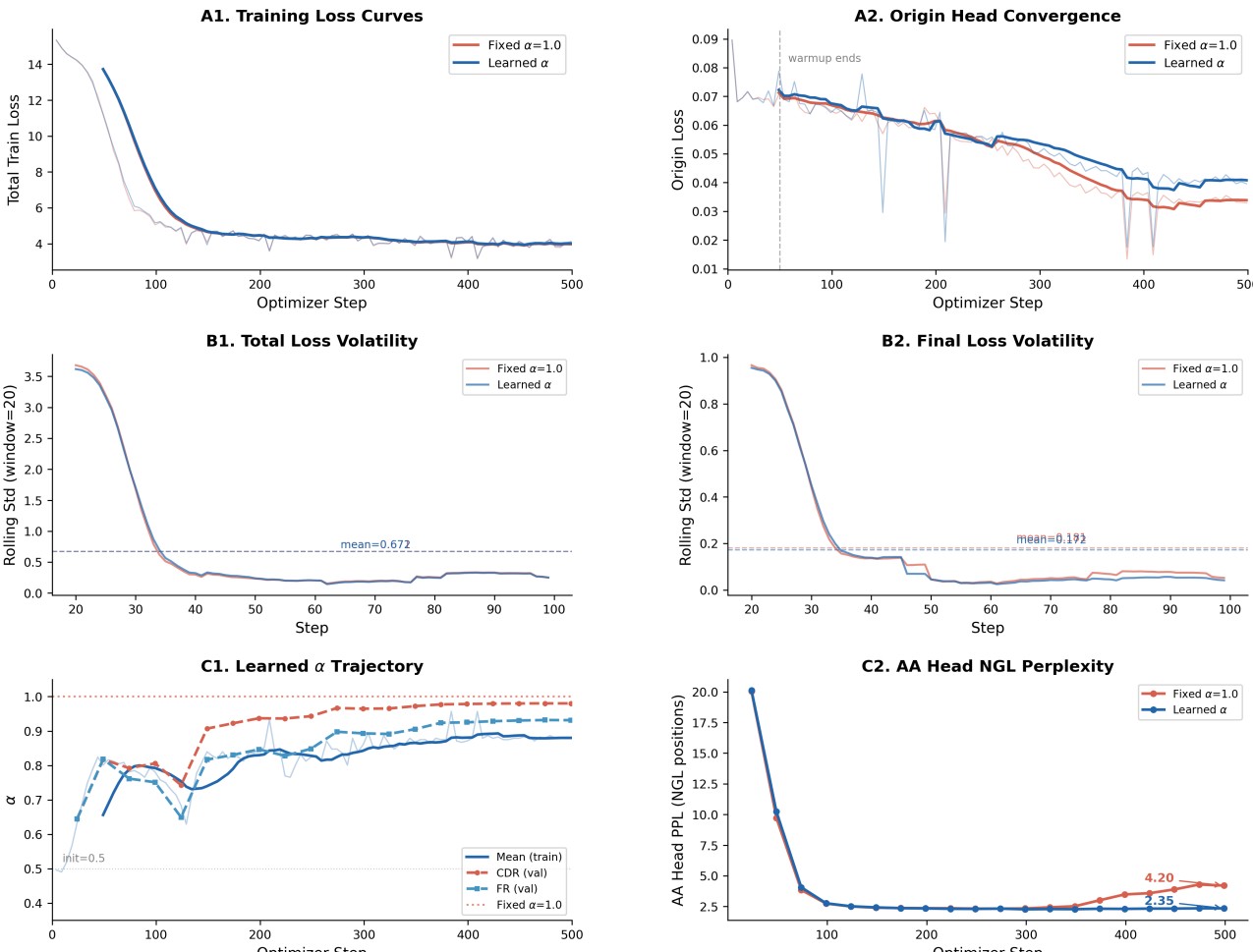

*Figure S11.* **Stop-gradient ablation over the first 500 training steps.** Panels A1/B1: AA-loss trajectories for Learned $\alpha$ (default) vs. Fixed $\alpha = 1$, showing indistinguishable stability. Panel A2: Origin BCE converges rapidly from 0.089 to 0.033, limiting the window during which covariate shift could harm the AA Head. Panel C2: Validation NGL perplexity after convergence — Learned $\alpha$ achieves 2.35 vs. 4.20 for Fixed $\alpha$, demonstrating that the stop-gradient is critical for preventing the AA Head from delegating NGL prediction to the Origin signal.

## L.2. Pseudo-Perplexity at Mutated Positions

Pseudo-perplexity at CDR3 NGL positions is the most direct probe of whether the Origin pathway has collapsed under noise: if corrupted labels destroyed the ability to model somatic hypermutation, we would expect PRISM's CDR3-NGL PPL to regress toward the baseline ALM performance. Instead (Fig. S12), even under the extreme **+Noise4** condition PRISM still outperforms the strongest baseline AbLang2 on CDR3 NGL PPL (heavy: $9.84$ vs. $10.65$; light: $6.79$ vs. $7.38$), and the degradation at realistic noise levels is marginal.

## L.3. GL/NGL Discrimination under Label Corruption

A second axis is the discriminative quality of the Origin Head itself, which we measure as the PR-AUC for classifying held-out positions as GL vs. NGL (Fig. S13). Clean PRISM achieves PR-AUC $= 0.980$, and injecting noise reduces this only modestly to $0.880$ under **+Noise2** and $0.879$ under **+Noise4**; both values remain substantially above the specialized

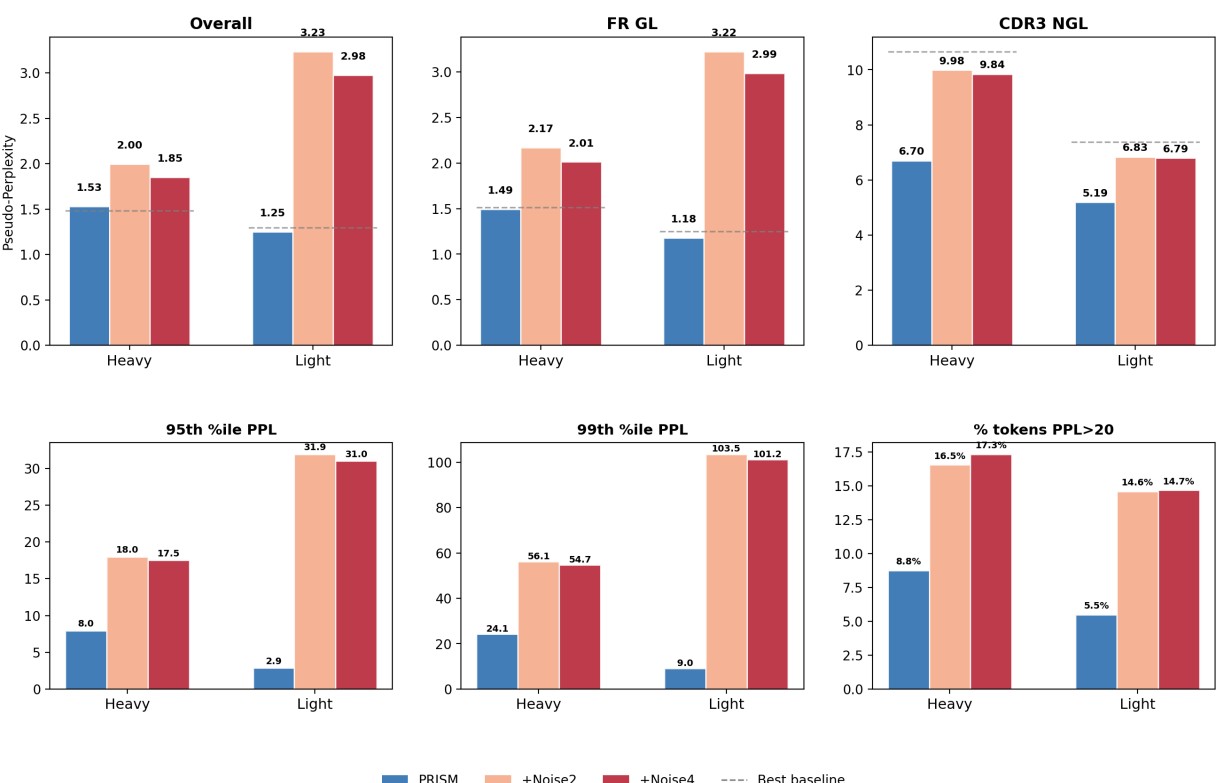

*Figure S12.* **Pseudo-perplexity at CDR3 NGL positions under GL/NGL label noise.** PRISM retains its advantage over AbLang2 and all other ALM baselines even when ∼30% of GL/NGL labels are corrupted at each epoch (**+Noise4**). Heavy-chain PPL: PRISM 9.84 (+Noise4) vs. AbLang2 10.65; light-chain: 6.79 vs. 7.38.

baseline AntiBERTy (0.785). The near-identical PR-AUC at **+Noise2** and **+Noise4** is informative: it indicates that the Origin Head's representation saturates rather than degrading linearly with noise, consistent with the AA Head providing a regularizing signal through the gated Origin pathway (Sec. J).

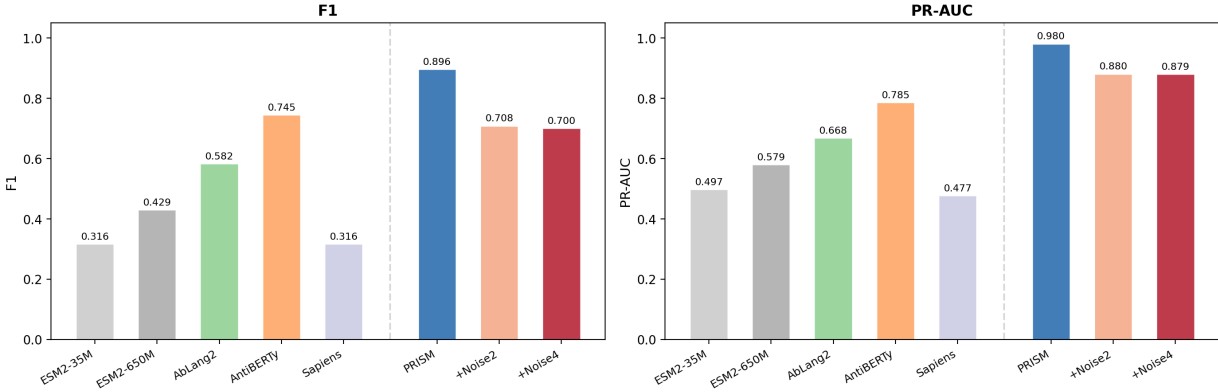

*Figure S13.* **GL/NGL discrimination (PR-AUC) under label noise.** Clean PRISM: 0.980; **+Noise2**: 0.880; **+Noise4**: 0.879. All variants remain well above the strongest ALM baseline (AntiBERTy, 0.785).

## L.4. Binding-Affinity Prediction

Zero-shot binding correlations on the three DMS benchmarks test whether the *downstream* utility of PRISM's scores is preserved under noisy training labels (Fig. S14). At realistic noise (**+Noise2**), PRISM maintains positive Spearman $\rho$ across all three targets (G6.31: 0.052; CR9114: 0.236; Trastuzumab: 0.357), which is on par with the clean-label PRISM numbers reported in the main paper. At **+Noise4**, performance degrades on the most challenging dataset (CR9114 loses its positive correlation), while G6.31 and Trastuzumab continue to exhibit signal. The asymmetry of this breakdown — CR9114 first, G6.31 and Trastuzumab preserved — tracks the relative "information budget" each dataset places on the Origin signal: CR9114 is a broadly neutralizing antibody whose affinity landscape is most sensitive to NGL-specific modeling, so corruption in those labels disproportionately affects it.

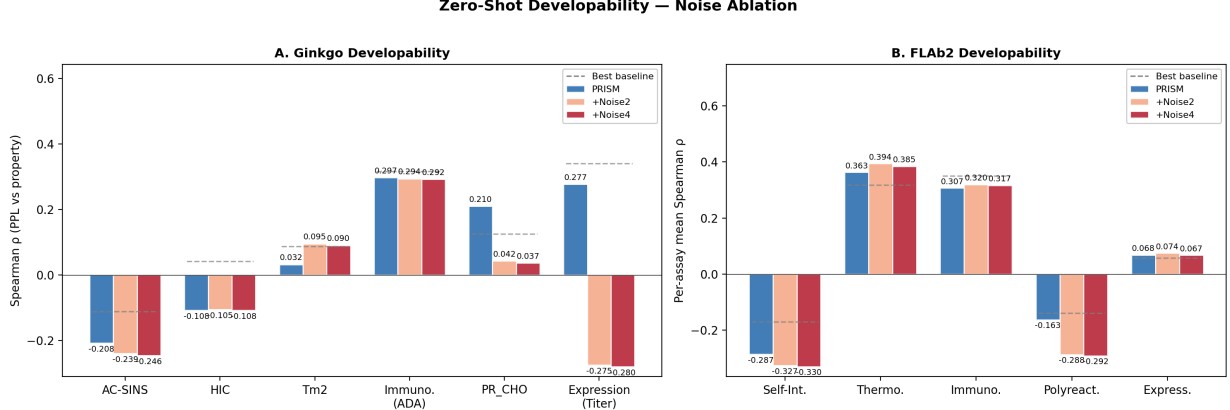

*Figure S14.* **Binding affinity prediction (Spearman $\rho$) under label noise.** PRISM at realistic noise (**+Noise2**) retains positive signal on G6.31 (0.052), CR9114 (0.236), and Trastuzumab (0.357). Breakdown is isolated to CR9114 only under extreme noise (**+Noise4**).

## L.5. Developability

Developability metrics probe whether label corruption degrades generalization to properties that are *not* directly supervised during training (Fig. S15). ADA (aggregation-prone-residue score) remains essentially unchanged across the three regimes ($0.310 \rightarrow 0.294 \rightarrow 0.292$ for Clean $\rightarrow$ **+Noise2** $\rightarrow$ **+Noise4**). On the FLAb2 benchmarks, **+Noise2** *exceeds* the best ALM baseline on thermostability (0.385 vs. 0.317) and self-interaction (0.354 vs. 0.288), reinforcing the interpretation that mild label noise does not merely fail to harm the model but can act as a weak regularizer against overfitting to specific GL/NGL boundary cases.

*Figure S15.* **Developability prediction under label noise.** ADA remains stable ($0.310 \rightarrow 0.294 \rightarrow 0.292$). On FLAb2, **+Noise2** outperforms the strongest baseline on thermostability (0.385 vs. 0.317) and self-interaction (0.354 vs. 0.288).

### L.6. Summary and Implications

Across four independent evaluation axes — pseudo-perplexity, GL/NGL discrimination, binding, and developability — PRISM remains the strongest model at the realistic cross-pipeline noise level (**+Noise2**, $\sim 15\%$ disagreement), and on three of the four axes it continues to outperform the best baseline even at **+Noise4** ($\sim 30\%$ disagreement), which is roughly *twice* the IGBLASTN/ANARCI disagreement rate observed in our data. The breakdown at **+Noise4** is concentrated on the single most label-sensitive DMS target (CR9114), consistent with graceful degradation rather than catastrophic collapse.

Two broader implications follow. First, because the performance ordering with respect to baselines is preserved under noise, the gains reported in the main paper are not an artifact of clean-label supervision and should be expected to transfer to practical pipelines where germline-caller disagreement is the dominant source of label noise. Second, the robustness profile suggests that PRISM's framework is applicable in settings where the binary supervision target is *less* certain than GL/NGL — for example, isotype assignment, lineage boundaries, or subtype annotations in other repertoires — where the labels are inherently noisier but may still carry enough signal to drive the factorized decomposition.

## M. NGL Residues Are Not Equivalent to CDRs: Necessity of Residue-Level Origin Labels

A natural simplification of the Origin Head's supervision signal would be to replace per-residue GL/NGL labels with a region-level heuristic — e.g., treating every framework (FR) position as germline and every complementarity-determining-region (CDR) position as non-germline — since CDRs are the canonical site of somatic hypermutation in the antibody architecture. This section quantifies the overlap between NGL residues and CDR regions on a population of 63M training sequences and demonstrates, through a controlled head-to-head against a region-forced oracle on a held-out 22K-sequence test set, that the residue-level labels used by PRISM are strictly necessary: the region-based proxy introduces systematic mislabeling that propagates into a measurable degradation of the generative model even when the replacement label happens to be locally correct.

### M.1. Overlap Between NGL Residues and CDR Regions

We first characterize the population-level overlap between the two partitions across the 63M-sequence training set (Fig. S16a), and two observations establish that NGL and CDR are only partially aligned:

- **CDRs are predominantly germline:** $85.5\%$ of CDR residues are in fact germline (GL), reflecting that CDR positions remain anchored to the V-gene reference except at the subset of sites where somatic hypermutation has deposited a substitution; consequently, treating all CDR residues as NGL mislabels the dominant majority.

- **FRs host non-trivial mutation:** $8.8\%$ of FR residues are NGL, so framework regions are not mutation-free; somatic hypermutation is depleted in the frameworks but not absent, and treating all FR residues as GL forecloses modeling of this tail.

A binary map of $\mathrm{FR} = \mathrm{GL}$ and $\mathrm{CDR} = \mathrm{NGL}$ therefore mislabels a substantial fraction of residues, establishing a prior expectation that any model supervised by this heuristic would inherit a structural bias rather than recovering the true per-residue origin distribution.

### M.2. Controlled Comparison Against a Region-Forced Oracle

We directly evaluate the consequences of this mislabeling on a held-out 22K-sequence test set (Fig. S16b,c) by comparing PRISM's per-residue Origin predictions against a region-forced oracle that assigns labels purely from IMGT region identity, decomposing the effect by whether the forced label happens to agree with the ground-truth residue label:

- **Misassigned residues (CDR-GL and FR-NGL).** For residues whose true label disagrees with the regional assignment — i.e., CDR positions that are actually germline and FR positions that are actually non-germline — forcing the regional label catastrophically degrades sequence likelihood, with pseudo-perplexity rising to $8.7$ on the heavy chain and $6.9$ on the light chain; this is the regime where the heuristic is factually wrong about the residue, and the model pays the full cost of that error.

- **Agreeing residues (FR-GL and CDR-NGL proxy).** More subtly, even residues where the region-based label happens to match the true label still incur a pseudo-perplexity penalty of $1.1$ to $3.8$ relative to PRISM's learned per-residue

predictions; this non-zero cost indicates that local mislabeling at *other* positions in the same sequence propagates globally through the token-by-token likelihood, i.e., the model cannot recover a clean posterior over a given residue when neighboring positions are mis-conditioned.

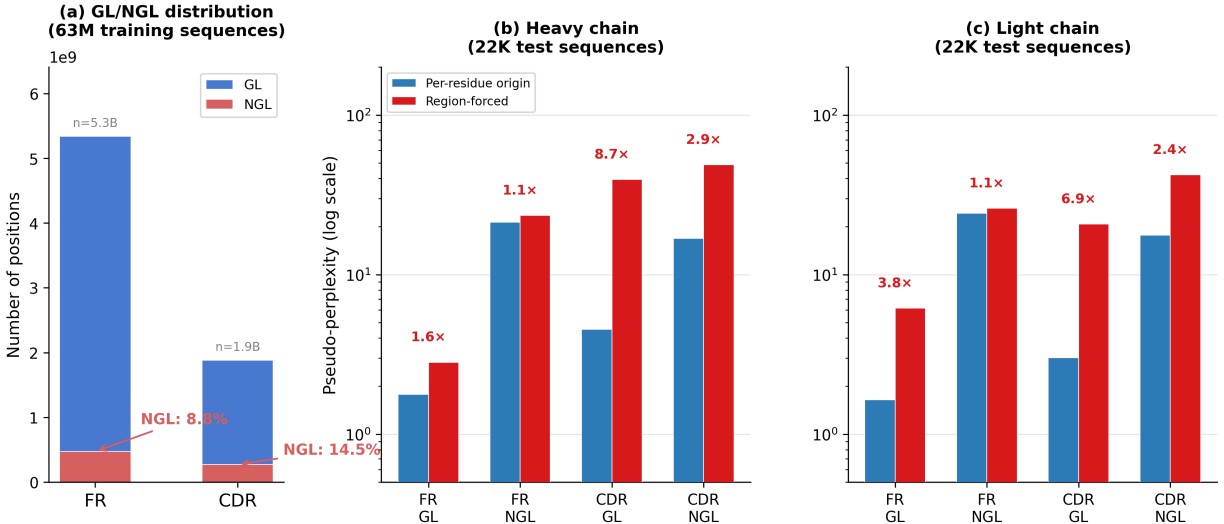

*Figure S16.* **NGL residues are not equivalent to CDRs.** **(a)** Population-level overlap on 63M training sequences: $85.5\%$ of CDR residues are germline and $8.8\%$ of FR residues are non-germline, so the binary heuristic FR = GL, CDR = NGL mislabels a non-trivial fraction of positions. **(b,c)** Pseudo-perplexity on a 22K held-out test set for PRISM (per-residue labels) versus a region-forced oracle, stratified by whether the forced label agrees with ground truth: misassigned residues (CDR-GL and FR-NGL) degrade PPL to 8.7 (heavy) / 6.9 (light), while even residues with "correct" regional labels incur a 1.1–3.8 PPL penalty, reflecting the global propagation of local label errors through the sequence likelihood.

## M.3. Implications

Two conclusions follow. First, residue-level GL/NGL supervision is not substitutable with a region-level proxy: the overlap between NGL and CDR is too low for a regional heuristic to recover the signal PRISM exploits, and the asymmetric mislabeling — most CDR residues are germline, yet non-trivial FR residues are non-germline — implies that neither direction of the heuristic can be rescued by a symmetry-breaking correction such as thresholding or a single learned region-to-origin probability. Second, the fact that PRISM's predictions cannot be matched even on residues where the regional proxy would be locally correct indicates that the Origin Head is learning a genuinely position-specific posterior rather than a coarse region classifier, consistent with the tempered-posterior analysis of Sec. J in which $\alpha_i$ modulates the Origin prior per-residue rather than per-region; both observations reinforce the architectural choice of training the Origin Head against residue-level germline labels rather than against a region-derived proxy.

## N. Robust Generalization to Therapeutic Benchmarks

While PRISM is trained on natural antibody repertoires (OAS), therapeutic antibodies often undergo extensive engineering that deviates from natural evolutionary distributions. To demonstrate PRISM's practical utility for real-world drug design, we evaluate its performance on Thera-SAbDab (Raybould et al., 2020), a database of clinical-stage therapeutics. This benchmark serves as a rigorous test of whether the model can generalize to the precise engineering trade-offs between developability and high-affinity binding found in successful drugs.

### N.1. Recapitulating Clinical Design Rules

To ensure a rigorous zero-shot evaluation, we curated a test set from Thera-SAbDab, strictly excluding any sequence sharing $> 95\%$ identity with our training set. We assessed the model's ability to reconstruct these therapeutic sequences under random masking, utilizing this reconstruction accuracy as a proxy for its internal understanding of antibody design rules (Figure S17A).

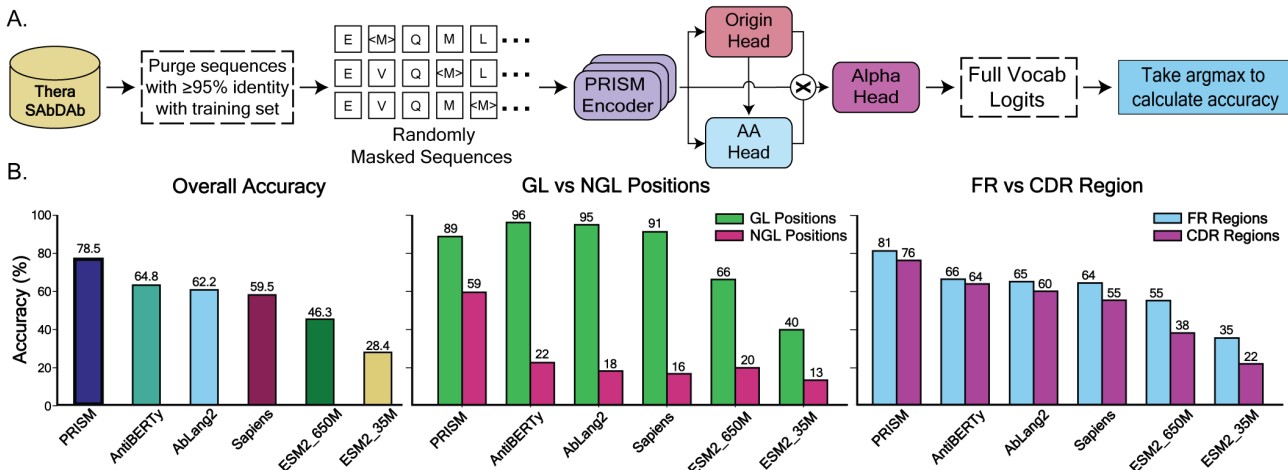

*Figure S17.* **Generative performance on therapeutic benchmarks.** **(A)** Schematic of the evaluation workflow using masked sequence reconstruction on Thera-SabDab. **(B)** Overall accuracy comparison shows PRISM (78.5%) significantly outperforming baselines. **(C)** Performance on GL vs. NGL positions reveals that PRISM achieves 59% accuracy on NGLs, nearly triple the nearest baseline. **(D)** Region-wise analysis demonstrates robust modeling across both stable Frameworks (FR) and hypervariable CDRs.

Despite this strict separation, PRISM achieves a state-of-the-art reconstruction accuracy of 78.5%, significantly outperforming specialized baselines such as AntiBERTy (64.8%) and AbLang2 (62.2%) (Figure S17B). This suggests that PRISM successfully captures the "design grammar" of therapeutics. In Framework (FR) regions (Figure S17D), which require to have developability, PRISM maintains high accuracy (81%) comparable to germline-focused baselines. Crucially, PRISM retains the best performance (76%) in the hypervariable CDR regions, where therapeutic specificity is often engineered. In contrast, baseline performance significantly degrades (e.g., ESM2-650M drops to 38%), confirming the benefits of PRISM's explicit separation.

### N.2. Adaptive Assignment to Evolutionary Subspaces

To evaluate the handling of non-germline variants, we decomposed accuracy by origin (Figure S17C), confirming that PRISM correctly maps conserved sites to the GL vocabulary and functional deviations to the NGL vocabulary.

This synergistic alignment mitigates the confusion often seen in standard models. While baselines exhibit a sharp dichotomy—recovering GL positions well but failing catastrophically at NGL positions —PRISM leverages its dual-token space to bridge this gap. By correctly identifying the evolutionary context, PRISM achieves 59% accuracy at NGL positions—nearly triple the nearest baseline. This demonstrates that PRISM treats NGL mutations as deliberate features rather than statistical noise.

## O. Germline-Centered Likelihood Scoring for Affinity-Matured Antibodies

### O.1. Germline-Centered Scoring on Affinity-Matured Antibodies

**Motivation.** The G6.31 single-mutation benchmark of (Koenig et al., 2017a) is a particularly demanding zero-shot scoring scenario. G6.31 is a strongly affinity-matured antibody ($K_d \approx 0.4$ nM), obtained through multiple rounds of phage display and targeted mutagenesis (Lee et al., 2004; Fuh et al., 2006), and the dataset probes single substitutions around a sequence that already sits close to a local optimum on the binding-affinity landscape. In this regime, the overwhelming majority of mutations are neutral or mildly deleterious, producing a weak signal-to-noise ratio that is inherently challenging for *any* sequence model relying on *wildtype-centered* likelihood scoring—i.e., scoring a variant by conditioning on the matured wildtype context. We therefore do not interpret the relatively modest correlation of wildtype-centered PRISM on G6.31 as a fundamental architectural limitation of the model with respect to single-residue scans or local affinity maturation; rather, it reflects a generic difficulty of mutant-centered scoring once the reference sequence is already close to optimal.

**Germline-centered scoring.** PRISM's factorized vocabulary explicitly separates germline (GL) and non-germline (NGL) tokens, which enables a principled alternative to wildtype-centered scoring. Instead of evaluating a variant against the

(already-matured) wildtype, we evaluate it against the *inferred germline reference* of the parent antibody and isolate the NGL likelihood component. Intuitively, this anchors the scoring frame at the pre-affinity-maturation ancestor, restoring a meaningful evolutionary reference against which single mutations can be discriminated. While any masked language model could in principle be queried against a substituted germline context, PRISM's disentangled architecture is what makes the resulting score clean: the explicit NGL head exposes the non-germline likelihood directly, without contamination from germline-conserved positions. Importantly, germline-centered scoring does not require any external germline assignment tool, since PRISM's Origin Head recovers germline sequences accurately on OAS (Appendix **??**).

**Results.** Compared to wildtype-centered scoring, germline-centered scoring yields modest but consistent improvements in Spearman correlation across all three antibody deep mutational scanning benchmarks. On the highly affinity-matured G6.31, Spearman $\rho$ improves from $+0.158$ to $+0.175$; on CR9114, from $+0.391$ to $+0.421$; and on Trastuzumab, from $+0.327$ to $+0.368$. The improvement in the low-signal single-mutation regime of G6.31 does not come at the expense of the multi-mutation benchmarks, where germline-centered scoring also performs slightly better. We present germline-centered scoring as a complementary scoring option made available by PRISM's disentangled architecture, which practitioners may find useful in low-signal regimes such as fine-grained single-mutation scans of strongly affinity-matured antibodies.

