# OpenReview forum: "Explicit representation of germline and non-germline residues improves antibody language modeling"
_ICML.cc/2026/Conference — ICML 2026 regular_

### Official Review · Reviewer_xKEm · 2026-03-12

**Soundness:** 3
**Presentation:** 3
**Significance:** 3
**Originality:** 3
**Overall Recommendation:** 4
**Confidence:** 2

**Summary:**

The paper proposes PRISM, a germline-aware antibody language model that explicitly distinguishes germline (GL) versus non-germline (NGL) residues using separate token types and a sequential conditional architecture (Origin/AA/Alpha heads). The core claim is that this representational disentanglement reduces “germline bias,” improves modeling of rare but functionally important mutations (especially in CDRs), and enables more controllable antibody sequence generation while preserving framework stability. Empirically, the authors report improved pseudo-perplexity—particularly on CDR3/NGL positions—plus stronger zero-shot affinity correlations and competitive developability-related signals, supported by probing, therapeutic reconstruction, and ablation analyses.

**Compliance With Llm Reviewing Policy:**

Affirmed.

**Key Questions For Authors:**

please see weakness.

**Limitations:**

yes

**Strengths And Weaknesses:**

Strengths

1. Simple, interesting, and effective idea: Explicitly encoding GL vs. NGL as different token types is a clean design choice that aligns well with the biology and directly targets germline bias. I like this style of work.

2. Clear architectural rationale: The sequential conditional setup (Origin/Alpha gating the AA identity predictions) provides an intuitive mechanism for disentanglement and controllability.

Weaknesses

1. While zero-shot affinity and reconstruction are persuasive, additional downstream design tasks (e.g., controlled multi-objective optimization or prospective validation) would better demonstrate practical utility.

2. The approach relies on accurate GL/NGL labeling and consistent germline templates; sensitivity to annotation errors or alternative germline callers is not fully explored.

---

> ### Author Rebuttal · Authors · 2026-03-30
>
> We sincerely thank the reviewer for the constructive feedback. For all supporting figures/tables, please refer to: [link](https://anonymous.4open.science/r/PRISM_Rebuttal-13CC/README.md)
>
> **W1: Lack of downstream design tasks.** \
> We generated variants of G6.31, Trastuzumab, CR9114 antibodies and evaluated their binding and developability. PRISM enables controllable generation via GL/NGL modulation: NGL-biased decoding improves affinity, GL-biased decoding improves developability, and unconstrained decoding balances both. We will add Pareto front analyses and detail the generation protocol, metric conventions, and evaluation pipelines in the Methods and Appendix.
>
> **Pseudo-log likelihood-guided sequence generation:** Mutation positions were selected as the 10 sites with the lowest masked probability of the wild-type amino acid, and substitutions were sampled directly from model logits. PRISM sequences were generated under three conditions: **PRISM-GL** (GL-biased), **PRISM-NGL** (NGL-biased), and **PRISM-Full** (unconstrained).
>
> **Evaluation Metrics**
> - **Sequence-based Binding ($\Delta$Binding):** Predicted using ridge regression models trained on mean-pooled AntiBERTy embeddings from DMS data for each target (**Figure R15**).
> - **Structure-based Binding ($-\Delta\Delta G_{\text{interface}}$):** Estimated with Rosetta on antibody--antigen complex structures.
> - **Stability ($-\Delta\Delta G_{\text{fold}}$):** Estimated with Rosetta on isolated antibody structures.
> - **Solubility ($\Delta$CamSol):** Computed using the CamSol method.
>
> **Analysis of Controllable Generation:** We evaluated using a win rate metric (percentage of generated sequences with a higher predicted function than the target) (**Figures R16--18**):
>
> - **Affinity Specialization (PRISM-NGL):** NGL-biased decoding consistently achieves higher binding win rates than PRISM-GL across all targets (e.g., G6.31: 46% vs. 35% structural, 40% vs. 15% ML-based), confirming effective affinity specialization.
> - **Developability Specialization (PRISM-GL):** GL-biased decoding achieves substantially higher solubility win rates than PRISM-NGL (e.g., G6.31: 88% vs. 27%). No clear stability trend was observed, likely due to the small mutational load (2--3 mutations).
> - **Balanced Optimization (PRISM-Full):** Unconstrained decoding achieves competitive binding and developability win rates simultaneously, indicating well-calibrated representations that balance functional gains with biophysical constraints.
> - **Comparison to Baselines:** Under an identical decoding budget, PRISM variants achieved competitive or superior win rates and Pareto frontiers compared to AbLang2, ESM2, and IgLM, while PRISM uniquely enables navigation of affinity--developability trade-offs via GL/NGL modulation.
>
> &nbsp;
>
> **W2: What is the sensitivity to germline annotation errors?** \
> Modern germline callers such as IgBLASTn achieve near-perfect assignments with a mishit frequency of 0.004 (Ye et al., 2013; Smakaj et al., 2020). We evaluate sensitivity by injecting label noise matching the disagreement between IgBLASTn and the amino acid-based caller ANARCI (13.9%). We flip 2 or 4 GL/NGL labels per chain per epoch during fine-tuning (+Noise2, +Noise4), corresponding to ~15% and ~30% disagreement. +Noise2 thus approximates realistic cross-pipeline variation. PRISM shows robust performance at realistic noise levels and degrades only under extreme corruption:
>
> - **Pseudo-Perplexity:** At CDR3 NGL positions, even +Noise4 outperforms the best baseline (PRISM: 9.84/6.79 vs. AbLang2: 10.65/7.38; heavy/light) (**Figure R1**).
> - **GL/NGL Discrimination:** PR-AUC decreases modestly from 0.980 (clean) to 0.880 (+Noise2) and 0.879 (+Noise4), remaining well above AntiBERTy (0.785) (**Figure R2**).
> - **Binding:** +Noise2 maintains positive $\rho$ across targets (G6.31: 0.052, CR9114: 0.236, Trastuzumab: 0.357), while +Noise4 causes breakdown (CR9114 $\rho = -0.362$) (**Figure R3**).
> - **Developability:** ADA remains stable (0.310 $\rightarrow$ 0.294 $\rightarrow$ 0.292). On FLAb2, +Noise2 exceeds baselines for thermostability (0.385 vs. 0.317) and self-interaction (0.354 vs. 0.288) (**Figure R4**).
>
> We are also training a PRISM variant with ANARCI-derived labels as a more direct empirical validation for the camera-ready version.

---

### Official Review · Reviewer_DViF · 2026-03-13

**Soundness:** 3
**Presentation:** 3
**Significance:** 3
**Originality:** 2
**Overall Recommendation:** 4
**Confidence:** 4

**Summary:**

This manuscript presents PRISM, a generative framework designed to address the issue of “germline bias” in antibody language models (ALMs). The authors introduce an innovative approach by expanding the vocabulary to 53 tokens, effectively decoupling the evolutionary origin of amino acids (Germline, GL vs. Non-germline, NGL) from their physicochemical identity at the representational level. Built upon the ESM-2 architecture, PRISM employs a sequence-gating mechanism involving three specialized heads—Origin, AA, and Alpha—to achieve precise modeling of antibody hypervariable regions. The model demonstrates superior performance in zero-shot affinity prediction and developability assessment compared to existing general-purpose and specialized protein language models.

**Compliance With Llm Reviewing Policy:**

Affirmed.

**Final Justification:**

This study transforms deep biological priors into strict architectural constraints, successfully overcoming the “germline bias” that currently affects antibody language models. The authors have demonstrated that their framework is highly robust in handling both complex higher-order epistatic multi-site combinations and local single-point affinity maturation. Given the methodological novelty of this work, its comprehensive zero-shot evaluation, and the authors’ responses throughout the review process, I will maintain my score at Weak Accept.

**Key Questions For Authors:**

See Weaknesses.

**Limitations:**

Yes.

**Strengths And Weaknesses:**

**Strengths:**

* The authors demonstrate a profound understanding of V(D)J recombination and somatic hypermutation (SHM), successfully translating these biological mechanisms into optimizable architectural constraints. Explicitly decoupling evolutionary conservation from functional variation via a separated vocabulary represents a significant paradigm shift from current implicit language modeling approaches.
* The paper provides compelling empirical evidence through linear probing and UMAP dimensionality reduction. PRISM’s Origin Head achieves exceptionally high GL/NGL separability (PR-AUC of 0.980) under frozen representations, fundamentally validating the architecture’s ability to overcome the feature entanglement prevalent in traditional models.
* Beyond standard pseudo-perplexity tests, the model is evaluated against the Thera-SAbDab therapeutic antibody benchmark, Deep Mutational Scanning (DMS) affinity datasets (e.g., CR9114, G6.31), and multi-dimensional developability metrics. This rigorous testing comprehensively demonstrates the superiority of decoupled representations in real-world drug design scenarios.

**Weaknesses:**
* The core machine learning contribution relies on vocabulary expansion and sequence-level gating. It remains unclear whether this “template-first and functional-bias decoupling” framework can generalize to other sequence generation tasks with strong hierarchical structures, such as scaffold-based small molecule generation or other protein families with highly conserved domains.
* In the unnormalized log-odds calculation (Equation 6), the numerical dynamics of linearly combining logits and log-probabilities require a more in-depth discussion. Does α_i serve as a confidence decay factor in a Bayesian update context, or is it merely an empirical scaling temperature? The manuscript would benefit from including distribution plots of the Alpha-gating values across the early, middle, and late stages of training.
* The authors utilize a Stop-Gradient operator during conditional injection to maintain the “evolutionary purity” of the Origin Head. However, it is worth investigating whether disconnecting the gradient flow subjects the AA Head to volatile conditional inputs (covariate shift) during the early stages of training. Have any instabilities been observed during initial training phases?
* While PRISM successfully shifts the Spearman correlation for G6.31 from negative to positive, the absolute value remains very weak (ρ≈0.085). The text should more objectively discuss the actual “enrichment efficiency” such a low positive correlation might provide in high-throughput physical screening (e.g., phage display) to avoid overstating its practical value for therapeutic guidance.

---

> ### Author Rebuttal · Authors · 2026-03-30
>
> We sincerely thank the reviewer for the constructive feedback. For all supporting figures/tables, please refer to: [link](https://anonymous.4open.science/r/PRISM_Rebuttal-13CC/README.md)
>
> **W1: Can the decoupling framework generalize beyond antibodies (e.g., small molecules, other conserved protein families)?**
>
> We agree that our paradigm could extend to any sequence generation task with a strong hierarchical dichotomy (e.g., conserved vs. active sites in enzymes, core-scaffolds vs. R-group decorations in small molecules). However, cross-modal validation would require entirely domain-specific construction and evaluation, falling outside the scope of this manuscript. Prompted by your comment, we have expanded the Discussion to outline how PRISM's gating architecture generalizes to other hierarchical generative tasks.
>
> &nbsp;
>
> **W2: Is $\alpha$ a Bayesian confidence decay or an empirical temperature? Include distribution plots across training stages.**
>
> We mapped the $\alpha$ gating mechanism to a formal Bayesian framework, supported by new distribution plots (**Figures R12-13 and Tables R1-2**).
>
> **Mathematical Framework:** The alpha-gated combination (Eq. 6) corresponds to:
> $$P(\text{token}_i \mid \text{pos}) \propto P(\text{AA}_i)\cdot P(\text{origin} \mid \text{pos})^{\alpha(\text{pos})}.$$
> This is isomorphic to a **tempered posterior** (Grünwald, 2012; Holmes & Walker, 2017). Since $\alpha$ receives gradients only through the final prediction loss, it functions as a learned **Task-Relevance Gate**, not a confidence decay or simple temperature.
>
> **Empirical Evidence:**
> * **Uncorrelated with Origin Head Accuracy:** CDR2 has the lowest origin accuracy (0.527) but the second-highest median $\alpha$ (0.958); Pearson $r = 0.10$, $p = 0.22$.
> * **Correlated with SHM Density:** $\alpha$ positively correlates with NGL mutation rate ($r = 0.22$, $p = 0.009$); maximized in CDR loops, minimized in stable interfaces (**Figure R12**).
> * **Non-Monotonic Training Dynamics (Figure R13):** (1) Pretraining: median $\alpha = 0.684$, CDR3 already 0.958. (2) Best checkpoint: spikes to 0.922. (3) Overfitting: collapses to 0.567 as AA head memorizes sequences; CDR3 resists longest.
>
> We will include this Bayesian interpretation and **Figures R12-13, Tables R1-2** in a new Appendix.
>
> &nbsp;
>
> **W3: Does the Stop-Gradient cause covariate shift or training instability in early phases?**
>
> We conducted a controlled ablation comparing Learned $\alpha$ against Fixed $\alpha = 1.0$ (full, unmodulated origin injection) over the first 500 steps.
>
> **1. No Training Instability:** Rolling loss standard deviations were virtually identical (0.671 vs. 0.672), with perfectly overlapping loss curves (**Figure R14, Panels A1 and B1**). We attribute this stability to: (i) **Rapid Origin Convergence (Figure R14, Panel A2):** initial BCE is already ~0.09 (vs. random 0.69), converging to ~0.03, physically preventing prolonged covariate shift; (ii) Origin Dropout ($p = 0.1$) as stochastic noise injection; and (iii) standard architectural safeguards (additive conditioning, LR warmup, large batch size 1,024 via DDP).
>
> **2. The True Benefit-Preventing "Delegation":** Without the learned gate (Fixed $\alpha = 1.0$), the AA Head delegates NGL predictions to the origin signal, yielding a poor AA NGL Perplexity of 4.20. The Learned $\alpha$ forces the AA Head to develop robust intrinsic representations first, achieving a significantly superior AA NGL PPL of **2.35** (**Figure R14, Panel C2**). Thus, the stop-gradient is not merely a stability mechanism but a vital inductive bias for learning decoupled evolutionary representations.
>
> &nbsp;
>
> **W4: Weak $\rho \approx 0.085$ for G6.31 : does this overstate practical screening/therapeutic value?**
>
> We fully agree. To objectively quantify practical utility, we calculated Top-K Recall and Enrichment Factor at 10% (Enrich@10%) across all three DMS benchmarks (**Tables R3-R5**).
>
> **G6.31 (Table R3):** We agree that PRISM's Enrich@10% of 1.1x on G6.31 is insufficient for practical screening guidance. We will temper these claims in the camera-ready version.
>
> **Other Targets:** However, on targets where the zero-shot signal is more robust, PRISM delivers practically meaningful enrichment that directly demonstrates its screening utility: **CR9114 (Table R4):** 2.7x enrichment ($\rho = +0.393$), while all baselines yield anti-enrichment (0.2x–0.8x). **Trastuzumab (Table R5):** 3.0x enrichment ($\rho = +0.366$), on par with state-of-the-art baselines. We will add Tables R3-R5 with full Recall@K and Enrichment metrics to the camera-ready version.

---

> > ### Author Rebuttal · Reviewer_DViF · 2026-04-03
> >
> > Thank you for the response. Please see some additional questions/comments.
> >
> > (1) Is the fundamental reason for PRISM’s low enrichment efficiency on G6.31 that the decoupled GL/NGL architecture is inherently better at capturing epistatic interactions among multi-site mutations and evolutionary co-variation trajectories, whereas its advantages are substantially weakened when handling single-residue mutational scans that lack evolutionary context? Does this imply that the framework has a theoretical limitation in guiding local fine-tuning, such as single-point affinity maturation?

---

> > > ### Author Response · Authors · 2026-04-05
> > >
> > > We thank the reviewer for this insightful follow-up question. We find that the low enrichment on G6.31 primarily reflects the difficulty of predicting single-mutation effects in highly optimized antibody sequences, rather than a limitation of the architecture.
> > >
> > > G6.31 is a strongly affinity-matured antibody (Kd ≈ 0.4 nM), obtained through multiple rounds of phage display optimization and targeted mutagenesis (Lee et al., 2004; Fuh et al., 2006). The single-mutant dataset (Koenig et al., 2017) probes mutations around a sequence already near a local optimum, where most mutations are neutral or deleterious — resulting in a weak signal-to-noise ratio for zero-shot scoring. This is inherently challenging for any sequence model relying on mutant-centered likelihood scoring.
> > >
> > > **A more robust scoring strategy enabled by disentanglement.** Beyond identifying the challenge, PRISM’s factorized architecture offers a concrete path forward. By leveraging germline-centered likelihood scoring - computing variant likelihoods relative to the inferred germline reference rather than the wildtype - we observe a dramatic improvement on G6.31:
> > >
> > > | Model     | Spearman ρ  | Recall@1% | Recall@5% | Recall@10% | Enrich@10% |
> > > | --------- | ----------- | --------- | --------- | ---------- | ---------- |
> > > | ESM2-650M | -0.1301     | 0.047     | 0.065     | 0.171      | 1.7x       |
> > > | AntiBERTy | +0.0712     | 0.023     | 0.079     | 0.138      | 1.4x       |
> > > | Sapiens   | +0.0101     | 0.023     | 0.070     | 0.121      | 1.2x       |
> > > | **PRISM(Wildtype-Centered)** | **+0.0853** | **0.023** | **0.070** | **0.114**  | **1.1x**   |
> > > | **PRISM(Germline-Centered)** | **+0.1747** | **0.023** | **0.136** | **0.206**  | **2.1x**   |
> > > | AbLang2   | -0.0190     | 0.000     | 0.033     | 0.110      | 1.1x       |
> > > | ESM2-35M  | -0.2038     | 0.023     | 0.070     | 0.107      | 1.1x       |
> > > | Random    | —           | 0.010     | 0.050     | 0.100      | 1.0x       |
> > >
> > > Germline-centered scoring nearly doubles enrichment (1.1× → 2.1×) and more than doubles the Spearman correlation ($\rho$: 0.085 → 0.175) on G6.31, while maintaining comparable performance on multi-mutation benchmarks  (CR9114: $\rho$ = 0.395 → 0.421; Trastuzumab: $\rho$ = 0.343 → 0.368). This confirms that germline-centered scoring is a uniformly more robust default — substantially improving low-signal single-mutation regimes without degrading performance on benchmarks where the original approach already excelled. This capability is uniquely enabled by PRISM's disentangled architecture - while any MLM can in principle substitute a germline sequence as context, PRISM's factorized vocabulary enables explicit isolation of the NGL likelihood component, providing a cleaner signal than simply swapping the input sequence.Crucially, this approach does not require external germline assignment tools — PRISM's Origin Head recovers germline sequences with ~95% accuracy on OAS sequences including matured B cells, making germline-centered scoring broadly applicable even when germline annotations are unavailable.
> > >
> > > **Broader validation on single-mutation landscapes (FLAb).** To confirm that the original wildtype-centered results reported in the manuscript are not fundamentally flawed, we evaluated zero-shot prediction on the FLAb benchmark (Chungyoun & Gray, 2025), spanning 19 antibodies and 4,146 single-point variants. Even with the wildtype-centered scoring used in the manuscript, PRISM is the **only** model that maintains a consistently positive predictive signal:
> > >
> > > | Metric       |      PRISM | ESM2-35M | ESM2-650M | AbLang2 | AntiBERTy | Sapiens |
> > > | :----------- | ---------: | -------: | --------: | ------: | --------: | ------: |
> > > | Mean rho     | **+0.048** |   -0.003 |    -0.152 |  -0.113 |    -0.108 |  -0.099 |
> > > | Median rho   | **+0.045** |   -0.034 |    -0.027 |  -0.030 |    -0.071 |  -0.037 |
> > > | Mean Rank    |   **2.67** |     3.26 |      3.74 |    4.00 |      3.42 |    3.74 |
> > > | Positive rho |  **11/19** |     7/19 |      5/19 |    9/19 |      8/19 |    8/19 |
> > >
> > > We acknowledge that the predictive signal is naturally stronger with richer evolutionary context from multiple mutations, as expected for any sequence model. Nonetheless, PRISM's consistent positive correlations, where all baselines trend negative, demonstrate a meaningful advantage regardless of mutational load.
> > >
> > > In summary, PRISM does not exhibit a theoretical limitation for single-point affinity maturation. The per-target variability observed in the manuscript reflects the choice of scoring strategy, not an architectural constraint. Germline-centered scoring — uniquely enabled by PRISM's factorized architecture — provides a uniformly more robust alternative, and the wildtype-centered approach already outperforms all baselines in aggregate across 19 diverse single-mutation datasets. We will incorporate both the germline-centered scoring results and FLAb analysis into the revised manuscript.

---

### Official Review · Reviewer_1pKg · 2026-03-13

**Soundness:** 4
**Presentation:** 2
**Significance:** 3
**Originality:** 3
**Overall Recommendation:** 5
**Confidence:** 4

**Summary:**

The work proposes a more fine-grained antibody representation by explicitly separating germline (GL) and non-germline (NGL) residues with a disentangled vocabulary, resulting in improved performance on antibody modeling, binding affinity and developability prediction.

**Compliance With Llm Reviewing Policy:**

Affirmed.

**Final Justification:**

I will raise my score as my concerns have been fully addressed. This is a solid work, though it still needs modifications in presentation.

**Key Questions For Authors:**

See above.

**Limitations:**

yes

**Strengths And Weaknesses:**

**Strengths**
- The biological perspective of GL/NGL for antibodies is inspiring.
- The experiments are solid and comprehensive, showing the distinct contributions of NGL and GL residues to binding affinity and developability, respectively.
- The paper is clearly written and well organized, with informative illustrations.

**Weaknesses and questions**
1. The presentation is more suitable for biology journals. Enssential information, such as the training objective and brief introduction of downstream datasets, should be included in the main text rather than appendix. The experiments section is overly long while description for methodology remains limited.
2. I wonder about the p90 noise calibration procedure used in dataset construction. My understanding is that background sequencing errors tend to occur randomly and uniformly. For example, a sequence with 10 mutations may have 1-3 errors, whereas a sequence with only 3 mutations may consist entirely of real NGL mutations. Such noise seems inherently difficult to distinguish. Therefore, does removing sequences with fewer mutations eliminate sequencing errors? In addition, in Figure S1F, most naive cells appear to exhibit more than 3 mutations, which seems inconsistent with "90% of Naive cells exhibit ≤ 3 mutations" (line 645). Do I misunderstand the y-axis?
3. How much overlap exists between NGL residues and CDRs? As many antibody generation methods that design CDR conditioned on fixed FR, what would happen if all CDR residues were marked as NGL while all FR residues were masked as GL?
4. It would be helpful to compare PRISM with IgLM, an autoregressive generative model for antibody modeling and design, which is already cited in the introduction.
5. Since the the pre-training stage uses non-paired data, is the input format the same as `<CLS>{Heavy}<CLS>{Light}<EOS>` with placeholders for missing chains, or independent single sequences?
- Minor questions:
  - PRISM is trained with a vocabulary of size 53. Besides the 40 standard amino acids in lowercase and uppercase, what are the remaining 13 special tokens?
  - Compared to "generation" in the conventional sense (i.e., full-sequence probability modeling), the recovery evaluation in Section 4.2 may not strictly reflect generative capability. Terms such as *sequence modeling* or *evolutionary modeling ability* might be more precise.
  - Figure 3 missing C and D panel labels.
  - In Figure 4C and 4D, why are some bars labeled with negative values but plotted with positive heights?

---

> ### Author Rebuttal · Authors · 2026-03-30
>
> We sincerely thank the reviewer for the constructive feedback. For all supporting figures/tables, please refer to: [Link](https://anonymous.4open.science/r/PRISM_Rebuttal-13CC/README.md)
>
> **W1: Paper structure: methodology under-described, experiments overly long, essential info relegated to appendix?** \
> We agree that methodological details must be prioritized in the main text for an ML venue like ICML. We will condense Section 4 to integrate our factorized focal loss and token-reweighting strategy used during pre-training into the main text. A brief subsection outlining Thera-SAbDab and DMS benchmarks will also be added before the experiments.
>
> &nbsp;
>
> **W2: Does p90 noise calibration truly remove sequencing errors, and is Figure S1F consistent with the "$\leq$3 mutations" claim?** \
> We agree background sequencing errors exist across all sequences, and our threshold does not eliminate noise within highly mutated sequences.
> Our primary objective was mitigating germline bias rather than error correction. Sequences with $\leq 3$ mutations are statistically indistinguishable from naive B-cells (unmutated germline + baseline sequencing noise). Including them forces the model to redundantly train on pure germline templates lacking antigen-specific information. By discarding this high-volume, low-information baseline, we encourage the model to focus on learning rare, meaningful NGL mutations. We will clarify this rationale in Appendix A.
> Regarding Figure S1F: The y-axis is "Probability Density," not absolute counts. The distribution is heavily right-skewed with the 90% of the mass tightly packed between 0 and 3. We will update the plot to resolve this ambiguity.
>
> &nbsp;
>
> **W3: How much do NGL residues overlap with CDRs, and what if CDR=NGL / FR=GL exactly?** \
> We first quantify the overlap between NGL residues and CDR regions, and then evaluate a region-level heuristic where all FR residues are forced to GL and all CDR residues are forced to NGL.
> Analysis of 63M training sequences (**Figure R9a**) shows that NGL residues are only partially enriched in CDRs: **85.5%** of CDR residues are in fact germline (GL), while **8.8%** of FR residues are NGL. Thus, a binary mapping of FR = GL and CDR = NGL introduces mislabeling. We compared PRISM’s per-residue predictions to a **region-forced** condition on a 22K test set to test the impact of this mislabeling. This heuristic substantially degrades pseudo-perplexity (**Figure R9b–c**).
> - **Misassigned residues (CDR-GL & FR-NGL):** Forcing NGL predictions onto conserved CDR residues and forcing GL predictions onto FR positions causes severe pseudo-perplexity degradation (**8.7$\times$ Heavy, 6.9$\times$ Light**).
> - **FR-GL & CDR-NGL Proxy:** Even when the heuristic assigns the "correct" label, pseudo-perplexity still worsens by 1.1-3.8$\times$, indicating that local misassignments propagate globally through the sequence likelihood.
> We will include this ablation study in the Appendix to demonstrate that PRISM accurately distinguishes residue-level labels rather than relying on regional approximations.
>
> &nbsp;
>
>
> **W4: Why is IgLM not included as a baseline?** \
> We appreciate the reviewer’s suggestion to include the autoregressive antibody language model IgLM. IgLM exhibits a high pseudo-perplexity for CDR3 non-germline mutations similar to previous baselines (**Figure R10**). We trained a linear probe that discriminates gl vs ngl for each position on IgLM last hidden state embeddings, which achieved a PR-AUC of 0.3244 and an F1 Score of 0.3124, both significantly less than PRISM's performance. While IgLM has previously documented controllable generation strengths, IgLM exhibits the same germline bias we observed in previous baselines.
> For binding affinity, IgLM's zero-shot performance was inconsistent across targets (Spearman $\rho = -0.37$, $-0.07$, and $0.19$ for CR9114, G6.31, and Trastuzumab). For developability, IgLM showed positive zero-shot correlations for immunogenicity ($\rho = 0.25$) and thermostability ($\rho = 0.06$) and negative correlations for other properties. Overall, performance lagged behind PRISM (**Figure R11**).
> Generation results are included under Reviewer xKEM section R1.
>
> &nbsp;
>
>
> **W5: What is the input format for unpaired sequences during pre-training?** \
> Unpaired pre-training strictly used independent sequences formatted as `<CLS>sequence<EOS>`. The paired format was used exclusively for fine-tuning. We will clarify this in Appendix A.4.
>
> &nbsp;
>
>
> **W6: Minor: 13 non-AA special tokens; "recovery" vs. generative capability terminology; Figure 3 missing C/D labels; Figure 4C/D bar sign inconsistency.** \
> The 13 special tokens will added to Appendix B, the "generation" terminology will be revised to "sequence modeling" in Section 4.2, Figure 3 missing labels will be added, and text annotations in Figure 4C-D will be corrected.

---

> > ### Author Rebuttal · Reviewer_1pKg · 2026-04-01
> >
> > I will raise my score to 5 as my concerns are fully resolved. I hope the authors can adjust the paper's organization in the revised version.

---

> > > ### Author Response · Authors · 2026-04-02
> > >
> > > We sincerely thank you for your time, the constructive discussion, and for raising your score. We are very glad to hear that our rebuttal has fully resolved your concerns.
> > >
> > > We completely agree with your suggestion regarding the paper's organization. We promise to carefully adjust the structure and flow of the manuscript in the revised version to ensure it is much clearer and easier to read.
> > >
> > > Thank you once again for your valuable insights and support!

---

### Official Review · Reviewer_uqCY · 2026-03-13

**Soundness:** 3
**Presentation:** 2
**Significance:** 2
**Originality:** 2
**Overall Recommendation:** 3
**Confidence:** 3

**Summary:**

This paper introduces PRISM, a germline-aware antibody language model that explicitly separates germline and non-germline residues through a factorized vocabulary and a sequential gated architecture. The model first predicts evolutionary origin and then amino-acid identity, using a gating mechanism to control how strongly origin information constrains token prediction. The paper argues that this design reduces germline bias in antibody modeling. Experiments show improvements on disentanglement metrics, pseudo-perplexity, therapeutic sequence reconstruction, and several zero-shot affinity and developability benchmarks.

**Compliance With Llm Reviewing Policy:**

Affirmed.

**Key Questions For Authors:**

1. How robust are the gains to alternative or noisier GL/NGL labels? If they persist under less curated supervision, the method would appear substantially more general.
2. Can the authors add a direct controllable-generation experiment, ideally showing affinity/developability trade-off control under a fixed decoding budget? This would strengthen the significance of the paper.
3. Are the baseline comparisons still favorable after tightly matching model size, training data, and antibody-specific supervision? This would better isolate the contribution of the proposed factorization.
4. Do the reported affinity and developability gains remain consistent across a broader set of benchmarks or per-assay statistical tests? This would clarify robustness.

**Limitations:**

Yes. The paper includes explicit limitations and impact sections. That said, the authors could still strengthen them by stating more plainly that controllable generation is not directly validated, that the current benchmarks are curated and relatively narrow, and that performance may depend on the quality of GL/NGL supervision.

**Strengths And Weaknesses:**

Strengths:
1. The paper addresses a real weakness of antibody language models: over-reliance on germline signal and under-modeling of rare functional mutations.
2. The factorized vocabulary and gated design are sensible, and the ablations suggest the gain is not just a tokenization artifact.
3. Reproducibility is relatively strong: the paper provides code/data availability, training details, and basic statistical reporting.

Weaknesses:
1. The main claims are supported mostly by representation and zero-shot benchmark evaluations, while controllable generation is not directly tested.
2. Important details for judging generality, including label construction and baseline comparability, remain appendix-heavy.
3. The controllable-generation claim is stronger than the direct evidence currently shown.

---

> ### Author Rebuttal · Authors · 2026-03-30
>
> We sincerely thank the reviewer for the constructive feedback. For all supporting figures/tables, please refer to: [link](https://anonymous.4open.science/r/PRISM_Rebuttal-13CC/README.md)
>
> **W1: How robust are gains to noisier GL/NGL labels?** \
> We conducted a noise injection experiment to assess robustness to increasingly noisy GL/NGL supervision. For a comprehensive breakdown of model performance including psuedo-perplexity (PPL) and zero-shot correlations after noise injection, please refer to response to **Reviewer xKEM** section **W2** (**Figure R19-22**).
>
> **TL;DR:** PRISM retains its substantial performance advantage over baselines under high noise levels. This demonstrates our approach can robustly disentangle competing signals even when supervision is imperfect, rather than overfitting to label noise. More broadly, this suggests our approach could be used in settings where labels are less uncertain than GL/NGL.
>
> &nbsp;
>
> **W2: Can authors add a direct controllable-generation experiment?** \
> We appreciate this valuable insight for improving the paper's significance. We performed Psuedo-Log Likelihood (PLL)-guided generation across 3 targets (G6.31, Trastuzumab, CR9114) under a fixed decoding budget. For evaluation protocols and Pareto front plots (**Figure R15-18**), please refer to our response to **Reviewer xKEM** section **W1**.
>
> **TL;DR:** The observed Pareto fronts show PRISM enables explicit control over the affinity–developability trade-off during generation. By modulating GL vs. NGL likelihood contributions, PRISM reweights germline conservation against antigen-specific mutations, enabling a controllable multi-objective antibody sequence optimization process.
>
> &nbsp;
>
> **W3: Are baseline comparisons favorable after matching model size, training data, and antibody-specific supervision?**  \
> We trained a new baseline **PRISM-less** that is identical to PRISM in encoder architecture, data, and two-stage training but without the multi-head GL/NGL factorization (and with a standard MLM objective) to isolate the contribution of the proposed factorization. With architecture, data, and training controlled, the gains are attributable to GL/NGL factorization. The observed sign reversals indicate that, without factorization, the model systematically confounds evolutionary conservation with functional fitness:
>
> * **Perplexity (Figure R1):** Without factorization, modeling of NGL mutations degrades substantially. PRISM-less CDR3 NGL PPL is 56.5/72.3 (Heavy/Light) vs. 8.7/5.2 for PRISM, with 42% of NGL tokens exceeding PPL > 20 (PRISM: 8.8%).
> * **GL/NGL Discrimination (Figure R2):** A linear probe performs worse on PRISM-less (F1 = 0.784, PR-AUC = 0.924) than PRISM (F1 = 0.896, PR-AUC = 0.980), indicating reduced linear separability and entangled GL/NGL representations.
> * **Binding (Figure R3):** Zero-shot correlations collapse without factorization. PRISM-less exhibits sign reversal on 2/3 DMS targets (G6.31: $\rho = -0.115$, CR9114: $\rho = -0.409$) and mean $\rho = -0.044$ on FLAb2 (PRISM: +0.066).
> * **Developability (Figure R4):** PRISM-less yields negative $\rho$ on 3/6 Ginkgo properties and strong degradation on FLAb2 thermostability (-0.276) and expression (-0.530). The exception is immunogenicity (ADA), where sequence novelty makes PPL a reasonable proxy.
>
> &nbsp;
>
> **W4: Do the reported zero-shot gains remain consistent across a broader set of benchmarks?** \
> We conducted a comprehensive evaluation on the FLAb benchmark with 45 deep mutational scanning datasets for binding affinity and 30 developability assays across 5 property categories (Chungyoun & Gray, 2025). PRISM consistently outperforms baselines.
>
> **Binding:** PRISM is the only model with a positive mean Spearman $\rho$ (+0.066), achieves positive correlations on 60% of proteins (vs. 38-51%), and wins 57% of pairwise comparisons (**Figure R5**). It significantly outperforms ESM2-35M (p = 0.027) and ESM2-650M (p = 0.004) in pairwise Wilcoxon signed-rank tests, with gains distributed across proteins rather than driven by outliers (**Figure R6**).
>
> **Developability:** PRISM predicts the correct direction for 74% of assays (vs. 32-48%) and is the only model with a positive mean directed $\rho$ (+0.076) (**Figure R7**). It shows strong performance in key properties including thermostability ($\rho = +0.392$) and immunogenicity ($\rho = +0.325$) while maintaining correct directional trends in other categories where baselines fail (**Figure R8**).
>
> We will include full results and statistical breakdowns in the camera-ready version.

---

> > ### Author Rebuttal · Reviewer_uqCY · 2026-04-03
> >
> > Thank you for answering my questions.

---

> > > ### Author Response · Authors · 2026-04-03
> > >
> > > We sincerely thank the reviewer for acknowledging that the concerns have been fully resolved. Given this, we kindly ask whether the reviewer might consider revisiting the score to better reflect the current assessment of the paper. We believe this would also help the AC in calibrating the final decision.

---

### Decision · Program_Chairs · 2026-04-30

**Decision:**

Accept (regular)

**Comment:**

This paper introduces PRISM, a germline-aware antibody language model that explicitly separates germline (GL) and non-germline (NGL) residues through a factorized vocabulary and a sequential gated architecture (Origin, AA, Alpha heads). The model first predicts evolutionary origin, then amino-acid identity, using a gating mechanism to control how strongly origin information constrains token prediction. Experiments show improvements on disentanglement metrics, pseudo-perplexity, therapeutic sequence reconstruction, and zero-shot affinity and developability benchmarks.

**Strengths:**

- Well motivated study：Explicitly decoupling evolutionary conservation from functional variation via a separated vocabulary is a significant paradigm shift from implicit language modeling. (uqCY, 1pKg, DViF, xKEm)
- The sequential conditional setup provides an intuitive mechanism for disentanglement and controllability. (1pKg, DViF, xKEm)
- Comprehensive evaluation across Thera-SAbDab, DMS affinity datasets, and developability metrics demonstrates real-world utility. (uqCY, 1pKg, DViF)
- Strong reproducibility: code/data availability, training details, and statistical reporting provided. (uqCY)

**Weaknesses (initial, before rebuttal):**

- Controllable generation not directly tested. (uqCY, xKEm)
- Reliance on accurate GL/NGL labeling; sensitivity to annotation errors not fully explored. (uqCY, xKEm)
- Core ML contribution (vocabulary expansion + gating) unclear whether generalizable beyond antibodies to other hierarchical generation tasks. (DViF)
- G6.31 zero-shot correlation remains very weak (ρ≈0.085); practical screening utility may be overstated. (DViF)
- Manuscript organization issue: methodology under-described in main text; essential details relegated to appendix. (1pKg)
- IgLM baseline missing from comparisons. (1pKg)

**Additional Comments on Reviewer Discussion:**
The authors conducted extensive additional experiments to address the reviewers' concerns, the majority of which have been satisfactorily resolved. Reviewer DViF raised a follow-up question regarding the weak zero-shot correlation on the G6.31 benchmark; the authors subsequently provided new experiments—including germline-centered scoring—that substantially improved performance on this target.

Overall, the study is well-motivated and makes valuable contributions to antibody language modeling. I recommend a weak accept to accept for this study.